



# The global water resources and use model WaterGAP v2.2d: Model description and evaluation

Hannes Müller Schmied[1,2], Denise Cáceres[1], Stephanie Eisner[3], Martina Flörke[4], Claudia Herbert[1], Christoph Niemann[1], Thedini Asali Peiris[1], Eklavyya Popat[1], Felix Theodor Portmann[1], Robert Reinecke[5], Maike Schumacher[6,7], Somayeh Shadkam[1], Camelia-Eliza Telteu[1], Tim Trautmann[1], and Petra Döll[1,2]

[1]Institute of Physical Geography, Goethe University Frankfurt, Frankfurt am Main, Germany
[2]Senckenberg Leibniz Biodiversity and Climate Research Centre (SBiK-F), Frankfurt am Main, Germany
[3]Norwegian Institute of Bioeconomy Research (NIBIO), Ås, Norway
[4]Engineering Hydrology and Water Resources Management, Ruhr-University of Bochum, Bochum, Germany
[5]International Centre for Water Resources and Global Change (UNESCO), Federal Institute of Hydrology, Koblenz, Germany
[6]Institute of Physics and Meteorology, University of Hohenheim, Stuttgart, Germany
[7]Computational Science Lab (CSL) at the University of Hohenheim, Germany

**Correspondence:** hannes.mueller.schmied@em.uni-frankfurt.de

**Abstract.** WaterGAP is a global hydrological model that quantifies human use of groundwater and surface water as well as water flows and water storage and thus water resources on all land areas of the Earth. Since 1996, it has served to assess water resources and water stress both historically and in the future, in particular under climate change. It has improved our understanding of continental water storage variations, with a focus on overexploitation and depletion of water resources. In
this paper, we describe the most recent model version WaterGAP 2.2d, including the water use models, the linking model that computes net abstractions from groundwater and surface water and the WaterGAP Global Hydrology Model WGHM. Standard model output variables that are freely available at a data repository are explained. In addition, the most requested model outputs, total water storage anomalies, streamflow and water use, are evaluated against observation data. Finally, we show examples of assessments of the global freshwater system that can be done with WaterGAP2.2d model output.

## 1 Introduction

In a globalized world with large flows of virtual water between river basins and international responsibilities for sustainable development of the Earth systems and its inhabitants, quantitative estimates of water flows and storages and of water demand by humans and freshwater biota *on all continents of the Earth* form the basis for a sustainable water, and more broadly, Earth system management. During the last three decades, global hydrological models (GHMs) have been developed and continually
improved to provide this information. They enable the determination of the spatial distribution and temporal development of water resources and water stress for both humans and other biota under the impact of global change (including climate change). In addition, global-scale knowledge about water flows and storages on land is necessary to understand the Earth



System, including interactions with the ocean and the atmosphere as well as gravity distribution and crustal deformation (affecting GPS).

Such models are frequently used in large scale assessments, such as the Environmental Performance Index (EPI), the assessment of virtual water flows for products (Hoff et al., 2014) within the framework of the Intergovernmental Panel of Climate Change and the assessment of impacts based on scenarios for a sustainable future (as e.g., the Sustainable Development Goals). Furthermore, global-scale modelling of water use and water availability is frequently used to evaluate large scale water issues, for example water scarcity and droughts (Meza et al., 2020; Döll et al., 2018; Veldkamp et al., 2017).

Some of these models are contributing to the Inter-Sectoral Impact Modelling Intercomparison Project (ISIMIP) (Frieler et al., 2017) where the focus is on both, the model evaluation / improvement and the impact assessment of anthropogenic changes such as human water use or climate change. A series of evaluation exercises (Veldkamp et al., 2018; Zaherpour et al., 2018; Wartenburger et al., 2018) shows that proper simulation is challenging due to uncertain process representation at the given resolution, input data uncertainty and unequal data availability in terms of spatial and temporal distribution, e.g. river 30  discharge observations (Coxon et al., 2015; Wada et al., 2017; Döll et al., 2016). In this context, a proper model description is of great value for a better understanding of the process representation and parameterization of such models, and a related work is in progress (Telteu et al., 2020).

A continuous improvement of process representations in GHMs is required to reduce uncertainty in assessments of water resources over historical periods (Schewe et al., 2019) and thus increase confidende in future projection assessments. In the 35  recent past, some of the GHM approaches consider new processes as e.g. the $CO_2$ fertilization effect (Schaphoff et al., 2018a, b) or gradient-based groundwater models (de Graaf et al., 2017; Reinecke et al., 2019). Improved methods for the estimations of agricultural and other water use (Flörke et al., 2013; Siebert et al., 2015) have been developed and total water storage data from satellite observations are being increasingly employed either for evaluation (Scanlon et al., 2018, 2019) or calibration/assimilation of models (Eicker et al., 2014; Döll et al., 2014; Schumacher et al., 2018). Ultimately, there are attempts 40  to achieve a finer spatial resolution than the typically used $0.5° \times 0.5°$ grid cell (Wood et al., 2011; Bierkens et al., 2015; Sutanudjaja et al., 2018; Eisner, 2015).

The GHM WaterGAP, which has been developed since 1996, is one of the pioneers in this field. A large number of model versions of WaterGAP 2, the model variant with a spatial resolution of $0.5° \times 0.5°$, have been developed and applied. The major model purpose was to quantify global scale water resources with specific focus on anthropogenic inventions due to 45  human water use and man-made reservoirs, to assess water stress. Furthermore, a lot of effort have been assigned to specific water storages like groundwater, lakes and wetlands. In the previously mentioned evaluation studies, WaterGAP has been qualified as a robust and qualitatively good-performing model in those key issues and for most climate zones worldwide.

Since the last complete model description of WaterGAP 2.2 (Müller Schmied et al., 2014), a number of modifications and improvements have been achieved. To be able to follow these changes and to transparently understand the process representa- 50  tion, a new model description can guide model output data users, especially in case of discrepant model outputs from a GHM ensemble approach, and the GHM developing community in general. Hence, the aim of this paper is to provide an overview of the newest model version WaterGAP 2.2d by





1. comprehensively describing the full model including all developments since WaterGAP 2.2 (Müller Schmied et al., 2014),

2. showing and discussing standard model output,

3. providing insights into model evaluation and

4. giving guidance for the users of model output.

The framework of WaterGAP 2.2d is presented in Sect. 2, followed by the in-depth description of the water use models (Sect. 3) and the global hydrological model (Sect. 4). The description of standard model outputs is given in Sect. 5 including caveats

of using the model outputs. In Sect. 6, model output is compared against multiple observation-based data sets, followed by typical model applications in Sect. 7. The discussion section highlights the current fields of scientific use of WaterGAP and shows the way towards the next model versions (Sect. 8) and is followed by the conclusions (Sect. 9).

## 2   WaterGAP 2 framework

WaterGAP 2 consists of three major components, the global water use models, the linking model Groundwater-Surface Water

Use (GWSWUSE) and the WaterGAP Global Hydrology Model (WGHM) (Fig. 1). Five global water use models for the sectors irrigation (Döll and Siebert, 2002; Portmann, 2017), livestock, domestic, manufacturing and cooling of thermal power plants (Flörke et al., 2013) compute consumptive water use and, in the case of the latter three sectors, also withdrawal water uses. Consumptive water use refers to the part of the withdrawn (=abstracted) water that evapotranspirates during use. Whereas the output of the Global Irrigation Model (GIM) is available at monthly resolution, annual time series are calculated by all non-

irrigation water use models (Sects. 3.1, 3.2). The linking model GWSWUSE serves to distinguish water use from groundwater and from surface water bodies (Sect. 3.3). It computes withdrawal water uses from and return flows to the two alternative water sources to generate monthly time series of net abstractions from surface water ($NA_s$) and from groundwater ($NA_g$) (Döll et al., 2012, 2014). These time series are input to the WGHM, affecting the daily water flows and storages computed by it (Sect. 4).

## 2.1   Spatial coverage and climate forcings

The WaterGAP 2 framework operates on the so-called CRU land-sea mask (Mitchell and Jones, 2005), which covers the global continental area (including small islands and Greenland but excluding Antarctica) with in total 67420 grid cells, each $0.5° \times 0.5°$ in size which represents approx. $55\,\mathrm{km} \times 55\,\mathrm{km}$ at the equator. WaterGAP uses the continental area of the grid cell, which is defined as the cell area (calculated with equal area cylindrical projection) minus the ocean area with the borders according

to the ESRI *worldmask* shapefile (ArcGIS - Worldmask). The continental area comprises land area and surface water body area (lakes, reservoirs and wetlands only; river area is not considered). Since WaterGAP 2.2a, surface water body areas, and consequently land area, are dynamic and are updated in each time step.

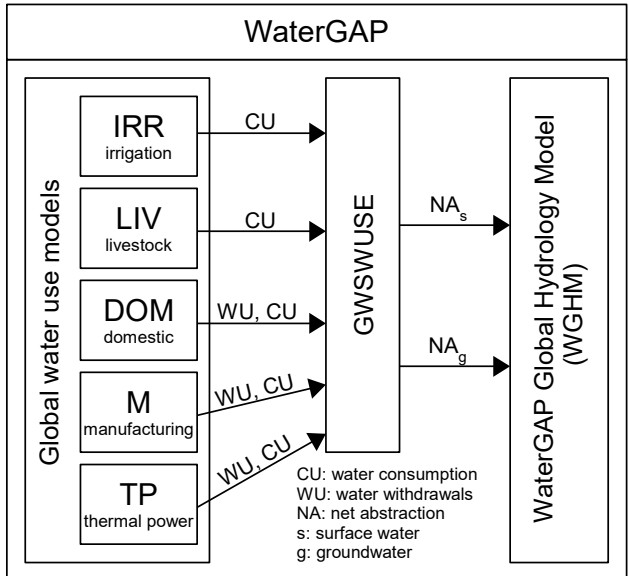

**Figure 1.** The WaterGAP 2.2d framework with its water use models and the linking module GWSWUSE that provides net water abstraction from groundwater and surface water as input to the WaterGAP Global Hydrology Model (WGHM). Figure adapted from Müller Schmied et al. (2014).

Both GIM and WGHM use meteorological input data that consist of air temperature, precipitation, downward shortwave radiation and downward longwave radiation, all with daily temporal resolution. Various global meteorological data sets (here-

after referred to as climate forcings) were developed by the meteorological community at the $0.5° \times 0.5°$ spatial resolution, such as WFD (Weedon et al., 2011), WFDEI (Weedon et al., 2014), GSWP3 (Hyungjun, 2014), the Princeton meteorological forcing (Sheffield et al., 2006) and recently ERA5 (ERA5) and WFDE5 (Cucchi et al., 2020). Alternative climate forcings may lead to significantly different WaterGAP outputs (Müller Schmied et al. (2016a)).

## 2.2 Modifications of WaterGAP since version 2.2

The general framework of WaterGAP 2.2d does not differ from model version 2.2 described in Müller Schmied et al. (2014). Improvements of water use modelling since WaterGAP 2.2 include, among others, deficit irrigation in regions with groundwater depletion (Sect. 3.3) as well as integration of the Historical Irrigation Data set (HID), which provides the historical cell-specific development of area equipped for irrigation (Siebert et al., 2015). Major improvements in WGHM include 1) a consistent river-storage-based method to compute river flow velocity, 2) simulation of land area dynamics in response to varying areas

of lakes, reservoirs and wetlands, 3) groundwater recharge from these surface water bodies in (semi)arid grid cells, 4) if daily precipitation is below a threshold value, the potential groundwater recharge remains in the soil and does not (as in WaterGAP 2.2) become surface runoff, 5) return flows to groundwater from surface water use are corrected (by adjusting $NA_g$) by the amount of $NA_s$ that cannot be satisfied and 6) the integration of reservoirs by taking into account their commissioning year





(and not assuming anymore that they have existed during the whole study period). Other changes concern model calibration or

consist in inclusion of new data sets and software improvements. A complete list of modifications of WaterGAP 2.2d compared to WaterGAP 2.2 is provided in Appendix A.

## 3 WaterGAP water use models

### 3.1 Global Irrigation Model

Irrigation accounts for 60-70% of global withdrawal water uses and 80-90% of global consumptive water uses, and for even

larger shares in almost all regions with severe water stress and groundwater depletion (Döll et al., 2012, 2014). Therefore, a reliable simulation of irrigation water use is decisive for the quality of WaterGAP simulations of streamflow and water storage in groundwater and surface water bodies as well as for the reliability of computed water stress indicators. Based on information on irrigated area and climate for each grid cell, GIM computes first cell-specific cropping patterns and growing periods and then irrigation consumptive water use (ICU), distinguishing only rice and non-rice crops (Döll and Siebert, 2002). ICU can be

regarded as the net irrigation requirement that would lead to optimal crop growth.

### 3.1.1 Computation of cropping patterns and growing periods of rice and non-rice crops

The cropping pattern for each cell with irrigated cropland describes if only rice, non-rice crops or both are irrigated during either one or two growing seasons. The growing period for both crop types is assumed to be 150 days. Seventeen cropping patterns are possible including simple variants (e.g., one cropping season with non-rice on the total irrigated area) and complex

variants (non-rice after rice on one part of the total irrigated area and non-rice after non-rice on the other). The following data are used to model the cropping pattern: total irrigated area, long-term average temperature and soil suitability for paddy rice in each cell, harvested area of irrigated rice in each country, and cropping intensity in each of 19 world regions. In a second step, the optimal start date of each growing season is computed for each crop. To this end, each 150-day period within a year is ranked based on criteria on long-term average temperature, precipitation and potential evapotranspiration provided in Döll

and Siebert (2002). The most highly ranked 150-day period(s) is/are defined as growing season(s).

### 3.1.2 Computation of consumptive water use due to irrigation

GIM implements the Food and Agriculture Organization of the United Nations (FAO) CROPWAT approach of Smith (1992) to compute crop-specific ICU per unit irrigated area ($\mathrm{mm\,d^{-1}}$) during the growing season as the difference between crop-specific optimal evapotranspiration $E_{pot_c}$ and effective precipitation $P_{irri,eff}$ if the latter is smaller than the former, with

$$ICU = \begin{cases} E_{pot_c} - P_{irri,eff} & E_{pot_c} > P_{irri,eff} \\ 0 & \text{otherwise} \end{cases} \tag{1}$$



where $E_{pot_c}$ is the product of potential evapotranspiration $E_{pot}$ and the dimensionless crop coefficient $k_c$ which depends on the crop and the crop development stage (Döll and Siebert, 2002). As a standard, $E_{pot}$ is calculated according to Eq. (7). $P_{irri,eff}$ is the fraction of the total precipitation $P$ (including rainfall and snowmelt) that is available to plants and is computed as a simple empirical function of precipitation. Equation (1) is implemented with a daily time step, but to take into account the storage capacity of the soil and to remain consistent with the CROPWAT approach, daily precipitation values are averaged over 10 days, except for rice-growing areas in Asia, where the averaging period is only 3 days to represent the limited soil water storage capacity in case of paddy rice (Döll and Siebert, 2002).

### 3.1.3 Irrigated area

In the standard version of WaterGAP 2.2d, irrigated area per grid cell used in GIM is based on the Historical Irrigation Data set (HID) (Siebert et al., 2015), which provides area equipped for irrigation (AEI) in 5 arc-min grid cells for 14 time slices between 1900 and 2005. HID data are aggregated to $0.5° \times 0.5°$ and temporally interpolated to obtain an annual time series of AEI. Cropping patterns and growing periods are generated for every year, with an individual combination of year-specific AEI and harvested area of rice and the respective 30-year climate averages, which are then used to calculate ICU for every day of the same year. Harvested area of rice per country from the MIRCA2000 data set, representative for the year 2000 (Portmann et al., 2010), is scaled according to annual AEI country totals, ensuring consistency to AEI.

To take into account that not the whole AEI is actually used for irrigation in any year, country-specific values of the ratio of area actually irrigated (AAI) to AEI are used to estimate AAI in each grid cell. AAI is then applied for calculating the consumptive irrigation water use in volume per time. AAI/AEI ratios were derived from the Global Map of Irrigation Area (GMIA) for 2005 (Siebert et al., 2013). In addition, to estimate AAI from 2006 to 2016, we used country-specific AAI for 2006-2010 from the AQUASTAT database of the FAO, other international organizations and national statistical services (e.g., EUROSTAT and USDA). For the other countries, AAI of 2005 was assumed for 2006-2016. For all 2011-2016, AAI was assumed to remain at the 2010 value everywhere.

Alternatively, as in previous WaterGAP versions, GIM in WaterGAP 2.2d can be executed based on a temporally constant dataset of AEI per grid cell, e.g. the Global Map of Irrigation Area GMIA for 2005 (Siebert et al., 2013). Cropping patterns and growing periods are then computed for AEI and harvested area of rice in a reference year and the pertaining 30-year average climate. For more details and application examples, we refer to Portmann (2017) and Döll and Siebert (2002).

### 3.2 Non-irrigation water uses

Although irrigation water use is the dominant water use sector globally, non-irrigation water uses, particularly in terms of withdrawal water uses, play a major role in Europe and America (FAO - AQUASTAT Main Database). Competition between agricultural and non-agricultural water uses are not uncommon (Flörke et al., 2018) and the estimation of water demands become even more crucial when water resources are scarce. Statistical information on withdrawal water uses and consumptive water uses for domestic, industrial and livestock purposes are difficult to obtain on a country basis since no comprehensive global database does exist. However, the FAO collects relevant water-related data from national statistics and reports to provide





a comprehensive view on the state of sectoral water uses. Unfortunately, the database lacks data in space and time and hence

modelling is of importance to fill these gaps (Flörke et al., 2013).

### 3.2.1    Livestock

Withdrawal water uses for livestock are computed annually by multiplying the number of animals per grid cell by the livestock-specific water use intensity (Alcamo et al., 2003). The number of livestock are taken from FAOSTAT. It is assumed that the withdrawal water uses for livestock are equal to their consumptive water use.

### 3.2.2    Domestic

Domestic water use comprises withdrawal water uses and consumptive water uses of households and small businesses and is estimated on a national level. The main concept is to first compute the domestic water use intensity $\mathrm{m^3\,cap^{-1}\,yr^{-1}}$ and then to multiply this by the population of water users in a country. The domestic water use intensity is expressed by a sigmoid curve which indicates how water use intensity (per capita water use) changes with income (GDP per cap) and is derived

from historical data on a national or regional level (Flörke et al., 2013). Besides changes driven by income and population, technological changes are considered to reflect improvement in water-use efficiency. Continuous improvements in technology make appliances more water efficient and hence, contribute to reductions in water use. Detailed data on domestic consumptive water uses do not exist from statistics but a simple balancing equation is used in WaterGAP since the year 2000 to simulate consumptive water uses as the difference between withdrawal water use and wastewater volume (i.e., return flow) as the

latter information is available from statistics. The calculation of consumptive water use before the year 2000 is based on the application of consumptive water use coefficients (Shiklomanov, 2000) that accounts for the proportion of the withdrawal water use that is consumed. In order to allow for a spatially explicit analysis country values of domestic water uses are allocated to grid cells ($0.5° \times 0.5°$) within the country based on the geo-referenced historical population density maps from HYDE version 3.1 (Goldewijk et al., 2010). Additionally, population numbers beyond 2005 as well as information on the ratio of rural to

urban population of each grid cell come from UNEP- The Environmental Data Explorer.

### 3.2.3    Manufacturing

The manufacturing sector is rather diverse in terms of water use and varies between countries and sub-sectors, for example highly water-intensive production processes in the chemical industry compared to the less water-using processes in the glass industry. In WaterGAP, the manufacturing water use model simulates the annual withdrawal water use and consumptive water

use of water that is used for production and cooling processes, whereas the water used for power generation is modelled separately. A manufacturing structural water intensity that describes the ratio of water abstracted over the manufacturing gross value added (GVA) is derived per country for the base year 2005 (in $\mathrm{m^3\,USD\,(constant\ for\ the\ year\ 2000)^{-1}}$) based on national statistics (Flörke et al., 2013). GVA is found to be positively correlated with the sector's withdrawal water uses





(Dziegielewski et al., 2002) and is used as the driving force to reflect the time variant system. In addition, technological improvements are considered through a technological change factor.

The consumptive water use for this sector is obtained by using the same approach as described for the domestic sector, i.e. the calculation of the difference between the withdrawal water use and the return flows (starting in the year 2000) and the application of a consumption factor before the year 2000. Contrary to the domestic sector, return flows from the manufacturing sector are further subdivided into cooling water and wastewater. For countries where no data are available, the fraction of consumptive water use is derived from neighbouring or economically comparable countries. Less information is available on the location of manufacturing industries, therefore country-level manufacturing water use is downscaled to grid cells proportional to its urban population (Flörke et al., 2013).

### 3.2.4 Thermal power

Water is abstracted and consumed for the production of thermal electricity, particularly for cooling purposes where water is used to condense steam from the turbine exhaust. The volume of cooling withdrawal water use and consumptive water use is modelled on a grid-cell level based on input data on the location, type and size of power stations from the World Electric Power Plants Data Set (UDI - World Electric Power Plants Database). Here, the annual cooling water requirements in each grid cell are calculated by multiplying the annual thermal electricity production with the respective water-use intensity of each power station (Flörke et al., 2013). Key driver is the annual thermal electricity production $\mathrm{MWh\,yr^{-1}}$ on a country basis which is downscaled to the level of thermal power plants according to their capacities. Time series on thermal electricity production per country until 2010 are available online from the Energy Information Administration (EIA - International Energy Statistics). Cooling water intensities in terms of withdrawal water use and consumptive water use vary between plant types and cooling systems. Therefore, the model distinguishes between four plant types (biomass and waste, nuclear, natural gas and oil, coal and petroleum) and three cooling systems (tower cooling, once-through cooling, ponds) (Flörke et al., 2012). The approach is complemented by considering technological change leading to reduced intensities.

In general, water abstractions of once-through flow systems are considerably higher compared to the withdrawal intensities of pond cooling or tower cooling systems. In contrast, consumptive water use of tower cooling systems is much higher than water consumed by once-through cooling systems. In ordering plant-type specific water intensities, i.e. water abstraction per unit electricity production, it becomes obvious that intensities are highest for nuclear power plants, followed by fossil, biomass, and waste-fuelled steam plants, while natural gas and oil combined-cycle plants have the lowest intensities, respectively. The model has been validated for the year 2005 by comparing modelled values with published thermoelectric withdrawal water uses (Flörke et al., 2013).

### 3.3 GWSWUSE

The linking model Groundwater-Surface Water Use (GWSWUSE) computes the fractions of all five sectoral water abstractions, or withdrawal water use, WU and consumptive water use CU in each grid cell that stem from either groundwater or surface water bodies (lakes, reservoirs and river). Time series for WU and CU from the sectoral water use models are an input to





GWSWUSE except for WU for irrigation. The latter is computed within GWSWUSE as water use efficiencies CU/WU for irrigation are assumed to vary between surface water and groundwater. Country-specific efficiency values are used for surface water irrigation, while in case of groundwater irrigation, water use efficiency is set to a relatively high value of 0.7 worldwide

(Döll et al., 2014). In GWSWUSE, CU due to irrigation is decreased to 70% of optimal CU in groundwater depletion areas; these areas were defined as grid cells with a groundwater depletion rate for 1980-2009 of more than 5 mm/yr and a ratio of WU for irrigation over WU for all sectors of more than 5% as computed for optimal irrigation in Döll et al. (2014).

Sectoral groundwater fractions were derived individually for each grid cell in case of irrigation (Siebert et al., 2010) and for each country in case of the other four water use sectors (Döll et al., 2012). They are assumed to be temporally constant. Water

for livestock and the cooling of thermal power plants is assumed to be extracted exclusively from surface water bodies.

Finally, GWSWUSE computes monthly time series of net abstraction from surface water $NA_s$ and from groundwater $NA_g$ which are used as input to WGHM. Net abstraction is the difference between total water abstraction from one of the two sources and the return flow to the respective source according to Eqs. 1, 3 and 4 in Döll et al. (2012). In all sectors except irrigation, return flows are only directed to surface water bodies. The fraction of return flow to groundwater in case of irrigation water

use is estimated as a function of degree of artificial drainage in the grid cell (section 2.1.3 in Döll et al. (2014)). Positive net abstraction values refer to the situation where storage is reduced due to human water use, negative values indicate an increase in storage. In case of groundwater, the latter only occurs if there is irrigation with surface water in the grid cell. The approach of direct net abstractions implicityl assumes instantaneous return flows. The sum of $NA_g$ and $NA_s$ is equivalent to consumptive water use. $NA_s$ as computed by GWSWUSE are potential net abstractions that may be adjusted depending on the availability

of surface water. In contrast, $NA_g$ is not constrained and groundwater storage can decrease without limit (Sect. 4.8).

## 4   WaterGAP Global Hydrology Model WGHM

The WaterGAP Global Hydrology Model (WGHM) simulates daily water flows and water storage in ten compartments (Fig. 2). The vertical water balance (dashed box in Fig. 2) encompasses the canopy (Sect. 4.2), snow (Sect. 4.3) and soil (Sect. 4.4) components. Water storage in glaciers is not simulated by WaterGAP2.2d. The lateral water balance includes groundwater

(Sect. 4.5), lakes, man-made reservoirs, wetlands (Sect. 4.6), and rivers (Sect. 4.7). Different to the vertical water balances, where the water balance is calculated based on water height units (mm), the lateral water balance is calculated in volumetric units ($m^3$). Water height units are converted to volumetric units by considering the land area (for flows) or continental area (for storages) of the grid cell, respectively. Local surface water bodies are defined to be recharged only by runoff generated in the cell itself, while global ones additionally receive streamflow from upstream cells (Fig. 2). Upstream-downstream relations

among the grid cells are defined by the drainage direction map DDM30 (Döll and Lehner, 2002). Each cell can drain only into one of the eight neighboring cells as streamflow. There is no groundwater flow between grid cells.

The amount of water reaching the soil is regulated by the canopy and snow water balance. Total runoff from the land fraction of the cell $R_l$ is calculated from the soil water balance. $R_l$ is then partitioned into fast surface and subsurface runoff $R_s$ and diffuse groundwater recharge $R_g$. Lateral routing of water through the storage compartments is based on the so-called





*fractional routing* scheme (Döll et al., 2014) and differs between (semi)arid and humid grid cells (red and green arrows in Fig. 2). The definition of (semi)arid and humid cells is given in Appendix B. To avoid that the whole runoff generated in the grid cell is added to local lake or wetland storage, only the fraction $f_{swb}$ times $R_s$ flows into surface water bodies and the remainder discharges into the river. The factor $f_{swb}$ is calculated as the relative area of wetlands and local lakes in a grid cell multiplied by 20 (representing the drainage area of surface water bodies), with its maximum value limited to the cell fraction of continental

area. In humid cells, groundwater discharge $Q_g$ is partitioned using $f_{swb}$ into discharge to surface water bodies and discharge to the river segment. In (semi)arid cells, surface water bodies (excluding rivers) are assumed to recharge the groundwater to mimic point recharge. To avoid a short circuit between groundwater and surface water bodies, the whole amount of $Q_g$ flows into the river. Loosing conditions, where river water recharges the groundwater are not modelled in WGHM.

In WaterGAP, human water use is assumed to affect only the water storages in the lateral water balance. Increases in soil

water storage in irrigated areas are not taken into account as the WaterGAP approach of direct net abstractions implicitly assumes instantaneous return flows. $NA_s$ is abstracted from the different surface water bodies except wetlands with the priorities shown as numbers in Fig. 2.

Outflow from the final water storage compartment in each cell, the river compartment, is streamflow ($Q_{r,out}$), which becomes inflow into the next downstream cell.

The ordinary differential equations describing the water balances of the ten storage compartment simulated in WGHM are solved sequentially for each daily timestep in the following order: canopy, snow, soil, groundwater, local lakes, local wetlands, global lakes, global reservoirs/regulated lakes, river (Fig. 2). An explicit Euler method is used to numerically solve all differential equations except those for global lakes and rivers, where an analytical solution is applied to compute storage change during one daily time step, which allows daily time steps instead of smaller time steps that would have been required

in case of explicit Euler method. As the water balances of global lakes, global reservoirs/regulated lakes and river of a grid cell are not independent from those of the upstream grid cells, the sequence of grid cell computations starts at the most upstream grid cells and continues downstream according to the drainage direction map DDM30 (Döll and Lehner, 2002).

### 4.1 General model variants of human water use and reservoirs

The standard model setup of WGHM in WaterGAP 2.2d simulates the effects of both human water use and man-made reservoirs

(including their commissioning years) on flows and storages and is referred to as "ant" simulation (anthropogenic). These stressors can be turned off in alternative model setups to simulate a world without these two types of human activities and to quantify the direct impact of human water use and reservoirs.

- "Nat" simulations compute naturalized flows and storages that would occur if there where neither human water use nor global man-made reservoirs/regulated lakes.

- "Use only" simulations include human water use but exclude global man-made reservoirs/regulated lakes.

- "Reservoirs only" simulations exclude human water use but include global man-made reservoirs/regulated lakes.

The following sections generally refer to "ant" simulations.





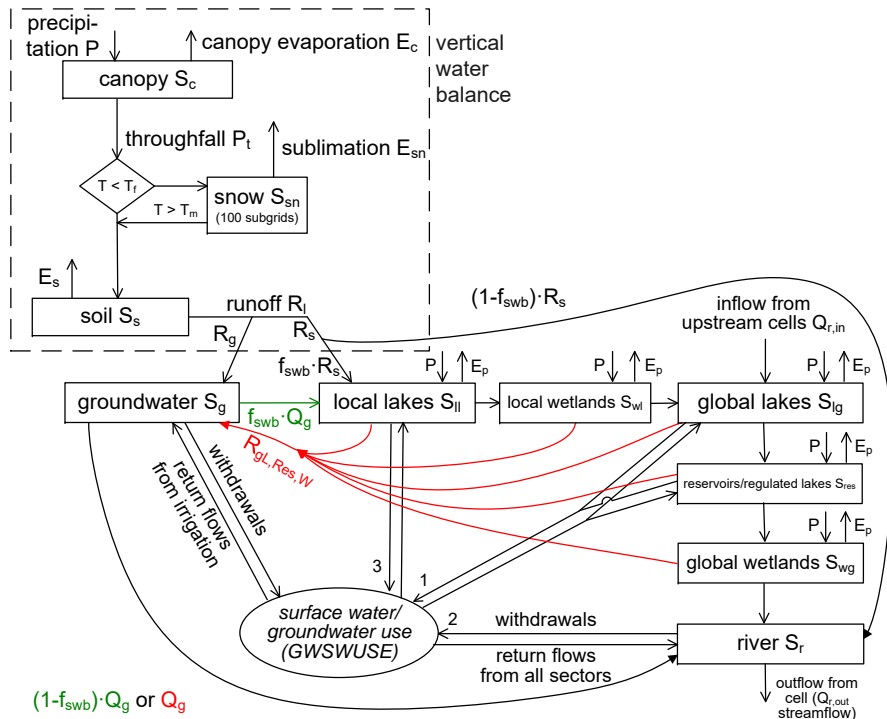

**Figure 2.** Schematic of WGHM in WaterGAP2.2d. Boxes represent water storage compartments, arrows represent water flows. Green (red) colour indicates processes that occur only in grid cells with humid ((semi)arid) climate. For details the reader is referred to the sections 4.2 to 4.8, in which the water balance equations of all ten water storage compartments are presented.

## 4.2 Canopy

Canopy refers to the leaves and branches of terrestrial vegetation that intercept precipitation. Modeling of the canopy processes
does not differentiate between rain and snow.

### 4.2.1 Water balance

The canopy storage $S_c$ (mm) is calculated as

$$\frac{dS_c}{dt} = P - P_t - E_c \tag{2}$$

where $P$ is precipitation $(\mathrm{mm\,d^{-1}})$, $P_t$ is throughfall, the fraction of $P$ that reaches the soil $(\mathrm{mm\,d^{-1}})$ and $E_c$ is evaporation
from the canopy $(\mathrm{mm\,d^{-1}})$.

### 4.2.2 Inflows

Daily precipitation $P$ is read in from the selected climate forcing (see Sect. 7.1).





### 4.2.3 Outflows

Throughfall $P_t$ is calculated as

$$P_t = \begin{cases} 0 & P < (S_{c,max} - S_c) \\ P - (S_{c,max} - S_c) & \text{otherwise} \end{cases} \tag{3}$$

where $S_{c,max}$ is maximum canopy storage calculated as

$$S_{c,max} = m_c * L \tag{4}$$

where $m_c$ is 0.3 mm (Deardorff, 1978) and $L\ (-)$ is one-side leaf area index. $L$ is a function of daily temperature and $P$ and limited to minimum or maximum values. Maximum $L$ values per land cover class (Table C1) are based on Schulze et al.

(1994) and Scurlock et al. (2001), whereas minimum $L$ values are calculated as

$$L_{min} = 0.1 f_{d,lc} + (1 - f_{d,lc}) c_{e,lc} L_{max} \tag{5}$$

where $f_{d,lc}$ is the fraction of deciduous plants and $c_{e,lc}$ is the reduction factor for evergreen plants per land cover type (Table C1).

The growing season starts when daily temperature is above $8°$ C for a land cover specific number of days (Table C1) and
cumulative precipitation from the day where growing season starts reaches at least 40 mm. In the beginning of the growing season, $L$ increases linearly for 30 days until it reaches $L_{max}$. For (semi)arid cells, at least 0.5 mm of daily $P$ is required to keep the growing season on-going. When growing season conditions are not fulfilled anymore, a senescence phase is initiated and $L$ linearly decreases to $L_{min}$ within the next 30 days ((Kaspar, 2003)).

Following Deardorff (1978), $E_c$ is calculated as

$$E_c = E_{pot} \left( \frac{S_c}{S_{c,max}} \right)^{\frac{2}{3}} \tag{6}$$

where $E_{pot}$ is the potential evapotranspiration ($\mathrm{mm\,d^{-1}}$) calculated with the Priestley-Taylor equation according to Shuttleworth (1993) as

$$E_{pot} = \alpha \left( \frac{s\,R}{s+g} \right) \tag{7}$$

where, following Shuttleworth (1993), $\alpha$ is set to 1.26 in humid and to 1.74 in (semi)arid cells. $R$ is net radiation ($\mathrm{mm\,d^{-1}}$)
that depends on land cover (Table C2) (for details in calculation of net radiation, the reader is referred to Müller Schmied et al. (2016b)) and $s$ is the slope of the saturation vapour pressure-temperature relationship ($\mathrm{k\,Pa\,°C^{-1}}$) defined as

$$s = \frac{4098(0.6108\,e^{\frac{17.27T}{T+237.3}})}{(T+237.3)^2} \tag{8}$$





where T (°C) is the daily air temperature and $g$ is the psychrometric constant $(\mathrm{k\,Pa\,°C^{-1}})$. The latter is defined as

$$g = \frac{0.0016286p}{l} \tag{9}$$

where $p$ is atmospheric pressure of the standard atmosphere (101.3 kPA) and $l$ is latent heat $\mathrm{MJkg^{-1}}$. Latent heat is calculated as

$$l = \begin{cases} 2.501 - 0.002361T & \text{if } T > 0 \\ 2.501 + 0.334 & \text{otherwise} \end{cases} \tag{10}$$

.

### 4.3 Snow

To simulate snow dynamics, each $0.5° \times 0.5°$ grid cell is spatially disaggregated into 100 non-localized subcells that are assigned different land surface elevations according to GTOPO30 (U.S. Geological Survey, 1996). Daily temperature at each subcell is calculated from daily temperature at the $0.5° \times 0.5°$ cell by applying an adiabatic lapse rate of 0.6°C per 100 m (Schulze and Döll, 2004). The daily snow water balance is computed for each of the subcells such that within a $0.5° \times 0.5°$ cell there may be subcells with and without snow cover or snowfall. For model output, subcell values are aggregated to $0.5° \times$
$0.5°$ cell values.

#### 4.3.1 Water balance

Snow storage accumulates below snow freeze temperature and decreases by snow melt and sublimation. Snow storage $S_{sn}$ (mm) is calculated as

$$\frac{dS_{sn}}{dt} = P_{sn} - M - E_{sn} \tag{11}$$

where $P_{sn}$ is the part of $P_t$ that falls as snow $(\mathrm{mm\,d^{-1}})$, $M$ is snowmelt $(\mathrm{mm\,d^{-1}})$ and $E_{sn}$ is sublimation $(\mathrm{mm\,d^{-1}})$.

#### 4.3.2 Inflows

Snowfall $P_{sn}$ $(\mathrm{mm\,d^{-1}})$ is calculated as

$$P_{sn} = \begin{cases} P_t & T < T_f \\ 0 & \text{otherwise} \end{cases} \tag{12}$$

where $T$ is daily air temperature (°C) and $T_f$ snow freeze temperature, set to 0 °C. In order to prevent excessive snow accu-
mulation, when snow storage $S_{sn}$ reaches 1000 mm in a subcell, the temperature in this subcell is increased to the temperature in the highest subcell with a temperature above $T_f$ (Schulze and Döll, 2004).





### 4.3.3 Outflows

Snow melt $M$ is calculated with a land-cover specific degree-day factor $D_F$ $(\mathrm{mm\,d^{-1}\,°C})$ (Table C2) when the temperature $T$ in a subgrid surpasses melting temperature $T_m$ (°C), set to 0 °C, as

$$
M = \begin{cases} D_F(T - T_m) & T > T_m, \ S_{sn} > 0 \\ 0 & \text{otherwise} \end{cases} \tag{13}
$$

Sublimation $E_{sn}$ is calculated as the fraction of $E_{pot}$ that remains available after $E_c$. For calculating $E_{pot}$ according to Eq. (7), land-cover specific albedo values are used if $S_{sn}$ surpasses 3 mm in the $0.5° \times 0.5°$ cell (Table C2).

$$
E_{sn} = \begin{cases} E_{pot} - E_c & E_{pot} - E_c > E_{sn} \\ S_{sn} & \text{otherwise} \end{cases} \tag{14}
$$

### 4.4 Soil

WaterGAP represents soil as a a one-layer soil water storage compartment characterized by a land-cover and soil-specific maximum storage capacity as well as soil texture. The simulated water storage represents soil moisture in the effective root zone.

#### 4.4.1 Water balance

The change of soil water storage (mm) over time (d) is calculated as

$$
\frac{dS_s}{dt} = P_{eff} - R_l - E_s \tag{15}
$$

where $P_{eff}$ is effective precipitation $(\mathrm{mm\,d^{-1}})$, $R_l$ is runoff from land $(\mathrm{mm\,d^{-1}})$ and $E_s$ is actual evapotranspiration from the soil $(\mathrm{mm\,d^{-1}})$. Once the water balance is computed, $R_l$ is partitioned into 1) fast surface and subsurface runoff $R_s$, representing direct surface runoff and interflow, and 2) groundwater recharge $R_g$ (Fig. 2) according to a heuristic scheme (Döll and Fiedler, 2008).

#### 4.4.2 Inflows

$P_{eff}$ is computed as

$$
P_{eff} = P_t - P_{sn} + M \tag{16}
$$

where $P_t$ is throughfall $(\mathrm{mm\,d^{-1}}$, see Eq. (3)), $P_{sn}$ is snowfall $(\mathrm{mm\,d^{-1}}$, see Eq. (12)) and $M$ is snowmelt $(\mathrm{mm\,d^{-1}}$, see Eq. (13)).



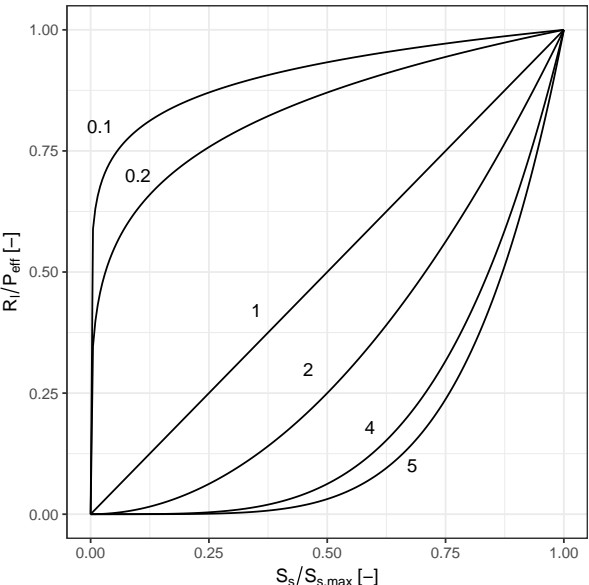

**Figure 3.** Relation between runoff from land $R_l$ as a fraction of effective precipitation $P_{eff}$ and soil saturation $S_s/S_{s,max}$ for different values of the runoff coefficient $\gamma$ in WaterGAP.

### 4.4.3 Outflows

$E_s$ is calculated as

$$E_s = min\left( (E_{pot} - E_c), (E_{pot,max} - E_c)\frac{S_s}{S_{s,max}} \right) \tag{17}$$

where $E_{pot}$ is potential evapotranspiration ($\mathrm{mm\,d^{-1}}$), $E_c$ is canopy evaporation ($\mathrm{mm\,d^{-1}}$, Eq. (6)) and $S_{s,max}$ is the maximum soil water content ($\mathrm{mm}$) derived as product of total available water capacity in the upper meter of the soil (Batjes, 2012) and land-cover-specific rooting depth (Table C2) (Müller Schmied, 2017). $E_{pot,max}$ is set to 15 $\mathrm{mm\,d^{-1}}$ globally. Following Bergström (1995), runoff from land $R_l$ is calculated as

$$R_l = P_{eff}\left( \frac{S_s}{S_{s,max}} \right)^{\gamma} \tag{18}$$

where $\gamma$ is the runoff coefficient ($-$). This parameter, which varies between 0.1 and 5.0, is used for calibration (Sect. 4.9). Together with soil saturation, it determines the fraction of $P_{eff}$ that becomes $R_l$ (Fig. 3). If the sum of $P_{eff}$ and $S_s$ of the previous day exceed $S_{s,max}$, the exceeding fraction of $P_{eff}$ is added to $R_l$. In urban areas (defined from MODIS data, Sect. C), 50% of $P_{eff}$ is directly turned into $R_l$.

$R_l$ is partitioned into fast surface and subsurface runoff $R_s$ and diffuse groundwater recharge $R_g$ calculated as

$$R_g = min(R_{g_{max}}, f_g R_l) \tag{19}$$





where $R_{g_{max}}$ is soil-texture specific maximum groundwater recharge with values of 7/4.5/2.5 $\mathrm{mm\,d^{-1}}$ for sandy/loamy/clayey

soils and $f_g$ the groundwater recharge factor ranging between 0 and 1. $f_g$ is determined based on relief, soil texture, aquifer type and the existence of permafrost or glaciers (Döll and Fiedler, 2008). If a grid cell is defined as (semi)arid and has coarse (sandy) soil, groundwater recharge will only occur if precipitation exceeds a critical value of 12.5 $\mathrm{mm\,d^{-1}}$, otherwise the water remains in the soil. The fraction of $R_l$ that does not recharge the groundwater becomes $R_s$, which recharges surface water bodies and the river compartment.

**4.5 Groundwater**

As there is no knowledge about the depth below the land surface where groundwater no longer occurs due to lack of pore space, groundwater storage can only be computed in relative terms but is assumed to be unlimited. The groundwater storage $S_g$ is always positive unless net abstractions from groundwater $NA_g$ are high and groundwater depletion occurs. Groundwater discharge is assumed to be proportional to (positive) $S_g$ and to stop in case of negative $S_g$.

**4.5.1 Water balance**

The temporal development of groundwater storage $S_g$ ($\mathrm{m^3}$) is calculated as

$$\frac{dS_g}{dt} = R_g + R_{g_{l,res,w}} - Q_g - NA_g \tag{20}$$

where $R_g$ is diffuse groundwater recharge from soil ($\mathrm{m^3\,d^{-1}}$, Eq. (19)), $R_{g_{l,res,w}}$ point groundwater recharge from surface water bodies (lakes, reservoirs and wetlands) in (semi)arid areas ($\mathrm{m^3\,d^{-1}}$, Eq. (26)), $Q_g$ groundwater discharge ($\mathrm{m^3\,d^{-1}}$) and

$NA_g$ net abstraction from groundwater ($\mathrm{m^3\,d^{-1}}$).

**4.5.2 Inflows**

$R_g$ is the main inflow in most grid cells, except in (semi)arid grid cells with significant surface water bodies where $R_{g_{l,res,w}}$ may be dominant. $R_{g_{l,res,w}}$ varies temporally with the area of the surface water body, which depends on the respective water storage (Sec. 4.6). In many cells with significant irrigation with surface water, $NA_g$ is negative, and irrigation causes a net

inflow into the groundwater due to high return flows (Sect. 3.3).

**4.5.3 Outflows**

$Q_g$ quantifies the discharge from groundwater storage to surface water storage, with

$$Q_g = k_g S_g \tag{21}$$

where $k_g = 0.01\ \mathrm{d^{-1}}$ is the globally constant groundwater discharge coefficient (Döll et al., 2014). The second outflow com-

ponent $NA_g$ is described in Sect. 3.3.





## 4.6 Lakes, man-made reservoirs and wetlands

Where lakes, man-made reservoirs and wetlands (LResW) of significant size exist, their water balances strongly affect the overall water balance of the grid cell due to their high evaporation and water retention capacity (Döll et al., 2003). WGHM uses the Global Lakes and Wetland Database (GLWD) (Lehner and Döll, 2004) and a preliminary but updated version of the

Global Reservoir and Dam (GRanD) database (Döll et al., 2009; Lehner et al., 2011) to define location, area and other attributes of LResW. It is assumed that surface areas given in the databases represent the maximum extent. Sect. E describes how the information from these databases is integrated into WGHM. Two categories of LResW are defined for WGHM, so-called "local" water bodies that receive inflow only from the runoff generated within the grid cell and so-called "global" water bodies that additionally receive the streamflow from the upstream grid cells (Fig. 2). Six different LResW types are distinguished in

WaterGAP.

- *Local wetlands* ($wl$) and *global wetlands* ($wg$) cover a maximum area of 3.743 million $km^2$ and 3.752 million $km^2$, respectively, an area that is at least at its maximum three times larger than the combined maximum area of lakes and reservoirs (Appendix E). However, 0.3 million $km^2$ of floodplains along large rivers are included as global wetlands, and their dynamics are not simulated suitably by WGHM. They are assumed to receive the total streamflow as inflow

while in reality only the part of the streamflow that does not fit in the river channel flows into the floodplain (Döll et al., 2020). All local (global) wetlands within a $0.5° \times 0.5°$ grid cell are simulated as one local (global) wetland that covers a specified fraction of the cell.

- *Local lakes* ($ll$) include about 250,000 small lakes and more than 5000 man-made reservoirs and are defined to have a surface area of less than 100 $km^2$ or a maximum storage capacity of less than 0.5 $km^3$. Like wetlands, all local lakes

in a grid cell are aggregated and simulated as one storage compartment taking up a fraction of the grid cell area. Small reservoirs are simulated like lakes as 1) the required lumping of all local reservoirs within a grid cell into one local reservoir per cell necessarily leads to a "blurring" of the specific reservoir characteristics, and 2) small reservoirs are likely not on the main river simulated in the grid cell but on a tributary. Therefore, a reservoir algorithm is not expected to simulate water storage and flows better than the lake algorithm.

- 1386 *global lakes* ($lg$), i.e. lakes with an area of more than 100 $km^2$, are simulated in WaterGAP. Since a global lake may spread over more than one grid cell, the water balance of the whole lake is computed at the outflow cell (Döll et al., 2009) (for consequences, see Sect. 5.2). Only the maximum area of natural lakes is known, not the maximum water storage capacity.

- *Global man-made reservoirs* ($res$) have a maximum storage capacity of at least 0.5 $km^3$ and *global regulated lakes*

(lakes where outflow is controlled by a dam or weir) have a maximum storage capacity of at least 0.5 $km^3$ or an area of more than 100 $km^2$. Both are simulated by the same water balance equation. There can be only one global reservoir/regulated lake compartment per grid cell. Outflow from reservoirs/regulated lakes is simulated by a modified





version of the Hanasaki et al. (2006) algorithm, distinguishing reservoirs/regulated lakes with the main purpose of irrigation from others (Döll et al., 2009). Like in the case of global lakes, water balance of global reservoirs/regulated lakes is computed at the outflow cell (for consequences, see Sect. 5.2). Different from lakes, information on maximum water storage capacity is available from the GRanD database, in addition to the main use and the commissioning year. In WGHM, reservoirs start filling at the beginning of the commissioning year, and regulated lakes then turn from global lakes into global regulated lakes (Sect. E). 1082 global reservoirs and 85 regulated lakes are taken into account, but as those that have the same outflow cell are aggregated to one water storage compartment by adding maximum storages and areas, only 1109 global reservoirs/regulated lakes compartments are simulated in WGHM (Sect. E). Under naturalized conditions (Sect. 4.1), there are no global man-made reservoirs and regulated lakes are simulated as global lakes; however, local reservoirs remain in the model.

In each grid cell, there can be a maximum of one local wetland storage compartment, one global wetland compartment, one local lake compartment, one global lake compartment and one global reservoir/regulated lake compartment. The lateral water flow within the cell follows the sequence shown in Fig. 2. For example, if there is a local lake compartment in a grid cell, it is this compartment that receives, under humid climate, a fraction of the outflow from the groundwater compartment and of the fast surface and subsurface outflow, and the outflow from the local lake becomes inflow to the local wetland if existing (Fig. 2). If there is no local wetland but a global lake, the outflow from the local lake becomes part of the inflow of the global lake. In case of having a global lake and a global reservoir/regulated lake in one cell, water is routed first through the global lake.

### 4.6.1 Water balance

The water balance for the five types of LResW compartments is calculated as

$$\frac{dS_{l,res,w}}{dt} = Q_{in} + A(P - E_p) - R_{g_{l,res,w}} - NA_{l,res} - Q_{out} \tag{22}$$

where $S_{l,res,w}$ is volume of water stored in the water body (m$^3$), $Q_{in}$ is inflow into water body from upstream (m$^3$ d$^{-1}$), $A$ is global (or local) water body surface area (m$^2$) in the grid cell at time $t$, $P$ is precipitation (m d$^{-1}$), $E_{pot}$ is potential evapotranspiration (m d$^{-1}$, Eq. (7)), $R_{g_{l,res,w}}$ is groundwater recharge from the water body (only in arid/semi-arid regions) (m$^3$ d$^{-1}$, Eq. (26)), $NA_{l,res}$ is the net abstraction from the lakes and reservoirs (m$^3$ d$^{-1}$) (Fig. 2 and Sect. 4.8), $Q_{out}$ is outflow from the water body to other surface water bodies including river storage (m$^3$) (Fig. 2).

The temporally varying surface area $A$ of the water body is computed in each daily time step using the following equation:

$$A = r * A_{max} \tag{23}$$

where $r$ is reduction factor (-), $A_{max}$ is maximum extent of the water body (m$^2$) from GRanD or GLWD databases. In case of local and global lakes

$$r = 1 - \left( \frac{|S_l - S_{l,max}|}{2S_{l,max}} \right)^p, \qquad 0 \leq r \leq 1 \tag{24}$$





where $S_l$ is the volume of the water (m$^3$) stored in the lake at time $t$ (d), $S_{l,max}$ is the maximum storage of the lake (m$^3$). $S_{l,max}$ is computed based on $A_{max}$ and a maximum storage depth of 5m, $p$ is the reduction exponent (-), set to 3.32. According to the above equation, the area is reduced by 1% if $S_l$ = 50% of $S_{l,max}$, by 10% if $S_l$ = 0 and by 100% if $S_l$ = -$S_{l,max}$ (Hunger and Döll, 2008). In case of global reservoirs/regulated lakes and local and global wetlands

$$r = 1 - \left( \frac{|S_{res,w} - S_{res,w,max}|}{S_{res,w,max}} \right)^p, \qquad 0 \leq r \leq 1 \tag{25}$$

where $S_{res,w}$ is the volume of the water (m$^3$) stored in the reservoir/regulated lake or wetland, $p$ is 2.814 and 3.32 for reservoirs/regulated lakes and wetlands, respectively. In case of wetlands, $S_{res,w,max}$ (m$^3$) is computed based on $A_{max}$ and a maximum storage depth of 2 m. Wetland area is reduced by 10% if $S_w$ = 50% of $S_{res,w,max}$ and by 70% if $S_w$ is only 10% of $S_{res,w,max}$. In case of reservoirs/regulated lakes, storage capacity $S_{res,w,max}$ is taken from the database. Reservoir area is reduced by 15% if $S_{res}$ is 50% of $S_{res,w,max}$ and by 75% if $S_{res}$ is only 10% of $S_{res,w,max}$. For regulated lakes without available maximum storage capacity, $S_{res,w,max}$ is computed as in case of global lakes.

While storage in reservoirs/regulated lakes and wetlands cannot drop below zero due to high outflows, high evaporation or $NA_s$, storage in lakes can become negative. This represents the situation where there is no more outflow from the lake to a downstream water body ($Q_{out}$ = 0). There, like groundwater storage, storage of local and global lakes is a relative and not an absolute water storage. Reservoir/regulated lakes storage is not allowed to fall below 10% of storage capacity.

With changing $A$ of the surface water compartments local wetland, global wetlands and local lakes, the land area fraction is adjusted accordingly. However, in case of global lakes and reservoirs/regulated lakes, which may cover more than one 0.5° × 0.5° cell, such an adjustment is not made as it is not known, in which grid cells the area reduction occurs. Therefore, land area fraction is not adjusted with changing $r$ and precipitation is assumed to fall on a surface water body with an area of $A_{max}$ instead of $A$.

### 4.6.2 Inflows

Calculation of $Q_{in}$ differs between local and global water bodies. In case of local lakes and local wetlands, they are recharged only by local runoff generated within the same grid cell. A fraction $f_{swb}$ of the fast surface and subsurface runoff generated within the grid cell $R_s$ (m$^3$ d$^{-1}$) and, only in case of humid grid cells, a fraction $f_{swb}$ of the base flow from groundwater $Q_g$ (m$^3$ d$^{-1}$) become inflow to local water bodies (Fig. 2, Sect. 4.4.3, 4.5.2). In case where one grid cell contains both local lake and wetland, then the outflow of the local lake will be the inflow to the local wetland according to Fig. 2. Global lakes, global wetlands, and global reservoirs/regulated lakes receive, in addition to local runoff, inflow from streamflow of the upstream grid cells as river inflow (Fig. 2). In many cells with significant groundwater abstraction, $NA_s$ is negative, and return flow leads to a net inflow into surface water bodies (Sect. 3.3).

### 4.6.3 Outflows

LResW lose water by evaporation $E_{pot}$, which is assumed to be equal to the potential evapotranspiration computed using the Priestley-Taylor equation with an albedo of 0.08 according Eq. (7). In semi-arid and arid grid cells (Sect. B), LResW are





assumed to recharge the groundwater with a focused groundwater recharge, $R_{g_{l,res,w}}$ with

$$R_{g_{l,res,w}} = K_{gw_{l,res,w}} * r * A \tag{26}$$

where $K_{gw_{l,res,w}}$ is the groundwater recharge constant below LResW (= 0.01 m d$^{-1}$). This process is applied only in the arid and semi-arid grid cells, as in humid areas groundwater mostly recharges the surface water bodies as explained in the Sect. 4.6.2 (Döll et al., 2014).

It is assumed that water can be abstracted from lakes and reservoirs but not from wetlands. An amount of $NA_{l,res}$ (m$^3$ d$^{-1}$) is the net abstractions from lakes and reservoirs, depends on the total unsatisfied water use $Rem_{use}$ and the water storage in the surface water compartment. In case of a global lake and a reservoir within the same cell, $NA_{l,res}$ is distributed equally. In a reservoir, abstraction is only allowed until water storage reaches 10% of storage capacity (after fulfilling $E$ and $R_{g_{l,res}}$). Outflow from LResW to downstream water bodies including river storage (Fig. 2) is calculated as a function of LResW

water storage. The principal effect of a lake or wetland is to reduce the variability of streamflow, which can be simulated by computing outflow $Q_{out}$ as

$$Q_{out} = k * S_{ll,wl} * \left(\frac{S_{ll,wl}}{S_{ll,wl,max}}\right)^a \tag{27}$$

where $S_{ll,wl}$ is the local lake or local wetland storage (m$^3$) and $k$ is the surface water outflow coefficient (= 0.01 d$^{-1}$). The exponent $a$ is set to 1.5 in case of local lakes, based on the theoretical value of outflow over a rectangular weir, while the

exponent of 2.5 used for local wetlands leads to a slower outflow (Döll et al., 2003). The outflow of global lakes and global wetlands is computed as

$$Q_{out} = k * S_{lg,wg} \tag{28}$$

Different from the commissioning year of a reservoir, which is the year the dam was finalized (Appendix E), the operational year of each reservoir is the 12-month period for which reservoir management is defined. It starts with the first month with a

naturalized mean monthly streamflow that is lower than the annual mean. To compute daily outflow, e.g., release, from global reservoirs/regulated lakes, the total annual outflow during the reservoir-specific operational year is determined first as a function of reservoir storage at the beginning of the operational year. Total annual outflow during the operational year is assumed to be equal to the product of mean annual outflow and a reservoir release factor $k_{rele}$ that is computed each year on the first day of the operational year as

$$k_{rele} = \frac{S_{res}}{S_{res,max} * 0.85} \tag{29}$$

where $S_{res}$ is the reservoir/regulated lake storage (m$^3$) and $S_{res,max}$ is the storage capacity (m$^3$). Thus, total release in an operational year with low reservoir storage at the beginning of the operational year will be smaller than in a year with high reservoir storage.

During the first filling phase of a reservoir after dam construction, $k_{rele}$ = 0.1 until $S_{res}$ exceeds 10% of $S_{res,max}$. If the

storage capacity to mean total annual outflow ratio is larger than 0.5, then the outflow from the reservoir is independent of the





actual inflow, and temporally constant in case of a non-irrigation reservoir. In case of an irrigation reservoir, outflow is driven by monthly $NA_s$ in the next five downstream cells or down to the next reservoir (Döll et al., 2009; Hanasaki et al., 2006). For reservoirs with a smaller ratio, the release additionally depends on daily inflow and is higher on days with high inflow (Hanasaki et al., 2006). If reservoir storage drops below 10% of $S_{res,max}$, release is reduced to 10% of the normal release to

satisfy a minimum environmental flow requirement for ecosystems. Daily outflow may also include overflow, which occurs if reservoir storage capacity is exceeded due to high inflow into the reservoir.

### 4.7   Rivers

The water balance of the river compartment is computed to quantify streamflow, one of the most important output variables of hydrological models.

#### 4.7.1   Water balance

The dynamic water balance of the river water storage in a cell is computed as

$$\frac{dS_r}{dt} = Q_{r,in} - Q_{r,out} - NA_{s,r} \tag{30}$$

where $S_r$ is the volume of water stored in the river (m³), $Q_{r,in}$ is inflow into the river compartment ($\mathrm{m^3\,d^{-1}}$), $Q_{r,out}$ is the streamflow ($\mathrm{m^3\,d^{-1}}$) and $NA_{s,r}$ is the net abstraction of surface water from the river ($\mathrm{m^3\,d^{-1}}$).

#### 4.7.2   Inflows

If there are no surface water bodies in a grid cell, $Q_{r,in}$ is the sum of $R_s$, $Q_g$ and streamflow from existing upstream cell(s). Otherwiese, part of $R_s$, and in the case of humid cells also part of $Q_g$, is routed through the surface water bodies (Fig. 2). The outflow from the surface water body preceding the river compartment then becomes part of $Q_{r,in}$. In addition, negative $NA_s$ values due to high return flows from irrigation with groundwater lead to a net increase in storage. Thus, if no surface water

bodies exist in the cell, negative NAs is added to $Q_{r,in}$ (Sect. 3.3 and Fig. 2).

#### 4.7.3   Outflows

$Q_{r,out}$ is defined as the streamflow that leaves the cell and is transferred to the downstream cell.

It is calculated as

$$Q_{r,out} = \frac{v}{l} * S_r \tag{31}$$

where $v$ ($\mathrm{m\,d^{-1}}$) is river flow velocity and $l$ is the river length (m). $l$ is calculated as the product of the cell's river segment length, derived from the HydroSHEDS drainage direction map (Lehner et al., 2008), and a meandering ratio specific to that cell (method described in Verzano et al. (2012)). $v$ is calculated according to the Manning-Strickler equation as

$$v = n^{-1} * R_h^{\frac{2}{3}} * s^{\frac{1}{2}} \tag{32}$$




where $n$ is river bed roughness $(-)$, $R_h$ is the hydraulic radius of the river channel (m) and $s$ is river bed slope $(\mathrm{m\,m^{-1}})$. Cal-
culation of $s$ is based on high resolution elevation data (SRTM30), the HydroSHEDS drainage direction map and an individual
meandering ratio. The pre-defined minimum $s$ is $0.0001\,\mathrm{m\,m^{-1}}$.

To compute the daily varying $R_h$, a trapezoidal river cross section with a slope of 0.5 is assumed such that it can be calculated
as a function of daily varying river depth $D_r$ and temporally constant bottom width $W_{r,bottom}$ (Verzano et al., 2012). Allen
et al. (1994) empirically derived equations relating river depth, river top width and streamflow for bankfull conditions. In
former model versions, these equation were also applied at each time step, even if streamflow was not bankfull, to determine
river width and depth required to compute $R_h$ and thus $v$. As usage of these functions for any streamflow below bankfull is not
backed by the data and method of Allen et al. (1994), WaterGAP2.2d implements a consistent method for determining daily
width and depth as a function of river water storage.

As bankfull conditions are assumed to occur at the initial time step, the initial volume of water stored in the river is computed
as

$$S_{r,max} = \frac{1}{2} * l * D_{r,bf} * (W_{r,bottom} + W_{r,bf}) \tag{33}$$

where $S_{r,max}$ is the maximum volume of water that can be stored in the river at bankfull depth $(\mathrm{m^3})$, $D_{r,bf}$ (m) and $W_{r,bf}$ (m)
are river depth and top width at bankfull conditions, respectively, and $W_{r,bottom}$ is river bottom width (m). River water depth
$D_r$ is simulated to change at each time step with actual $S_r$ as

$$D_r = -\frac{W_{r,bottom}}{4} + \sqrt{W_{r,bottom} * \frac{W_{r,bottom}}{16} + 0.5 * \frac{S_r}{l}} \tag{34}$$

Using the equation for a trapezoid with a slope of 0.5, $R_h$ is then calculated from $W_{r,bottom}$ and $D_r$. Bankfull flow is assumed
to correspond to the maximum annual daily flow with a return period of 1.5 years (Schneider et al., 2011) and is derived from
daily streamflow time series.

The roughness coefficient $n$ of each grid cell is calculated according to Verzano et al. (2012), who modeled $n$ as a function
of various spatial characteristics (e.g., urban or rural area, vegetation in river bed, obstructions) and a river sinuosity factor to
achieve an optimal fit to streamflow observations. Because of the implementation of a new algorithm to calculate $D_r$, we had to
adjust their gridded $n$-values to avoid excessively high river velocities (Schulze et al., 2005). By trial-and-error, we determined
optimal $n$-multipliers at the scale of thirteen large river basins that lead to a good fit to monthly streamflow time series at the
most downstream stations and basin-average total water storage anomalies from GRACE. We found that in nine out of thirteen
basins, multiplying $n$ by 3 resulted in the best fit between observed and modeled data. We therefore set the multiplier to 3
globally, except for the remaining four basins, where other values proved to be more adequate; this concerns the Lena basin,
where $n$ is multiplied by 2, the Amazon basin, where $n$ is multiplied by 10 and the Huang He and Yangtze basin, where $n$ are
kept at their original value (Fig. D3).

Net cell runoff $R_{nc}$ $(\mathrm{mm\,d^{-1}})$, the part of the cell precipitation that has neither been evapotranspirated or stored with a time
step, is calculated as

$$R_{nc} = \frac{(Q_{r,out} - Q_{r,in})}{A_{cont}} \times 10^9 \tag{35}$$





where $A_{cont}$ is continental area ($0.5° \times 0.5°$ grid cell area minus ocean area) of the grid cell (m$^2$). Renewable water resources are calculated as long-term mean annual $R_{nc}$ computed under naturalized conditions (Sect. 4.1). Renewable water resources can be negative if evapotranspiration in a grid cell is higher than precipitation due to evapotranspiration from global lakes, reservoirs or wetlands that receive water from upstream cells.

### 4.8 Abstraction of human water use in WaterGAP Global Hydrological Model

The global water use models (Sec. 3) together with GWSWUSE (Sec. 3.3) calculate potential $NA_g$ and $NA_s$, which are independent of actual water availability. Potential $NA_g$ is always satisfied in WGHM due to the assumed unlimited groundwater storage that can be depleted.

Satisfaction of potential $NA_s$ depends on the availability of water in surface water bodies including the river compartment, considering the abstraction priorities shown in Fig.2. If the surface water in a grid cell cannot satisfy potential $NA_s$ of the grid cell on a certain day, two processes are used to distribute the unsatisfied water use spatially and temporally and thus to potentially increase the amount of satisfied $NA_s$:

1. Unsatisfied water use of a cell is allocated to the neighboring cell with the largest river and lake storage ("second" cell), and water required in the cell is abstracted in this neighboring cell.

2. Unsatisfied water use is added to $NA_s$ of the next day until the end of the calendar year.

In addition, potential $NA_s$ of riparian cells of global lakes and reservoirs (where the water balance is calculated in the outflow cell), identified based on the lake/reservoir polygons, can be satisfied by global lake or reservoir storage. If demand $NA_s$ can still not be fulfilled, actual $NA_s$ becomes smaller than potential $NA_s$.

In case of irrigation by surface water, it is assumed the any decrease of $NA_s$ is due to a decrease of withdrawal water uses for irrigation. This also reduces return flow to groundwater. Therefore, in WaterGAP 2.2d, $NA_g$ is increased in each time step in the water demand cell in accordance with the unfulfilled potential $NA_s$ in the cell (after steps 1 and 2).

### 4.9 Calibration and regionalization

#### 4.9.1 Calibration approach

The main purpose of WaterGAP is to quantify water resources and water stress. Therefore, WGHM has been calibrated against observed streamflow from the very beginning of model development to avoid that average water resources are misrepresented (Döll et al., 2003; Kaspar, 2003). Calibration is required due to uncertain model parameters, input data (e.g., deviations of precipitation from meterological forcings to observation networks (Wang et al., 2018)) and model structure including the spatial resolution. The rationale behind the approach can be summed up by the phrase "If the model is not able to properly capture the average observed hydrological conditions, how well founded are future projections?" (see also the discussion in Krysanova et al. (2018)). In order to minimize the problem of equifinality, WGHM is calibrated in a very simple basin-specific manner to match long-term mean annual observed streamflow ($Q_{obs}$) at the outlet of 1319 drainage basins that cover ~54%





of the global drainage area (except Antarctica and Greenland) (Fig. 4). The runoff coefficient $\gamma$ (Eq. (18)) and up to two additional correction factors (the areal correction factor CFA and the station correction factor CFS), if needed, are adjusted

homogeneously for all grid cells within the drainage basin. Calibration starts in upstream basins and proceeds to downstream basins, the streamflow from the already calibrated upstream basin as inflow.

While the calibration approach in WaterGAP2.2d is generally the same as in previous model versions (Döll et al., 2003; Hunger and Döll, 2008; Müller Schmied et al., 2014), it was modified (Müller Schmied, 2017, Sect. A3) to allow for a $\pm$ 10% gauging station observation uncertainty (following Coxon et al., 2015; Pascolini-Campbell et al., 2020) instead of $\pm$ 1% in

previous model versions. The source of streamflow data and selection criteria for stations are the same as in Müller Schmied et al. (2014) (their Sect. B2) but the 30-year period was shifted (if available) from 1971-2000 to 1978-2009 to capture a more recent time period.

Calibration follows a four-step scheme with specific calibration status (CS):

1. CS1: adjust the basin-wide uniform parameter $\gamma$ (Eq. (18)) in the range of [0.1-5.0] to match $Q_{obs}$ within $\pm$ 1%.

2. CS2: adjust $\gamma$ as for CS1, but within 10% uncertainty range (90-110% of observations).

3. CS3: as CS2 but apply the areal correction factor CFA (adjusts runoff and, to conserve the mass balance, actual evapotranspiration as counterpart of each grid cell within the range of [0.5-1.5]) to match $Q_{obs}$ with 10% uncertainty.

4. CS4: as CS3 but apply the station correction factor CFS (multiplies streamflow in the cell where the gauging station is located by an unconstrained factor) to match $Q_{obs}$ with 10% uncertainty to avoid error propagation to the downstream

basin. Note that with CFS, actual evapotranspiration of this grid cell is not adapted accordingly to avoid unphysical values. Hence, mass is not conserved in case of CS4 for the grid cell where CFS is applied in the upstream basin. For global water balance assessment, the mass balance is kept by adjusting the actual evapotranspiration component.

For each basin, calibration steps 2-4 are only performed if the previous step was not successful.

### 4.9.2   Regionalization approach

The calibrated $\gamma$ values are regionalized to river basins without sufficient streamflow observations using a multiple linear regression approach that relates the natural logarithm of $\gamma$ to basin descriptors (mean annual temperature, mean available soil water capacity, fraction of local and global lakes and wetlands, mean basin land surface slope, fraction of permanent snow and ice, aquifer-related groundwater recharge factor). Just like the calibrated $\gamma$-values, the regionalized values are limited between 0.1 and 5.0; CFA and CFS are set to 1.0 in uncalibrated basins. A manual modification of the regionalized $\gamma$ value to 0.1

was done (from values of 3-5) for basins covering the North China Plain in northeastern China as groundwater depletion was overestimated by a factor of 4 in this region (Döll et al., 2014); a lower $\gamma$ allows higher runoff generation that translates into higher groundwater recharge and thus a weaker overestimation.

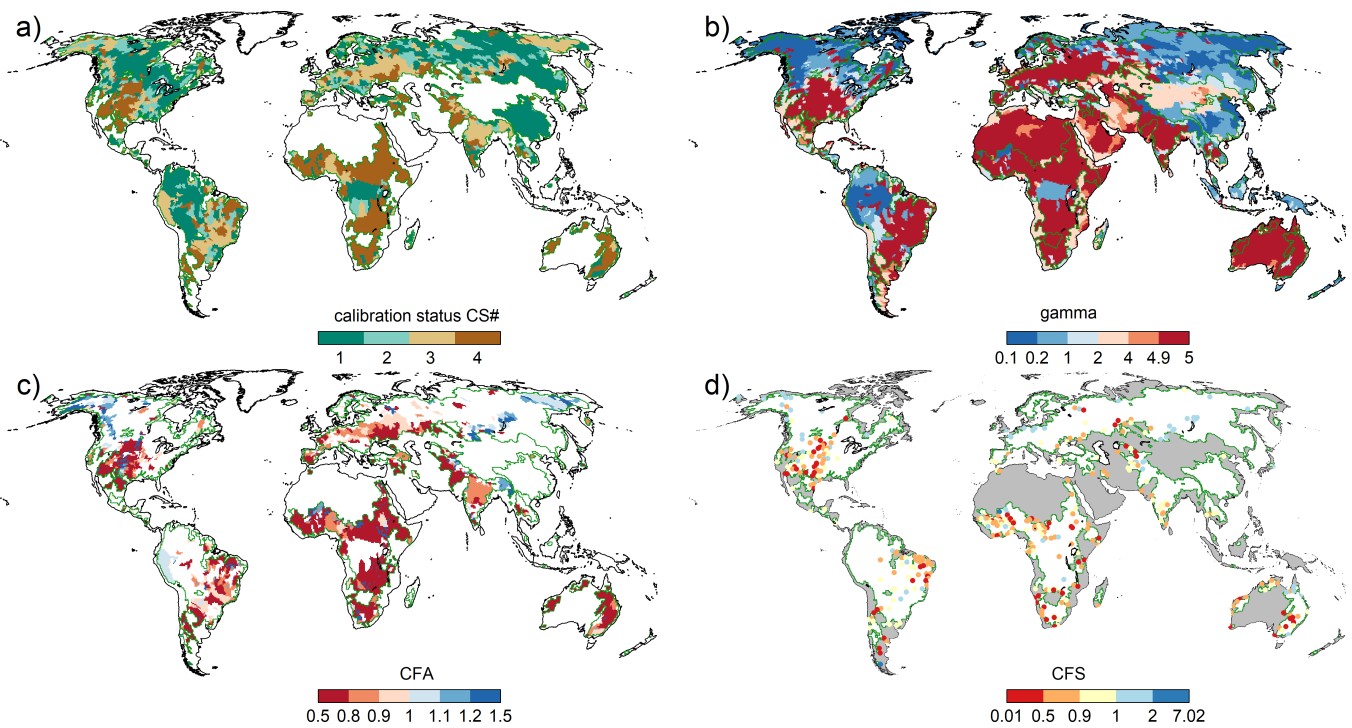

**Figure 4.** Results of the calibration of WaterGAP 2.2d to the standard climate forcing with a) the calibration status (see Sect. 4.9.1) of each calibration basin b) calibration parameter $\gamma$, c) areal correction factor CFA, and d) station correction factor CFS. Grey areas in d) indicate regions with regionalized calibration parameter $\gamma$ and for a-d) dark green outlines indicate the boundaries of the calibration basins.

### 4.9.3 Calibration and regionalization results

Calibration of WaterGAP 2.2d driven by the standard climate forcing (Sect. 7.1) results in 485 basins with calibration status

CS1, 185 basins with calibration status CS2, 277 basins with calibration status CS3 and 372 basins with calibration status CS4. This means that in 72% of the calibration basins, the usage of the station correction factor CFS is not required to match the simulated long-term annual streamflow to observations. The spatial distribution of the calibration parameters and status is shown in Fig. 4.

## 5 Standard model output

### 5.1 Data provided at PANGAEA repository

A set of standard model outputs is provided via the data publisher and repository PANGAEA hosted by Alfred Wegener Institute, Helmholtz Center for Polar and Marine Research (AWI), Center for Marine Environmental Sciences and University of Bremen (MARUM), under the Creative Commons Attribution Non Commercial 4.0 International license (CC-BY-NC-4.0).

**Table 1.** Standard WaterGAP output variables: 1) Water storages. Units are $\mathrm{kg\,m^{-2}}$ (mm e.w.h.). Temporal resolution is monthly.

| Storage type | PANGEA file | Symbol |
|---|---|---|
| Total water storage[1,2] | tws | $S_{tws}$ |
| Canopy water storage | canopystor | $S_c$ |
| Snow water storage | swe | $S_{sn}$ |
| Soil water storage | soilmoist | $S_s$ |
| Groundwater storage[2] | groundwstor | $S_g$ |
| Local lake storage[2] | loclakestor | $S_{ll}$ |
| Global lake storage[2] | glolakestor | $S_{lg}$ |
| Local wetland storage | locwetlandstor | $S_{wl}$ |
| Global wetland storage | locwetlandstor | $S_{wg}$ |
| Reservoir storage | reservoirstor | $S_{res}$ |
| River storage | riverstor | $S_r$ |

[1] Sum of all compartments below

[2] relative water storages, only anomalies with respect to a reference
period can be evaluated

The data are stored using the network Common Data Form (netCDF) format developed by UCAR/Unidata (Unidata, 2019) and
are available at https://doi.pangaea.de/10.1594/PANGAEA.918447.

The available storages and flows are listed in Table 1 and Table 2, respectively. To convert between equivalent water heights
(e.w.h.) and volumetric units, the cell-specific continental area used in WaterGAP 2.2d is also provided. The assumed water
density is $1\,\mathrm{g\,cm^{-3}}$. The following additional static data used to produce the storages and flows are available: flow direction
(Döll and Lehner, 2002), land cover (Sect. C), location of outflow cells of global lakes and reservoirs/regulated lakes (Sect.
4.6), rooting depth (Sect. 4.4.3), maximum soil water storage ($S_{s,max}$) and reservoir commissioning year (Sect. 4.6.3). The
netCDF files contain metadata with detailed information regarding characteristics of the data (e.g., whether a storage type
contains anomaly or absolute values) and a legend where applicable.

## 5.2   Caveats in usage of WaterGAP model output

Based on feedback from data users and own experience, here we describe caveats regarding analysis of specific WaterGAP2.2d
model output with the aim of guiding output users.

- WaterGAP does not consider leap years. This implies that model output (typically provided in NetCDF file format)
  corresponding to leap years contains the "fill value" instead of a data value at the position of February $29^{th}$.

- The water balance of large lakes and reservoirs is calculated in the outflow cell only. Hence, large numerical values can
  occur for storages and flows, especially in case of very large water bodies.





**Table 2.** Standard WaterGAP output variables: 2) Flows. Units are $\mathrm{kg\,m^{-2}\,s^{-1}}$ ($\mathrm{mm\,e.w.h.\,s^{-1}}$), except $\mathrm{m^3\,s^{-1}}$ for $Q_{r,out}$ and $Q_{r,out,nat}$. Temporal resolution is monthly.

| Flow type | PANGEA file | Symbol |
|---|---|---|
| Monthly precipitation | precmon | $P$ |
| Fast surface and fast subsurface runoff[1] | qs | $R_s$ |
| Diffuse groundwater recharge | qrdif | $R_g$ |
| Groundwater recharge from surface water bodies | qrswb | $R_{g_{l,res,w}}$ |
| Total groundwater recharge[2] | qr | $R_{g_{tot}}$ |
| Runoff from land[3] | ql | $R_l$ |
| Groundwater discharge[4] | qg | $Q_g$ |
| Actual evapotranspiration [5] | evap | $E_a$ |
| Potential evapotranspiration | potevap | $E_p$ |
| Net cell runoff | ncrun | $R_{nc}$ |
| Naturalized net cell runoff[6] | natncrun | $R_{nc,nat}$ |
| Streamflow[7] | dis | $Q_{r,out}$ |
| Naturalized streamflow[7] | natdis | $Q_{r,out,nat}$ |
| Actual net abstraction from surface water | anas | $NA_s$ |
| Actual net abstraction from groundwater | anag | $NA_g$ |
| Actual total consumptive water use [8] | atotuse | $WC_a$ |

[1] fraction of total runoff from land that does not recharge the groundwater; [2] sum of qrdif and qrswb; [3] sum of qs and qrdif; [4] groundwater runoff; [5] sum of soil evapotranspiration $E_s$, sublimation $E_{sn}$, evaporation from canopy $E_c$, evaporation from water bodies and actual consumptive water use $WC_a$; [6] equals renewable water resources if averaged over e.g., 30yr time period; [7] river discharge; [8] sum of anas and anag

• In case the station correction factor $CFS$ (Sect. 4.9.1) is applied in the grid cell corresponding to the calibration station, multiplication of streamflow by $CFS$ destroys the water balance for this particular grid cell. Hence, the calculation of water balance at various spatial units requires that the amount of reduced/increased streamflow is taken into account in order to close the water balance. A direct inclusion of modified streamflow in e.g., evapotranspiration is not done to avoid physically implausible values for this variable. Water balance is preserved in case $CFA$ is used.

• Gridded model output always relates to the continental area (grid cell area minus ocean area within cell). If flows like runoff from land or diffuse groundwater recharge are simulated to occur only on the land area, i.e. the fraction of the continental area that is not covered by surface water bodies, these flow variables can be small in cells with large water bodies, e.g. groundwater recharge along the Amazon river with riparian wetlands (Fig. 11c).





- Groundwater recharge below surface water bodies (Eq. (26)) can lead to very high values in case of large surface water
  bodies and especially in inland sinks that contain large lakes. Temporal changes of this variable can be implausibly high
  ($> 10^3 \, \mathrm{mm \, yr^{-}1}$).

- Renewable water resources (Fig. 11a) are defined as the amount of precipitation that is not evapotranspired on the long
  term (30 years) under naturalized conditions (no water use, no reservoirs). Data users should keep in mind that this
  variable can only be calculated from naturalized runs and the long-term average of the variable "net cell runoff" $R_{nc,nat}$
  (Tab. 2). A calculation of renewable water resources using other model setups is not meaningful.

## 6 Model evaluation

This section comprises an evaluation of WaterGAP 2.2d using independent data of withdrawal water uses, streamflow and total
water storage anomalies TWSA as well as a comparison to the previous model version 2.2 (Müller Schmied et al., 2014).

### 6.1 Model set-up and simulation experiments

In order to compare WaterGAP 2.2d with model version 2.2 (Sect. 6.5), both versions were calibrated and run with the same
climate forcing. However, version 2.2 was calibrated using the calibration routine of Müller Schmied et al. (2014). The differ-
ences between model versions 2.2 and 2.2d are listed in Sect. A.

A homogenized combination of WATCH Forcing Data based on ERA40 (Weedon et al., 2011) (for 1901-1978) and WATCH
Forcing Data methodology applied to ERA-Interim reanalysis (Weedon et al., 2014) (for 1979-2016), with precipitation ad-
justed to monthly precipitation sums from GPCC (Schneider et al., 2015) was used. The homogenization method is described
in Müller Schmied et al. (2016a). The calibrated models have been run for the time period 1901-2016, with a spin up of 5 years
in which the model input for 1901 was used.

### 6.2 Evaluation data sets

#### 6.2.1 AQUASTAT withdrawal water use data

AQUASTAT is the Food and Agricultures Organization of the United Nations Global Information System on Water and Agri-
culture (FAO - AQUASTAT). It contains information on country-level withdrawal water uses for different sectors. These data
represent estimates mainly provided by the individual countries. In particular irrigation withdrawal water uses are, for most
countries, not based on observations. Six different withdrawal water use variables (Table 3) were available for comparison to
WaterGAP2.2d. For the evaluation, all database entries available on FAO - AQUASTAT were used, hence it contains yearly
values per country as data unit.





**Table 3.** AQUASTAT variables used for evaluating WaterGAP 2.2d potential withdrawal water use WU, including variable ID reference of AQUASTAT.

| No. | WU variable | Description | AQUASTAT equivalent (variable ID) |
|---|---|---|---|
| 1 | Total WU | Total WU from all sectors | Total freshwater withdrawal water use (4263) |
| 2 | Groundwater WU | As 1 but from groundwater resources only | Fresh groundwater withdrawal water use (4262) |
| 3 | Surface water WU | As 1 but from all surface water resources only | Fresh surface withdrawal water use (4261) |
| 4 | Irrigation WU | WU for irrigation | Irrigation withdrawal water use (4475) |
| 5 | Industrial WU | WU for manufacturing and cooling of thermal power plants | Industrial withdrawal water use (4252) |
| 6 | Domestic WU | WU for domestic sector | Municipal withdrawal water use (4251) |

#### 6.2.2 GRDC streamflow data

Monthly streamflow time series from 1319 calibration stations from the Global Runoff Data Center (GRDC) were used for evaluating performance of WaterGAP 2.2d and 2.2. As the GRDC archive has certain gaps in some regions and times and the calibration objective is to benefit from a maximum of observation data, the typical split-sampling calibration/validation is not

appropriate. Even though the same observation data are used for calibration and validation, the validation against monthly time series is meaningful as only long-term mean annual streamflow values have been used for calibration.

#### 6.2.3 GRACE total water storage anomalies

Three mascon solutions of monthly time series of total water storage anomalies TWSA from the Gravity Recovery And Climate Experiment (GRACE) satellite mission are considered. The Jet Propulsion Laboratory (JPL) mascon dataset (Watkins et al.,

2015; Wiese et al., 2018, 2016) from the GRACE Tellus Website (Monthly mass grids - global mascons (JPL RL06_v02)) is based on the Level-1 product processed at JPL. A geocenter correction is applied to the degree-1 coefficients following the method from Swenson et al. (2008), the $c_{20}$ coefficient is replaced with the solutions from Satellite Laser Ranging (SLR; Cheng et al. (2011)) and a glacial isostatic adjustment (GIA) correction is applied based on the ICE6G-D model published in Richard Peltier et al. (2018). The Center of Space Research (CSR) RL05 GRACE mascon solution (Save et al., 2016) from the

University of Texas website (CSR - GRACE RL05 mascon solutions) performs the same degree-1 and $c_{20}$ replacements (but following Cheng et al. (2013)) and removes the GIA signal based on the model from Geruo et al. (2012). Last, the Goddard Space Flight Center (GSFC) GRACE mascon solutions Luthcke et al. (2013) from the Geodesy and Geophysics Science Research Portal (NASA Earth Sciences - Geodesy and Geophysics) applies trend corrections for the $c_{21}$ and $s_{21}$ coefficients following Wahr et al. (2015) in addition to the degree-1, $c_{20}$ and GIA corrections described for CSR.

Monthly TWSA values are provided on $0.5° \times 0.5°$ grid cells for JPL and CSR, while GSFC provides equal area grids with a spatial resolution of around $1° \times 1°$ at the equator. In this study, the grid values are spatially averaged over 143 river basins





with a total area of more than 200,000 $\mathrm{km}^2$ each, out of the 1319 basins used for calibration. The considered time span for this study is 2003-2015 full years of data, limited by available monthly solutions from GSFC between January 2003 and July 2016.

## 6.3 Evaluation metrics

### 745 6.3.1 Nash-Sutcliffe Efficiency

The Nash-Sutcliffe efficiency metric $NSE$ $(-)$ (Nash and Sutcliffe, 1970) is a traditional metric in hydrological modelling. It provides an integrated measure of modelling performance with respect to mean values and variability and is calculated as:

$$NSE = 1 - \frac{\sum_{i=1}^{n}(O_i - S_i)^2}{\sum_{i=1}^{n}(O_i - \overline{O})^2} \tag{36}$$

where $O_i$ is observed value (e.g., monthly streamflow), $S_i$ is simulated value and $\overline{O}$ is mean observed value. The optimal value

of $NSE$ is 1. Values below 0 indicate that the mean value of observations is better than the simulation (Nash and Sutcliffe, 1970). For assessing the performance of low values of water abstraction (Sect. 6.4.1), a logarithmic $NSE$ was calculated in addition by applying logarithmic transformation before calculation of the performance indicator.

### 6.3.2 Kling-Gupta Efficiency

The Kling-Gupta efficiency metric $KGE$ (Kling et al., 2012; Gupta et al., 2009) transparently combines the evaluation of bias,

variability and timing and is calculated (in its 2012 version) as:

$$KGE = 1 - \sqrt{(KGE_r - 1)^2 + (KGE_b - 1)^2 + (KGE_g - 1)^2} \tag{37}$$

where $KGE_r$ is the correlation coefficient between simulated and observed values $(-)$, an indicator for the timing, $KGE_b$ is the ratio of mean values (Eq. (38)) $(-)$, an indicator of biases regarding mean values and $KGE_g$ is the ratio of variability (Eq. (39)) $(-)$, an indicator for the variability of simulated $(S)$ and observed $(O)$ values.

$$KGE_b = \frac{\mu_S}{\mu_O} \tag{38}$$

$$KGE_g = \frac{CV_S}{CV_O} = \frac{\sigma_S/\mu_S}{\sigma_O/\mu_O} \tag{39}$$

where $\mu$ is mean value, $\sigma$ is standard deviation and $CV$ is coefficient of variation. The optimal value of $KGE$ is 1.

### 6.3.3 TWSA-related metrics

For the evaluation of total water storage anomaly performance, the following metrics were used: $R^2$ (coefficient of determination) as strength of linear relationship between simulated and observed variables, the amplitude ratio as indicator for variability and trend of both GRACE and WaterGAP data. Amplitude and trends were determined by a linear regression for estimating





the most dominant temporal components of the GRACE time series. The time series of monthly TWSA was approximated by a constant $a$, a linear trend $b$, an annual and a semi-annual sinusoidal curve as follows

$$y(t) = a + b * t + c * sin(2 * \pi * t) + d * cos(2 * \pi * t) + e * sin(4 * \pi * t) + f * cos(4 * \pi * t) + r \tag{40}$$

where $r$ denotes the residuals. The parameters $a$ to $f$ were estimated via least-squares adjustment. The annual amplitude can be computed by $A = sqrt(c^2 + d^2)$, and thus, the annual ratio was calculated by $A_{WGHM}/A_{GRACE}$.

## 6.4 Evaluation results

### 6.4.1 Water witdrawals

The performance of WaterGAP potential withdrawal water uses is generally of reasonable quality (Fig. 5). Highest agreement in terms of performance indicator is shown for the total withdrawal water uses with both efficiency metrics close to the optimum value. Slightly less agreement is visible for the separation to groundwater withdrawals (underestimation of WaterGAP) and surface water withdrawals (overestimation of WaterGAP). The domestic sectoral withdrawal water uses are best simulated with WaterGAP, followed by the industrial sector. Here, large differences between $NSE$ and logarithmic $NSE$ are visible,

indicating that WaterGAP has specific problems in representing the small values and tending to a general overestimation of industrial withdrawal water uses. WaterGAP performs reasonably well in the irrigation sector with slightly better logarithmic $NSE$ metric but with overall lowest sectoral performance in terms of $NSE$ (no visible direction in under- or overestimation).

### 6.4.2 Streamflow

With a median $NSE$ ($KGE$) of 0.52 (0.61) for the fit of monthly streamflow time series at 1319 gauging stations (Fig. 6),

performance of WaterGAP 2.2d in terms of streamflow is rather satisfying. However, $NSE$ values below 0 for 259 stations show the complete failure of WaterGAP2.2d to simulate streamflow dynamics in one fifth of the evaluated basins. The median for $KGE_r$ of 0.79 indicates a relatively satisfactory simulation of the timing of monthly streamflows both seasonally and interannually. As the model is calibrated to match long-term annual river discharge (Sect. 4.9), the median of bias measure $KGE_b$ is, with a value of 1.01, close to the optimum value. In rare cases, values outside the range of 0.9-1.1 occur as for

calibration the individual basins were run for the calibration time period (plus 5 initialization years) while the evaluation run was a global run from 1901 to 2016. In the normal global runs, water demand can be fulfilled from neighbouring grid cells while this is not possible in the calibration runs. This partially explains the larger biases also seen in Fig. 8. Streamflow variability is mostly underestimated by WaterGAP2.2d, and median $KGE_g$ is 0.85 (Fig. 6).

When analyzing the spatial distribution of streamflow performance indicators, note that a highly seasonal streamflow regime

tends to lead to high $NSE$ and $KGE_g$ not due to the quality of the evaluated hydrological model but the highly seasonal precipitation input. The global distribution of $NSE$ classes shows a diverse pattern (Fig. 7). Whereas large parts of central Europe, Asia and southern America are simulated reasonably well, the performance in northern America and large parts of Africa is in many cases below a value of 0.5. Based on $NSE$ alone it remains unclear, why WaterGAP consistently fails



**Figure 5.** Comparison of potential withdrawal water uses from WaterGAP 2.2d with AQUASTAT (FAO - AQUASTAT). Each data point represents one yearly value (if present in the database) per country for the time span 1962-2016

.

to satisfactorily simulate large parts of the well observed northern America. Further insights can be gained by assessing the spatial distribution of $KGE$ and its components (Fig. 8). The broad picture of overall $KGE$ (Fig. 8a) is similar to the $NSE$ spatial distribution (Fig. 7). In a large fraction of river basins with low $NSE$ and $KGE$, the timing is off, with $KGE_r$ <0.5. One reason could be the inappropriate modelling of the dynamics of lakes and wetland (mainly in Canada) and of reservoir regulations. As most snow-dominated basins in Alaska, Euroe and Asia show a reasonably high $KGE_r$ of >0.8, it is not likely that snow dynamics are the dominant cause for low correlations between observed and simulated streamflow. For many other regions (e.g., central Asia and Nile basin), streamflow regulations due to reservoir as well as the timing of water abstractions are most likely to cause low performance in timing. The indicator of variability $KGE_g$ shows a medium to strong underestimation of streamflow variability in most of the northern snow-dominated basins. Underestimation in the Amazon basin is caused by



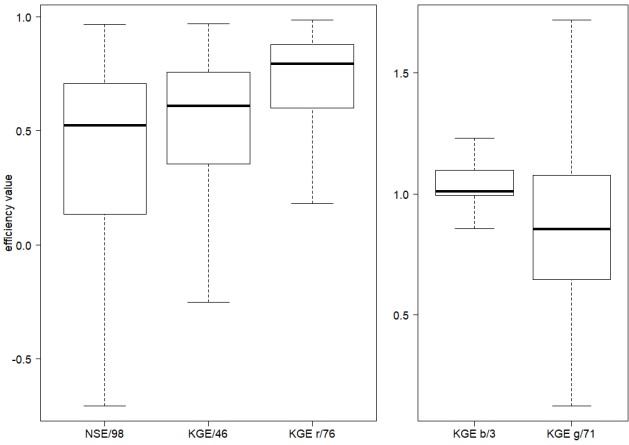

**Figure 6.** Efficiency metrics for monthly streamflow of WaterGAP 2.2d at the 1319 GRDC stations with $NSE$, $KGE$ and its components. Outliers (outside 1.5x inter-quartile range) are excluded but number of stations that are defined as outliers are indicated after the metric.

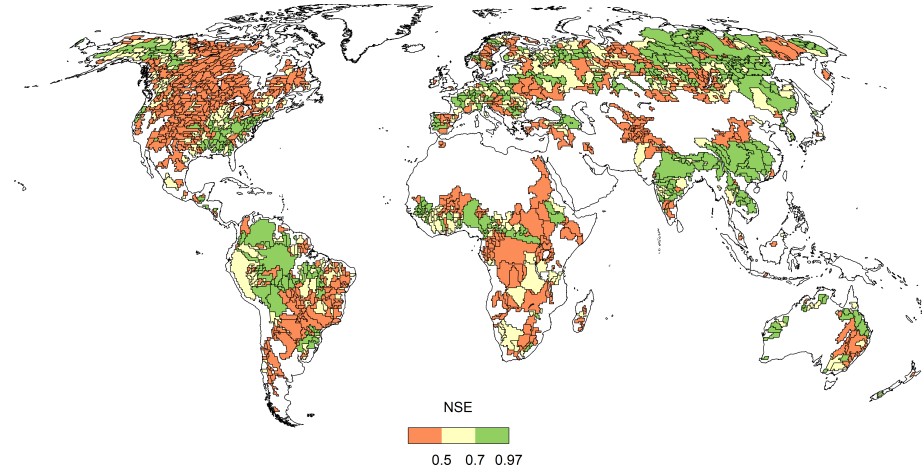

**Figure 7.** Classified $NSE$ efficiency metric for the 1319 river basins in WaterGAP 2.2d.

the inability of WaterGAP to simulate wetland dynamics there. There are also many gauging stations for which WaterGAP overestimates seasonality, even by more than 50%. Further research and development is needed for improving the GHMs in
this respect (Veldkamp et al., 2018).

### 6.4.3 TWSA

WaterGAP 2.2d underestimates the mean annual TWSA amplitude in 54% of the 143 investigated river basins by more than 10% (Fig. 9). Most of these basins are located in Africa, in the northern and monsoon regions of Asia, in Brazil and in western North America. In contrast, the mean annual amplitude is overestimated in western Russia as well as in eastern and central



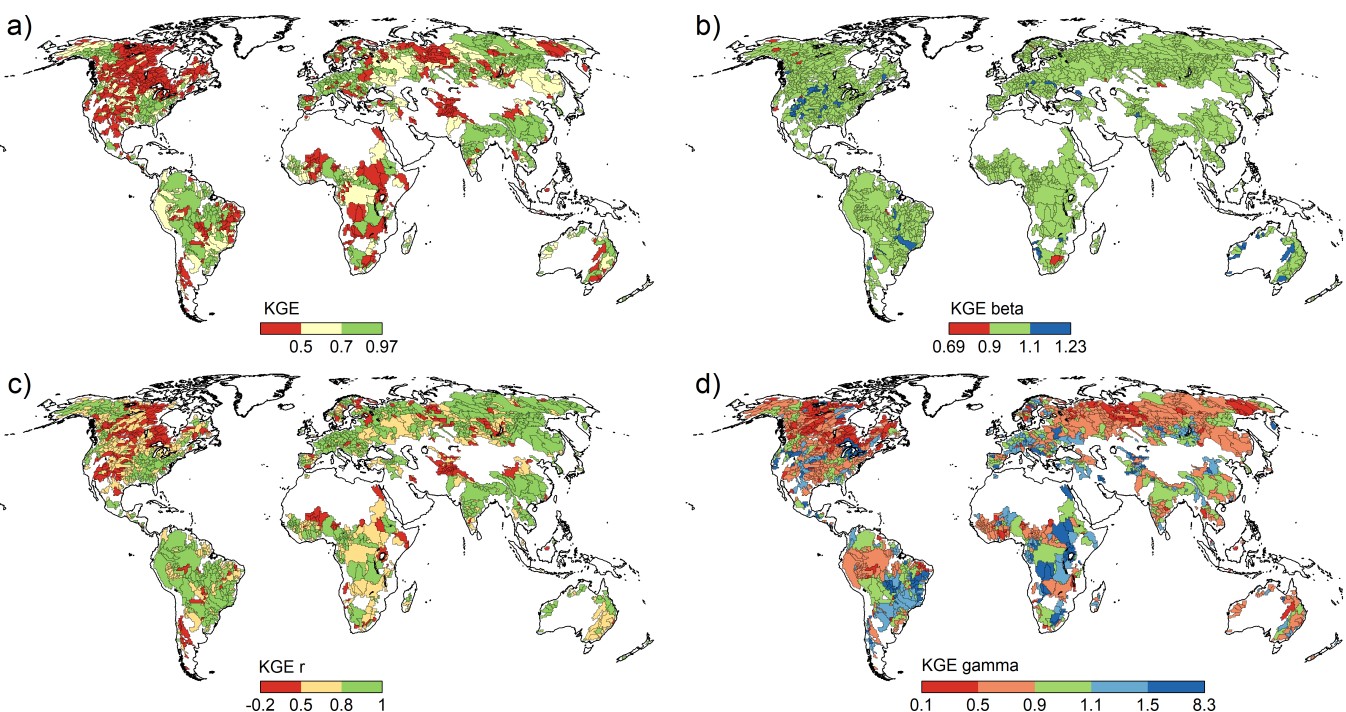

**Figure 8.** Classified $KGE$ efficiency metric and its components for the 1319 river basins in WaterGAP 2.2d.

North America. The correlation coefficient exceeds 0.7 in almost 75% of the river basins and 0.9 in 22%. Only 8% of the basins show a correlation coefficient below 0.5.

The comparison of the TWSA trends shows that GRACE and WaterGAP 2.2d agree in the sign of the trend for 63% of the 143 basins, for example most European basins, nearly the entire South American continent, and several basins in North America, Asia and Australia, but trends are often underestimated, e.g. in the Amazon and western Russia. Basins with different

signs of the trend are scattered around the globe. GRACE suggests strong decreases of water storage in Alaskan basins, which is likely due to glacier mass loss, while WaterGAP determines small mass increase, likely because WaterGAP does not simulate glaciers. Comparing the spatial pattern of Figs. 9 and 8, no obvious interrelation can be derived between the performances of streamflow and TWSA.

### 6.5  Performance comparison between WaterGAP2.2d and WaterGAP 2.2

Performance differences are expected due to modifications in model algorithms and the calibration routine (for details on modifications see Appendix. A). When comparing the $NSE$ of monthly streamflow (Figs. 7 and D7), the broad picture is similar. WaterGAP 2.2d shows some improvements in northern South America (esp. Amazon) but in the same time gets worse in southern South America. Slight decreases in performance for WaterGAP 2.2d are observed in southern Africa. No major changes are visible in North America, Europe and Asia, with small bidirectional changes. $KGE$ patterns are also relatively



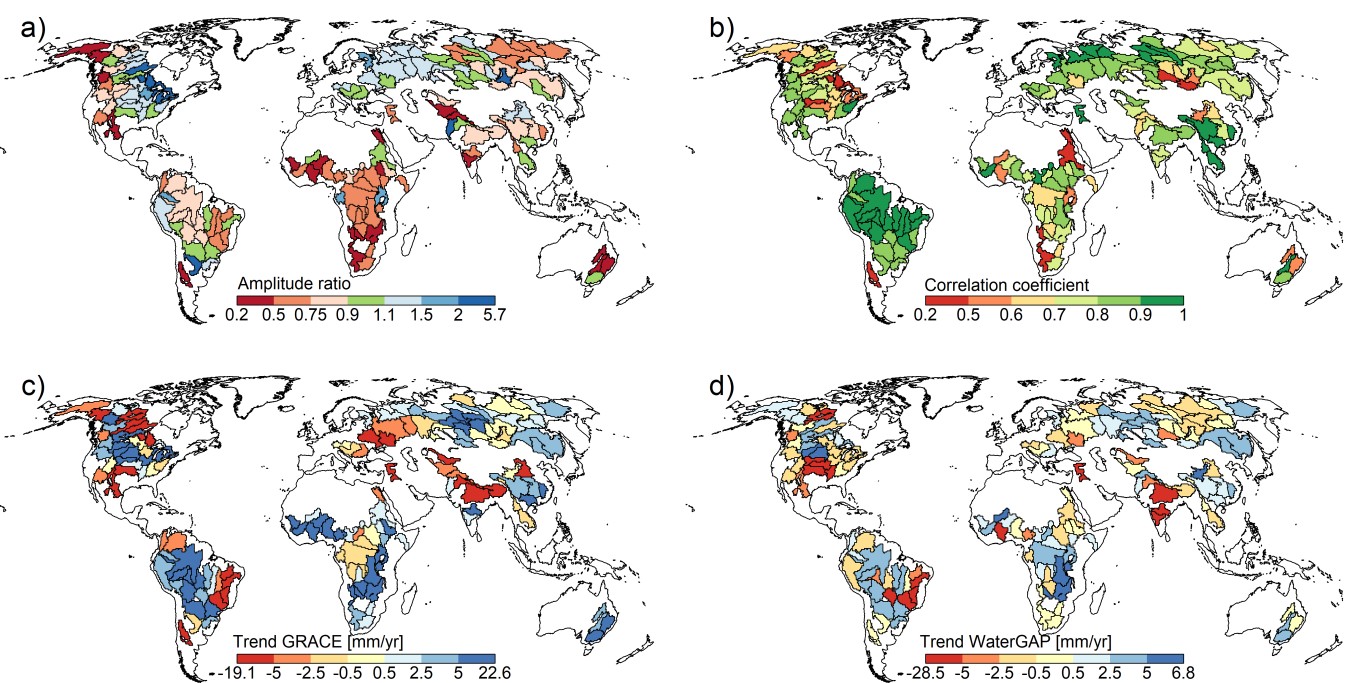

**Figure 9.** Comparison of basin-average TWSA of WaterGAP 2.2d and the average values of three GRACE mascon products for 143 basins larger than 200,000 km$^2$, with a) ratio of amplitude (reddish colours indicate underestimated amplitude of WaterGAP, vice versa for bluish), b) correlation coefficient, c) trend of GRACE and d) trend of WaterGAP 2.2d. All values based on the time series 01/2003 - 12/2015.

similar for both versions (Figs. 8 and D4) and follow generally the differences in $NSE$. However, there are more regions in Europe and Asia where WaterGAP 2.2d performs better in overall $KGE$, resulting mainly from an improvement of $KGE_r$. This is also visible in the number of basins per Koeppen climate zone, where especially in the tropical A and dry B climates WaterGAP 2.2d has higher performance in $KGE_r$ (Table 4). The differences of $KGE_b$ are negligible.

$KGE_g$ shows significant differences between both model versions, in both directions, but performance of WaterGAP2.2d is significantly better. Summarizing the basin statistics per Koeppen climate zone, 272 instead of only 241 basins are within $\pm 10\%$ of observed variability in WaterGAP 2.2d in all climate zones except E (Table 5). Less river basins (56% compared to 61% in 2.2) are subject to an underestimation of streamflow variability. However, the number of basins with overestimation increases slightly from 21% for WaterGAP2.2 to 23% for WaterGAP 2.2d.

The performance of streamflow of the 1319 basins (Fig. D5) is similar for most indicators. The higher variation in $KGE_b$ stems from modifications in the calibration routine, where up to a $\pm$ 10% uncertainty of observed streamflow is allowed. Similarly, the performance statistics of both, streamflow and TWSA (for the 143 basins > 200,000 km$^2$) are very similar for both model versions (Fig. D6).

A comparison of simulated seasonality of streamflow and TWSA in 12 selected large river basins across climate zones shows that performance with respect to both variables are improved in WaterGAP 2.2d for the Lena, Amazon and Yangtze





**Table 4.** Model performance with respect to streamflow timing: Number of calibration basins per $KGE_r$ category and Köppen–Geiger climate zone

| Model | Class | $KGE_r$ | A | B | C | D | E | Sum |
|-------|-------|---------|-----|----|-----|-----|----|-----|
|       | 1     | >0.8    | 159 | 35 | 173 | 251 | 16 | 634 |
| 2.2d  | 2     | 0.5-0.8 | 109 | 47 | 77  | 200 | 17 | 450 |
|       | 3     | <0.5    | 17  | 45 | 18  | 146 | 9  | 235 |
|       | 1     | >0.8    | 160 | 28 | 169 | 250 | 16 | 623 |
| 2.2   | 2     | 0.5-0.8 | 104 | 46 | 80  | 202 | 18 | 450 |
|       | 3     | <0.5    | 21  | 53 | 19  | 145 | 8  | 246 |

**Table 5.** Model performance with respect to streamflow variability: Number of calibration basins per $KGE_g$ category and Köppen–Geiger climate zone

| Model | Class | $KGE_g$ | A | B | C | D | E | Sum |
|-------|-------|---------|-----|----|-----|-----|----|-----|
|       | 1     | >1.5    | 37  | 15 | 22  | 29  | 4  | 107 |
|       | 2     | 1.1-1.5 | 46  | 22 | 71  | 58  | 5  | 202 |
| 2.2d  | 3     | 0.9-1.1 | 59  | 26 | 78  | 99  | 10 | 272 |
|       | 4     | 0.5-0.9 | 124 | 51 | 88  | 281 | 10 | 554 |
|       | 5     | <0.5    | 19  | 13 | 9   | 130 | 13 | 184 |
|       | 1     | >1.5    | 29  | 16 | 19  | 27  | 3  | 94  |
|       | 2     | 1.1-1.5 | 46  | 18 | 57  | 54  | 6  | 181 |
| 2.2   | 3     | 0.9-1.1 | 48  | 21 | 74  | 88  | 10 | 241 |
|       | 4     | 0.5-0.9 | 141 | 49 | 109 | 277 | 10 | 586 |
|       | 5     | <0.5    | 21  | 20 | 12  | 151 | 13 | 217 |

basins (Fig. 10). Simulations for the Congo, Mekong, Mackenzie and Murray basins do not differ. In some basins (Orange, Volga) the simulation of streamflow is improved in WaterGAP 2.2d whereas TWSA seasonality remains similar. In other basins (Rio Parana) seasonality agreement of TWSA remains the same for WaterGAP 2.2d but streamflow seasonality agreement decreases.

## 7 Examples of model application

This section provides some examples of the WaterGAP 2.2d model applications for characterizing historical freshwater conditions at the global scale.



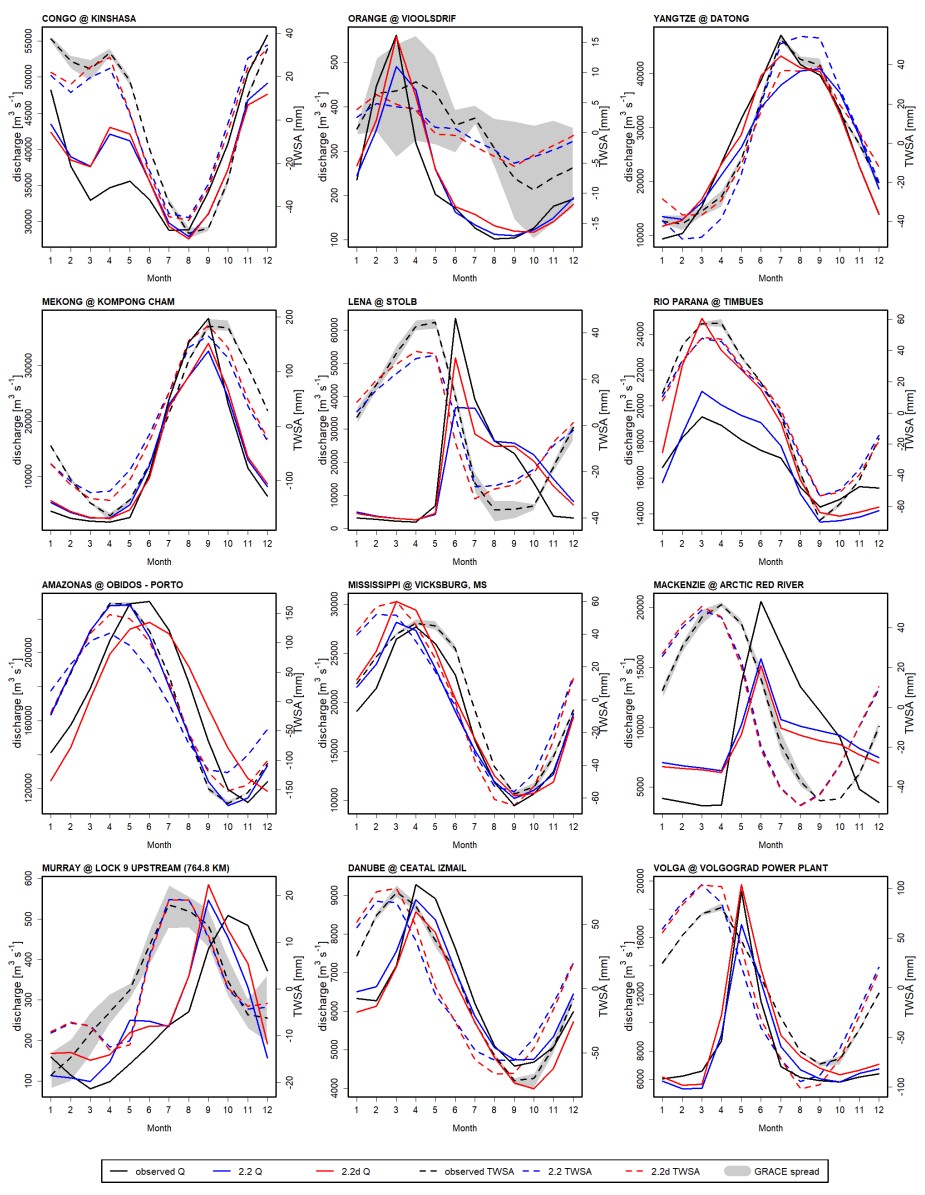

**Figure 10.** Seasonality of streamflow and TWSA of selected large river basins: Model results of WaterGAP2.2d and WaterGAP2.2 as well as streamflow and TWSA observations.

## 7.1 Model setup

The model setup is similar to those for the evaluation (Sect. 6.1). For the purpose of model examples, the model was run in both the naturalized (nat) and the anthropogenic (ant) variant (Sect. 4.1).





## 7.2 Spatial patterns of the global freshwater system

### 7.2.1 Renewable water resources

The quantification of (total) renewable water resources is one of the key elements of WaterGAP model application. They are defined as the long-term annual difference between precipitation and actual evapotranspiration of a spatial unit, or long-term annual net cell runoff. As runoff and evapotranspiration are influenced by human interference, renewable water resources are calculated based on the naturalized model variant, by averaging $R_{nc}$ (Sect. 4.7.3) over e.g., a 30-yr time period, resulting in $R_{nc,lta,nat}$. On around 42.6% of the global land area (excluding Greenland and Antarctica), total water resources are calculated to be <100 mm yr$^{-1}$ during the period 1981-2010, whereas on 19.8% values are >500 mm yr$^{-1}$(Fig. 11a). Globally averaged renewable water resources are computed to be 307 mm yr$^{-1}$ or 40678 km$^3$ yr$^{-1}$. The global map of inter-annual variability of runoff production (Fig. 11b), here defined as the ratio of runoff in a 1-in-10 dry year to total renewable water resources, shows regions with relatively constant and relatively variable annual runoff generation, in bluish and reddish colours, respectively. High variability is linked with low renewable water resources.

Total renewable water resources include renewable groundwater resources which are the sum of long-term average diffuse groundwater recharge $R_g$, (Fig. 11c) and long-term average point (or focused) groundwater recharge from surface water bodies $R_{g_{l,res,w}}$ (Fig. 11d). While focused recharge is the major type of groundwater recharge in some (semi)arid grid cells, its quantification is highly uncertain, and diffuse groundwater recharge dominates in most cells. For 1981-2010, global mean diffuse groundwater recharge is calculated as 111.0 mm yr$^{-1}$, and global mean focused recharge as 12.8 mm yr$^{-1}$. Note that as $R_g$ is calculated on (time-variable) land area (continental area minus fraction of lakes, reservoirs, wetlands) but is related to continental area in the standard output (Sect. 5.2), grid cells with large gaining surface water bodies, e.g. wetlands along the Amazon river, show significant lower $R_g$ values than surrounding grid cells.

The sum of diffuse and focused renewable groundwater resources amounts to 40% of total renewable water resources, highlighting the important contribution of groundwater resources. There have been a number of studies on the potential impact of climate change on renewable groundwater resources (either including or excluding focused recharge), in which WaterGAP was applied as impact model (Portmann et al., 2013; Döll, 2009; Döll et al., 2018; Herbert and Döll, 2019).

### 7.2.2 Streamflow

Streamflow (or river discharge) $Q_{r,out}$ is the model output that integrates all model components and human intervention, routing runoff along the river network. The global map of long-term average annual streamflow under anthropogenic conditions distinctly shows the very high spatial variability of streamflow and very distinctly the large river systems of the Earth (Fig. 12a). Temporal variability of monthly streamflow is much higher in the (semi)arid areas than in humid area, increasing the spatial discrepancy of streamflow; this can be seen in Fig. 12c, which presents the ratio of the statistical low flow $Q_{90}$ (the streamflow that is exceeded in 9 out of 10 months) to long-term average annual streamflow. The regions with a ratio of less than 5% of low flow contribution on average streamflow (the hydrological highly variable regions) follow in general the definition of (semi)arid grid cells (Fig. B1) with some exceptions as for northern Asia. Different from the spatial pattern of interannual variability of

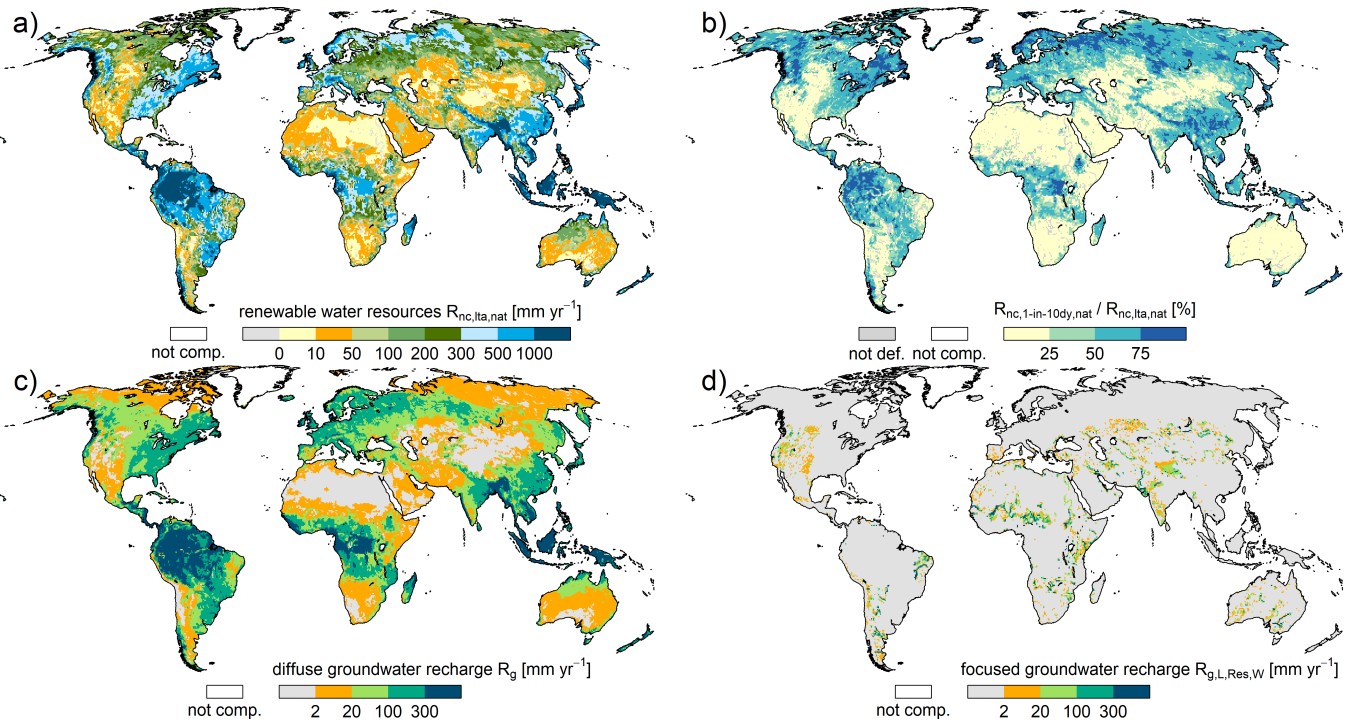

**Figure 11.** Water resources assessment 1981-2010 using WaterGAP 2.2d, with a) total renewable water resources defined as long-term annual net cell runoff $R_{nc,lta,nat}$ [mm yr$^{-1}$], b) 1-in-10 dry year runoff generation in percent of total renewable water resources [%], c) long-term annual diffuse groundwater recharge $R_g$ [mm yr$^{-1}$], d) long-term annual focused groundwater recharge $R_{g_{l,res,w}}$ [mm yr$^{-1}$]. Results are based on naturalized model runs. In a) note that negative values for total water resources are possible (Sect. 4.7.3). In b) areas where the denominator is $< 10^{-5}$ are labelled as not defined.

long-term average net cell runoff (Fig. 11b), the spatial pattern of streamflow is characterized by low temporal variability in cells with large rivers, due to the integration of runoff from diverse grid cells as well as large water storage capacities in lakes, 890 reservoirs or wetlands.

The impact of human interventions (human water use and man-made reservoirs) on streamflow is assessed in Fig. 12b for long-term average and Fig. 12d for statistical low flow indicator $Q_{90}$ (please note the different legend for both subfigures). In general, human interventions reduce long-term average streamflow by at least 10% (50%) in 11.3% (1.8%) of the global land area, mainly due to reduced groundwater discharge to lakes, reservoirs, wetlands and rivers as a consequence of groundwater 895 abstractions, in particular groundwater depletion (compare the red pattern with net abstraction from groundwater in Fig. 15a). There is only a minor share (0.7%) of global land area, where long-term annual streamflow has been increased by more than 10% due to human interventions (mainly return flow from groundwater abstractions). The impact of human interventions on $Q_{90}$ is more pronounced (Fig. 12d). Large reddish patterns (consistent to net abstraction from groundwater in Fig. 15a) indicate the reduction of low flows by at least 10% (90%) on 29.7% (14.4%) of the global land area. However, there are also bluish



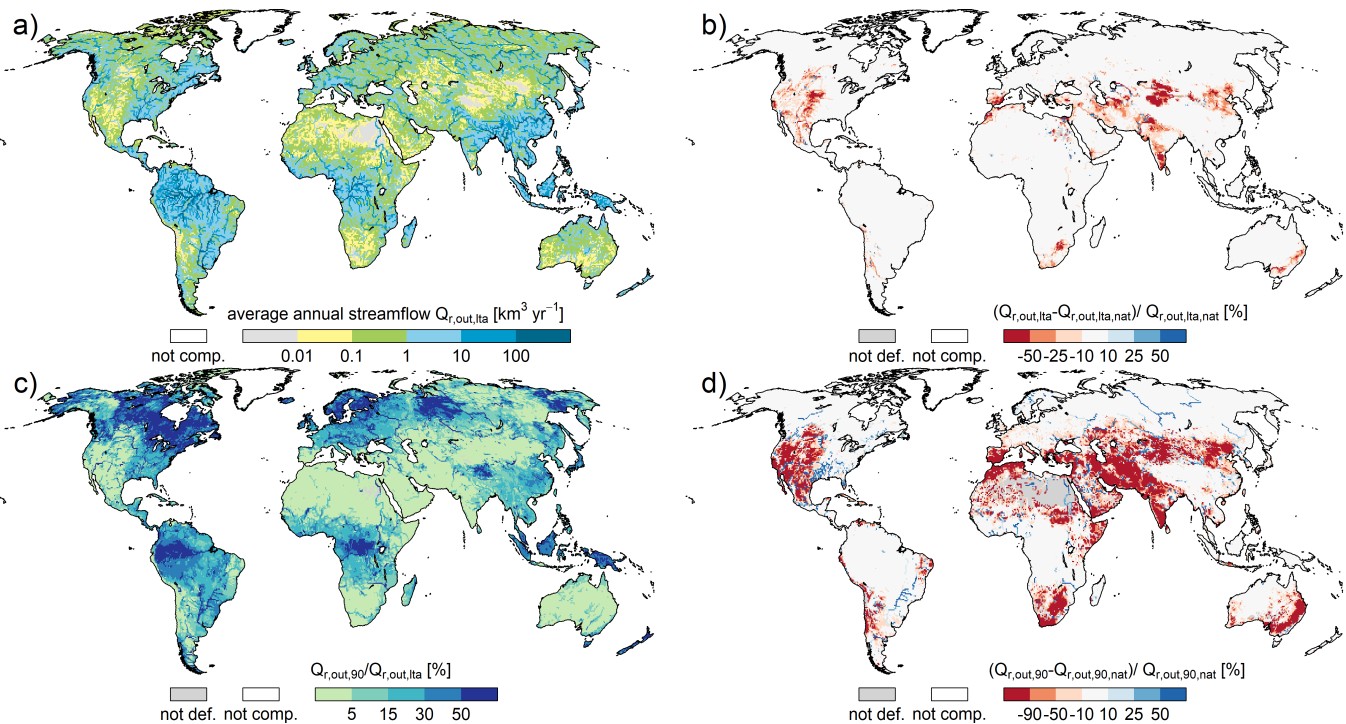

**Figure 12.** Streamflow indicators of WaterGAP 2.2d for 1981-2010 with a) long-term average annual streamflow $Q_{r,out,lta}$, (km$^3$ yr$^{-1}$), b) indication of streamflow alteration due to human water use and man-made reservoirs, reddish indicates less streamflow for ant conditions, blue the opposite, c) statistical monthly low flow $Q_{r,out,90}$ in percent of $Q_{r,out,lta}$, d) differences of long term average statistical monthly low flows as indication of low flow alteration due to human water use and man-made reservoirs. Not defined are areas where the denominator is smaller than $10^{-5}$ km$^3$ yr$^{-1}$.

river systems visible which represents a global land area of 5.3% with increase of low flows of more then 10%. Those areas are located downstream of large reservoirs that due to their storage capacity attenuate the flow regime towards a temporally less variable streamflow. As WaterGAP 2.2d considers only the largest reservoirs with reservoir management algorithm and handles the remaining ~6000 reservoirs of GRanD as unmanaged water bodies, the impact of streamflow regulation is most likely underestimated.

**7.2.3   Water stress**

A major motivation for the initial WaterGAP development was to consistently assess water stress on all land areas of the globe (Döll et al., 2003; Alcamo et al., 1998). A common water stress indicator ($WSI$) is calculated as the ratio of long-term average annual withdrawal water uses (or water abstractions of withdrawal water use) (Sect. 3) and total renewable water resources for different spatial units (e.g., river basins). Renewable water resources in a basin are equal to long-term average naturalized

annual streamflow at the outlet of the basin. $WSI$ of 0.2 - 0.4 is generally assumed to indicate mild water stress and $WSI$>0.4

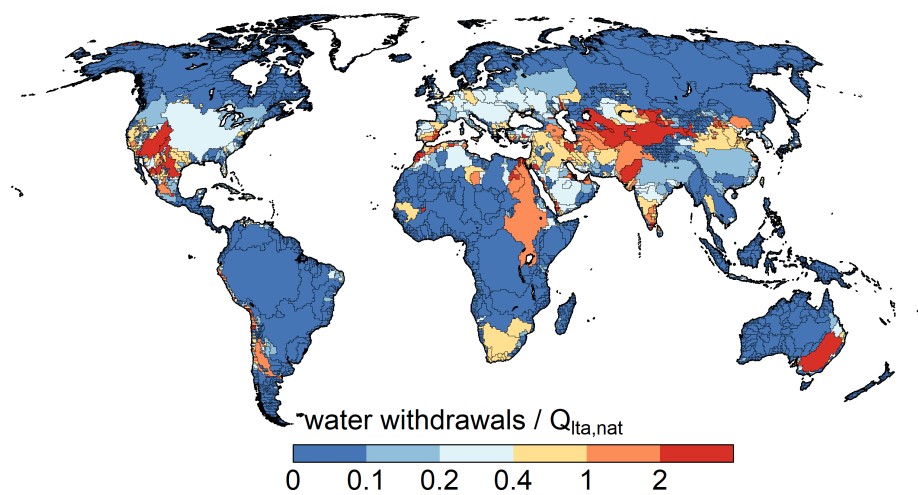

**Figure 13.** Water stress in zero-order river basins for 1981-2010, computed as the ratio of the basin sum of long-term average annual potential total withdrawal water uses (Sect. 3) to long-term average annual streamflow $Q_{r,out,lta,nat}$ of the basin (i.e. at its outflow cell to the ocean or at its inland sink).

severe water stress (e.g., Greve et al., 2018), while $WSI{>}1.0$ represents a situation, where withdrawal water uses are larger than renewable water resources, indicating extreme water scarcity (e.g., Veldkamp et al., 2017). For this example, zero-order river basins (basins that drain to the oceans or inland sinks) were chosen as spatial units (Fig. 13). River basins covering 73.6% of global land area have a $WSI{<}0.2$ and thus are calculated to have none to only minor water stress. Mild (severe, but below

extreme) water stress is represented in river basins that cover 9.7% (6.9%) of global land area. Extreme water stress ($WSI{>}1.0$) is simulated in river basins that cover 9.9% of global land area (red colours in Fig. 13). The spatial pattern of river basins with water stress is similar to the pattern of modification of statistically low flow alteration due to human interventions (Fig. 12d).

Output of global models is usually shown in the form of two-dimensional planar global maps, which are necessarily distorted. While the Robinson projection that we normally use when presenting WaterGAP results is pleasing to the eye, it does not

preserve the actual area of the land surface, and areas closer to equator are shown relatively smaller than the areas closer to the poles. Using an equal-area projection as in Fig. 14b, Africa is shown larger than in the traditional Robinson maps. For Africa, large blue areas indicate high per-capita total renewable water resources. However, very few people live in these large areas. For representing water resources for people instead of on areas, cartograms with population numbers as distorter can be used (Fig. 14 b). In cartograms, map polygons representing spatial units on the Earth's surface are distorted in a way that the units'

polygon areas on the map are proportional to a quantitative attribute of the spatial unit (Döll, 2017), here the population in $0.5°$ $\times\ 0.5°$ grid cells in 2010. The latter was derived by aggregating 2010 GPWv3 gridded population estimate for the year 2010 CIESIN (2016) from its original resolution of 2.5 arc-minutes. Clearly, with a higher share of red areas, the cartogram indicates a world with less water availability than the "normal" map and it leads the eye to regions where humans are affected by water scarcity.



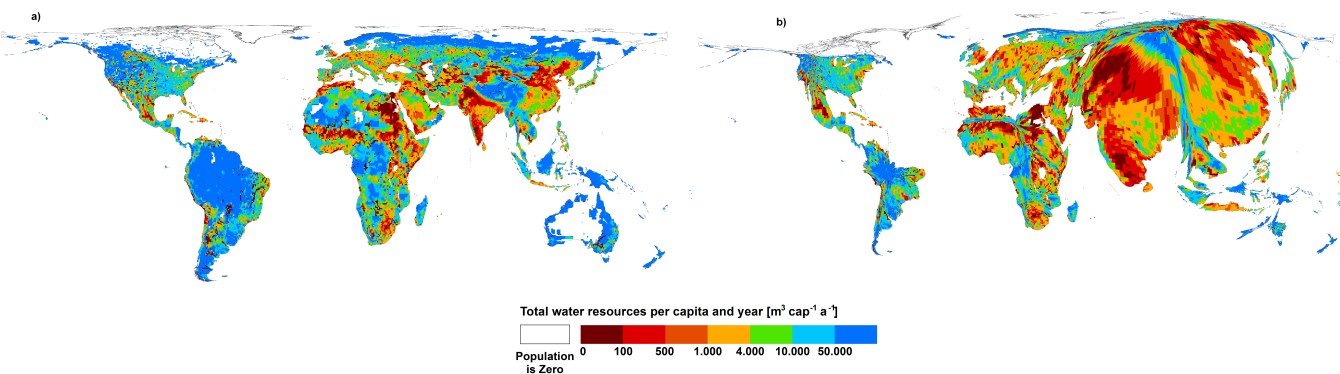

**Figure 14.** Water availability indicator per capita renewable water resources $Q_{r,out,lta,nat}$ ($\mathrm{m}^3\,\mathrm{cap}^{-1}\,\mathrm{yr}^{-1}$) for 1981-2010 visualized in a) an equal area projection and b) as a cartogram with population in 2010 as distorter. In the cartogram each half degree grid cell is distorted such that its area is proportional to the population of the grid cell.

### 7.2.4 Water abstractions

With human water use being essential for the estimation of water stress, quantification of sectoral water uses was a focus already in the initial stages of WaterGAP development (Alcamo et al., 1998). However, a distinction of the sources of water abstractions and the sinks of return flows (groundwater or surface water) was only implemented later, such that potential net abstractions from groundwater and from surface water could be computed (Döll et al., 2012, 2014). Model refinements (see Sect. A2) have lead to a more consistent computation of actual net abstractions from both sources. The general patterns of potential net abstractions (Fig. 15a and b) are consistent with the earlier assessment of Döll et al. (2012). The ratio of actual to potential net abstractions (Fig. 15c) shows a heterogeneous pattern, with adjacent grid cells with values below 0.9 and above 1.1. This is explained by the option to satisfy water demand from a neighbouring grid cell. In case of negative $NA_s$, potential and actual values are always the same, as it is assumed in the model that $NA_g$ can always be fulfilled so that return flows to surface water are not changed. There are only a few longer river stretches with ratios below 1, where global actual $NA_s$ is smaller than the potential value. In grid cells, in which a positive $NA_s$ related to irrigation cannot be fulfilled due to lack of water, the return flows to groundwater decrease and actual values of $NA_g$ increase compared to their potential values. For example, in case of a positive (negative) potential $NA_g$, a ratio of 1.1 (0.9) means that the difference between actual and potential $NA_g$ is a 10% of the absolute value of potential $NA_g$. In most grid cells, actual $NA_g$ is equal to the potential value (Fig. 15d).



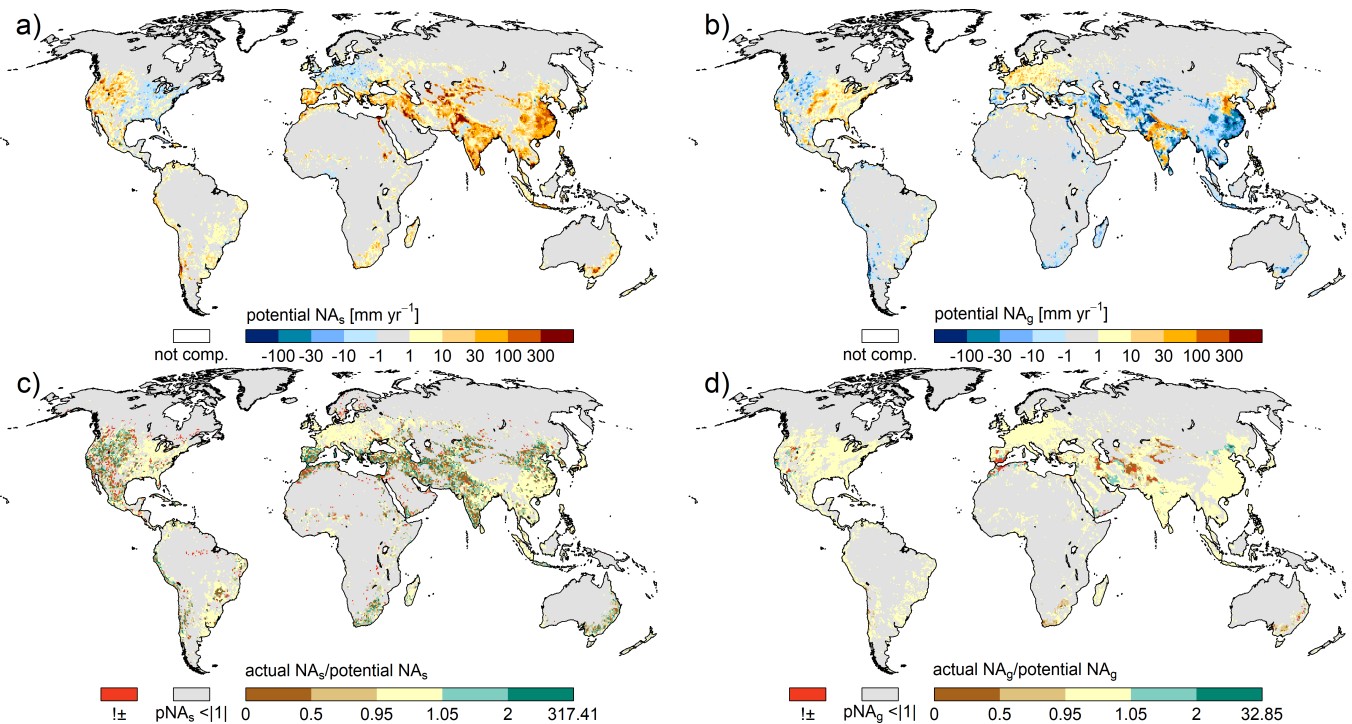

**Figure 15.** Long term (1981-2010) annual net abstractions: potential net water abstractions from surface water bodies a), potential net water abstractions from groundwater b), ratio of actual net water abstractions from surface water bodies to its potential value c) and ratio of actual net water abstractions from ground water to its potential value. In a) and b) negative values indicate a net recharge of surface water and groundwater, respectively, due to return flows caused by human water use, while positive values indicate a net removal of water from the sources. In c) and d), cells with potential net water abstractions smaller than $|1|$ mm yr$^{-1}$ are greyed out. Furthermore, grid cells where the sign of water abstractions changes between potential and actual net abstractions are displayed in red.

## 7.3 Globally aggregated components of the land water balance components

### 7.3.1 Major water balance components

Estimation of globally aggregated components of the land water balance components is an intrinsic application field of GHMs. Independent of the time span assessed in Table 6, streamflow into oceans and inland sinks, equivalent to global renewable water resources, amounts to around 40000 km$^3$ yr$^{-1}$ (with a range of around 1000 km$^3$ yr$^{-1}$). Actual evapotranspiration is estimated to be around 71000 km$^3$ yr$^{-1}$ (with a range of 1200 km$^3$ yr$^{-1}$). Renewable water resources estimates are in the range of the estimates of previous WaterGAP model versions and of other global assessments (compare Müller Schmied et al. (2014), their Table 3). Temporal trends of precipitation, actual evapotranspiration and streamflow may not be reliable due to uncertainty of the climate forcing and WaterGAP2.2d. With less than 10$^{-1}$ km$^3$ yr$^{-1}$, the water balance error is negligible (Table 6), which is an improvement compared to earlier model versions (see Müller Schmied et al. (2014), their Table 2).





### 7.3.2 Water storage components

Total actual consumptive water use has increased over time and reaches the maximum in the most recent time period 2001-2016. Negative values of actual net abstraction from surface water indicate that human water use causes on average a small net recharge of the groundwater, which has, however, decreased over time. There has been a continuous total water storage decline on land since 1961, which accelerates over the decades. The decreasing trend is mainly caused by human water use leading to a strongly accelerating groundwater depletion (Table 7). Loss of groundwater was partially balanced by increased impoundment of water in man-made reservoirs, but impoundment has decelerated. However, WaterGAP2.2d underestimates water storage increases because only the largest reservoirs are simulated as reservoirs including their commissioning year and because the GRanD v1.1 database used in WaterGAP2.2d does not include some of the major reservoirs that were put into operation after 2000 (Cáceres et al., 2020). Soil water storage also contributes significantly to total water storage changes, showing increases since 1981. Different from what may be expected due to global warming, simulated global snow storage does not decrease over time (Table 7).

### 7.3.3 Water use components

For the time period 1991-2016, Table 8 presents global sums of annual sectoral potential withdrawal water uses and consumptive water uses as well as the respective fractions that are taken from groundwater (Sect. 3.3). Potential net abstractions from surface water (groundwater) are calculated by GWSWUSE to be 1406 (-153) $\mathrm{km^3\,yr^{-1}}$ (Sect. 3.3). Actual net abstractions from surface water (groundwater) are computed by WGHM to be 1304 (-66) $\mathrm{km^3\,yr^{-1}}$ due to restricted surface water availability and consequently less return flows to groundwater from irrigation with surface water. It is thus estimated that 98.8% of potential consumptive water use of 1253 $\mathrm{km^3\,yr^{-1}}$ could be fulfilled during 1991-2016, albeit causing groundwater depletion.

## 8 Discussion

WaterGAP has been used in a broad field of applications. To evaluate recent usage of WaterGAP model output for research, we assessed the publications that cite the paper describing WaterGAP2.2, Müller Schmied et al. (2014), hereafter referred to as MS2014. In https://webofknowledge.com, 130 citations were found until 08.04.2020. Of course, other WaterGAP studies (as e.g. Alcamo et al. (1998); Döll et al. (2003); Müller Schmied et al. (2016a); Döll et al. (2014)) were also cited numerous times since the publication of MS2014, but we assume that the assessment based on the citations of this paper can provide a representative overview of WaterGAP usage.

Topic-wise, MS2014 was cited in the scope of climate change impact assessments (18), Life Cycle Analyses (14), TWSA applications, mostly in combination with GRACE (12), model evaluation (11), model development and calibration (10), groundwater stress, depletion and storage change (8), (model) reviews (8), data assimilation (7), water scarcity/stress (7) and water use (5). Other application fields with more than one citation are sea-level rise, water-energy-food nexus, economy, geodesy methodology, drought, ecology / environmental flows, floods, commentary / editorials and root zone-specific data sets. These




**Table 6.** Global-scale (excluding Antarctica and Greenland) water balance components for different time spans as simulated with Water-GAP 2.2d. All units in $\mathrm{km^3\,yr^{-1}}$. Long-term average volume balance error is calculated as the difference of component 1 and the sum of components 2,3 and 7.

| No. | Component | 1961-1990 | 1971-2000 | 1981-2010 | 1991-2016 | 2001-2016 |
|---|---|---|---|---|---|---|
| 1 | Precipitation | 111388 | 111582 | 111616 | 112052 | 112559 |
| 2 | Actual evapotranspiration[1] | 70734 | 71604 | 71979 | 72225 | 72328 |
| 3 | Streamflow into oceans and inland sinks | 40659 | 40009 | 39678 | 39930 | 40357 |
| 4 | Actual consumptive water use[2] | 906 | 1023 | 1146 | 1238 | 1302 |
| 5 | Actual net abstraction from surface water | 1002 | 1108 | 1220 | 1304 | 1353 |
| 6 | Actual net abstraction from groundwater | -96 | -85 | -74 | -66 | -50 |
| 7 | Change of total water storage | -6 | -31 | -40 | -104 | -125 |
| 8 | Long-term average volume balance error | 0.34 | 0.23 | 0.11 | 0.03 | 0.01 |

[1] including actual consumptive water use

[2] sum of rows 5 and 6

**Table 7.** Globally aggregated (excluding Antarctica and Greenland) water storage component changes during different time periods as simulated by WaterGAP 2.2d. All units in $\mathrm{km^3\,yr^{-1}}$.

| No. | Component | 1961-1990 | 1971-2000 | 1981-2010 | 1991-2016 | 2001-2016 |
|---|---|---|---|---|---|---|
| 1 | Canopy | 0.0 | 0.0 | 0.1 | 0.0 | 0.0 |
| 2 | Snow | 16.6 | -6.3 | 3.7 | -12.6 | 5.0 |
| 3 | Soil | -9.4 | -2.2 | 16 | 14.5 | 17.3 |
| 4 | Groundwater | -62.9 | -62.7 | -90.8 | -108.8 | -138.2 |
| 5 | Local lakes | -1.1 | -0.8 | 2.8 | -0.3 | -1.9 |
| 6 | Local wetlands | -1.4 | -3.0 | 3.5 | 0.0 | 4.0 |
| 7 | Global lakes | -4.3 | -5.2 | -0.4 | 4.0 | 9.9 |
| 8 | Global wetlands | -5.8 | 2.4 | 0.2 | 0.1 | -10.3 |
| 9 | Reservoirs and regulated lakes | 68.2 | 43.6 | 28.1 | 5.7 | -3.6 |
| 10 | River | -5.6 | 3.3 | -3.2 | -6.4 | -7.7 |
| 11 | Total water storage | -5.8 | -31.0 | -40.0 | -103.9 | -125.3 |

usages fit well into the motivation of WaterGAP development as highlighted in Alcamo et al. (1998) and Döll et al. (2003), especially as water use and water availability are studied in both historical and future scenario perspectives.

The spatial coverage of the citing literature has been global in most cases (66), followed by multiple basins (19), single
(large) basins (17), single countries (14) and single continents (9). The high amount of global-scale usage indicates the demand





**Table 8.** Globally aggregated (excluding Antarctica and Greenland) sectoral potential withdrawal water use WU and consumptive water use CU ($km^3\,yr^{-1}$) as well as use fractions from groundwater (%) as simulated by GWSWUSE of WaterGAP 2.2d for the time period 1991-2016. These values represent demands for water that cannot be completely satisfied in WGHM due to lack of surface water resources (row 5 in Table 6)

| Water use sector | WU | Percent of WU from groundwater | CU | Percent of CU from groundwater |
|---|---|---|---|---|
| Irrigation | 2363 | 25 | 1100 | 37 |
| Thermal power plants | 599 | 0 | 16 | 0 |
| Domestic | 348 | 36 | 56 | 35 |
| Manufacturing | 272 | 27 | 53 | 26 |
| Livestock | 29 | 0 | 29 | 0 |
| Total | 3610 | 22 | 1253 | 36 |

of spatially consistent and ubiquitously available model output for assessment purposes and model evaluation. The relatively high subglobal-scale usage indicates that, for many regions of the globe, the global WaterGAP model is considered to be a very important source of data.

While 35 out of 130 citing publications only used methods and assessments of MS2014, the others directly used Water-
GAP output data. Usage of water storage output (either total or single/multiple components) was dominant (35), followed by streamflow and runoff (31), and water use (25). In particular, the GRACE satellite mission boosted the evaluation of Water-GAP water storage estimates and allowed for novel ways of data integration and model output evaluation. The high share of studies incorporating streamflow and runoff indicates the importance of these variables as they are the basis for multiple climate change impact assessment and evaluation studies. Most likely, the basin-specific calibration, which results in a rela-
tively high model performance as compared to other GHMs, increases the value of runoff and streamflow output. Within the Life Cycle Assessment community, water use and availability estimates of WaterGAP have been used frequently. In five studies, groundwater-related output and, in four cases, multiple model outputs were applied. Single studies analyzed WaterGAP evapotranspiration and radiation.

Even though MS2014 describes the WaterGAP 2.2 model (with a 0.5° × 0.5° spatial resolution), seven studies refer to
this paper even though WaterGAP 3 model output (with 5 × 5 arc-min spatial resolution) was studied. The hydrological process representations are similar in both model version families, however the technical settings are different. 21 studies refer to MS2014 in relation to ISIMIP (www.isimip.org), which highlights the contribution of WaterGAP to this societally and scientifically relevant initiative.



## 9 Conclusions and outlook

A globally consistent quantification of water flows and storages as well as of human water use is needed but challenging, not only due to a lack of observation data but also the difficulty of appropriate process representation in necessarily coarse grid cells (Döll et al., 2016). This study fully describes the state-of-the-art GHM WaterGAP in its newest version 2.2d. Evaluation of model performance using independent data or observations of the key output variables, namely withdrawal water uses, streamflow and total water storage indicates a reasonable model performance and points to potential areas of model improvement.

Model output has been widely used for studying diverse research problems but also for informing the public about the state of the global freshwater system. The description of model algorithms, model outputs and related caveats will allow for better usage of model outputs by other researchers, who can now access these data from the PANGAEA repository.

Ongoing WaterGAP development aims to fully integrate a gradient-based groundwater model (Reinecke et al., 2019), improve the floodplain dynamics of large river basins (e.g. the Amazon) as proposed by Adam (2017) and to integrate glacier

mass data (Cáceres et al., 2020). In addition, an update of the data basis for water use computations is planned. To enhance cross-sectoral integration in the framework of ISIMIP, modelling of river water temperature according to Van Beek et al. (2012) and Wanders et al. (2019) will be implemented.

*Code and data availability.* WaterGAP 2.2d is on the way to open source but still in the process of clarifying licensing and copyright issues. Hence, source code cannot be made publicly available but will be available for referees and editors. The model output data availability is

described in Sect. 5. For latest papers published based on WaterGAP 2, we refer to http://www.watergap.de, last access: 25 March 2020.

*Author contributions.* HMS and PD led the development of WaterGAP2.2d. HMS led the software development, supported by DC, CH, CN, TAP, EP, FTP, RR, SS, TT, PD. The paper was conceptualized by HMS and PD. HMS did the calibrations, simulations, data analysis, visualization and model validation, supported by MS regarding validation against GRACE TWS. CN prepared model output for the PANGAEA data repository. The original draft was written by HMS, with specific parts drafted and reviewed by all authors. All authors contributed to the

final draft.

*Competing interests.* The authors declare that they have no conflict of interest.

*Acknowledgements.* We thank Tim Schön for generating Fig. 3 and data processing for Fig. 5, Hans-Peter Ruhlhof-Döll for processing and generating Fig. 14. We furthermore thank Florian Herz for polishing the reference list and for technical support during manuscript preparation. We are grateful for Edwin Sutanudjaja for providing insights to the withdrawal water use comparison of PCR-GLOBWB. We

acknowledge the evaluation data sets from GRDC (The Global Runoff Data Centre, 56068 Koblenz, Germany), AQUASTAT and GRACE





(CSR RL05 GRACE mascon solutions downloaded from http://www2.csr.utexas.edu/grace and JPL GRACE mascon data are available from http://grace.jpl.nasa.gov, supported by the NASA MEaSUREs Program).

**Appendix A: Description of changes between the model versions 2.2 and 2.2d**

**A1 Modifications of water use models compared to WaterGAP2.2**

- Deficit irrigation with 70% of optimal (standard) consumptive irrigation water use was applied in grid cells, which were selected based on Döll et al. (2014) and have 1) groundwater depletion of $> 5\,\mathrm{mm\,yr^{-1}}$ over 1989–2009 and 2) a >5% fraction of mean annual irrigation withdrawal water uses in total withdrawal water uses over 1989–2009 (Sect. 3.3). In WaterGAP 2.2, optimal irrigation allowing the plants to evapotranspirate at 100% of PET was assumed to be done everywhere.

- Integration of time series of Historical Irrigation Dataset HID for 1900 to 2005 (Siebert et al., 2015) into the global irrigation model GIM (Sect. 3.1) (Portmann, 2017). In WaterGAP 2.2, irrigated areas of the static Global Map of Irrigation Areas GMIA (Siebert et al., 2005) were scaled by time series of irrigated area per country. In addition to that the newly available country-specific Area Actually Irrigated (AAI) which is available for 47 countries were used to update computed ICU until 2010. Version 2.2d enables to consider cells-specific AAI/AEI-ratio (for details see Portmann (2017)).

- Non-irrigation water uses (domestic, manufacturing) were corrected to plausible values for coastal cells with small continental areas to avoid unrealistically high total water storage values in those cells.

**A2 Modifications of WGHM compared to WaterGAP 2.2**

General

- With the introduction of dynamic extents of surface water bodies, land area fractions became variable in time as well (Sect. 2.2).

- Modified routing approach where water is routed through the storages depends upon the fraction of surface water bodies; otherwise water is routed directly into the river (Sect. 4) (Döll et al., 2014) .

- Since WaterGAP 2.2b, net cell runoff $R_{nc}$ is the difference between the outflow of a cell and inflow from upstream cells at the end of a time step (Sect. 4.7.3). In the versions before, cell runoff was defined as outflow minus inflow into the river storage.

- Modified calibration routine: an uncertainty of 10% of long-term average river discharge is allowed (following Coxon et al. (2015)), meaning that calibration runs in four steps as described in Sect. 4.9.1.





- Since WaterGAP 2.2b, all model parameters which are potentially used for the calibration/data assimilation integration (including also parameter multiplicators) are read from a text file in Javascript Object Notation (JSON) format.

- The differentiation into semi-arid/humid grid cells are defined with a new standard methodology (Sect. B).

- For WaterGAP 2.2d, the return flows from surface water resources are scaled according to actual $NA_s$ (see results in Sect. 7 and Fig. 15). Return flows induced by irrigation from surface water resources were calculated in WaterGAP 2.2 under the assumption that $NA_s$ can be fully satisfied. However, this can lead to implausible negative total actual consumptive water use, if surface water availability leads to smaller actual $NA_s$ than the return flows.

- Implementation of a new storage-based river velocity algorithm (Sect. 4.7.1).

- The realisation of naturalized runs was improved. In WaterGAP2.2, reservoirs were treated like global lakes in naturalized runs, while now, global reservoirs are completely removed (but local reservoirs are still handled as local lakes) (Sect. 4.1). Please note that in the studies of (Döll et al., 2009; Döll and Zhang, 2010) performed with even older model version, all reservoirs were removed in naturalized runs.

Soil

- The total water capacity input was newly derived and is now based on Batjes (2012) (Müller Schmied, 2017) (Sect. 4.4.3) whereas in WaterGAP 2.2 it was based on Batjes (1996).

Groundwater

- Groundwater recharge below surface water bodies (LResW) is implemented in semi-arid and arid regions of Döll et al. (2014) in WaterGAP 2.2d.

- Regional changes since WaterGAP 2.2b based on Döll et al. (2014): 1) for Mississippi Embayment Regional Aquifer, groundwater recharge was overestimated, and thus the fraction of runoff from land recharging groundwater was reduced from 80–90% to 10% in these cells by adapting the groundwater factor $f_g$ (Fig. D1); 2) groundwater depletion in the North China Plain was overestimated by a factor of 4, and thus runoff coefficient $\gamma$ was reduced from 3–5 to 0.1 in this area (Fig. D2); 3) all wetlands in Bangladesh were removed since diffuse groundwater recharge was unrealistically low.

- In WaterGAP 2.2d and for semi-arid/arid grid cells: In case of less precipitation than $12.5\ \mathrm{mm\,day^{-1}}$, groundwater recharge remains in the soil column, and not handled as runoff anymore as in the versions before (Sect. 4.4.3).

LResW

- Precipitation on surface water bodies is now also multiplied with the evaporation reduction factor (like evaporation) to keep water balance consistent (Sect. 4.6.3).

- Update of reservoir information, including year when reservoir began operation (commissioning year, Sect. 4.6.3) (Müller Schmied et al., 2016a; Müller Schmied, 2017).





- Implementing reservoir commissioning years to reservoir algorithm (Sect. 4.6.3) (Müller Schmied et al., 2016a; Müller Schmied, 2017); before this year, the reservoir is not present and in case of a regulated lake it is simulated as global lake. In the versions before 2.2d, reservoirs and regulated lakes are simulated to be always present.

- For global lakes and reservoirs (where the water balance is calculated in the outflow cell), water demand of all riparian cells is included in the water balance of the outflow cell and thus can be satisfied by global lake or reservoir storage (Sect. 4.6.3).

- All water storage equations in horizontal water balance are solved analytically in WaterGAP 2.2d (except for local lakes). Those equations now include net abstractions from surface water or groundwater. As a consequence, sequence of net abstractions has been changed to 1) global lakes, regulated lakes or reservoirs, 2) rivers, 3) local lakes (Sect. 4.6.3).

- Areal correction factor (CFA) is included in water balance of lakes and wetlands in WaterGAP 2.2d (Sect. 4.6.3).

- In WaterGAP 2.2d (as in versions before WaterGAP 2.2), local and global lake storage can drop to $-S_{max}$ as described in Hunger and Döll (2008). The area reduction factor (corresponding to the evaporation reduction factor in Hunger and Döll (2008) (their eq. 1) has been changed accordingly (denominator: 2 x $S_{max}$). If lake storage $S$ equals $S_{max}$, the reduction factor is 1; if $S$ equals $-S_{max}$, the reduction factor is 0 (Sect. 4.6.3)

- Active reservoir storage is not anymore assumed to be 85% but 100% of reported storage (based on comparisons with literature) (Sect. 4.6.3)

**Appendix B: Definition of arid and humid grid cells**

The definition of semi-arid and arid grid cells is the basis for e.g., fractional routing (Sect. 4.5.3), groundwater recharge scheme (Sect. 4.4.3, 4.6.3) and for PET equation (Sect. 4.2.3). In the model versions before WaterGAP 2.2c as used in Müller Schmied et al. (2016a), we defined the input file for semi-arid/arid or humid grid cells according to the climate forcing used. However, it turned out that this leads to problems when comparing model outputs from different model versions and climate forcings. For example, if well-known non-humid regions (e.g., the High Plains Aquifer and the North China Plain) are classified as humid to a large extent due to uncertain climate forcing (and the approach used), this is not representing reality and can lead to implausible calculation of hydrological processes in those regions. Therefore, a static definition of semi-arid/arid and humid grid cells was developed (Fig. B1).

Following Shuttleworth (1993), the Priestley-Taylor $\alpha$ is set to a value of 1.26 for humid regions and of 1.74 for semi-arid/arid regions. WaterGAP 2.2c was run with EWEMBI (Lange, 2019) for 1981-2010 with all grid cells defined as humid to avoid pre-definition of areas with high or low PET due to initial setup of the $\alpha$. Following Middleton and Thomas (1997), drylands were defined based on aridity index ($AI = P/PET$) with ($AI < 0.65$) and non-drylands with $AI \geq 0.65$. Due to the definition of $\alpha$ to a humid value globally, PET might be too low, especially for transitional zones between drylands and



non-drylands. Therefore, and based on visual inspection, we defined all grid cells with ($AI < 0.75$) as semi-arid/arid grid cells.
Furthermore, we defined all grid cells north of 55°N as humid grid cells.

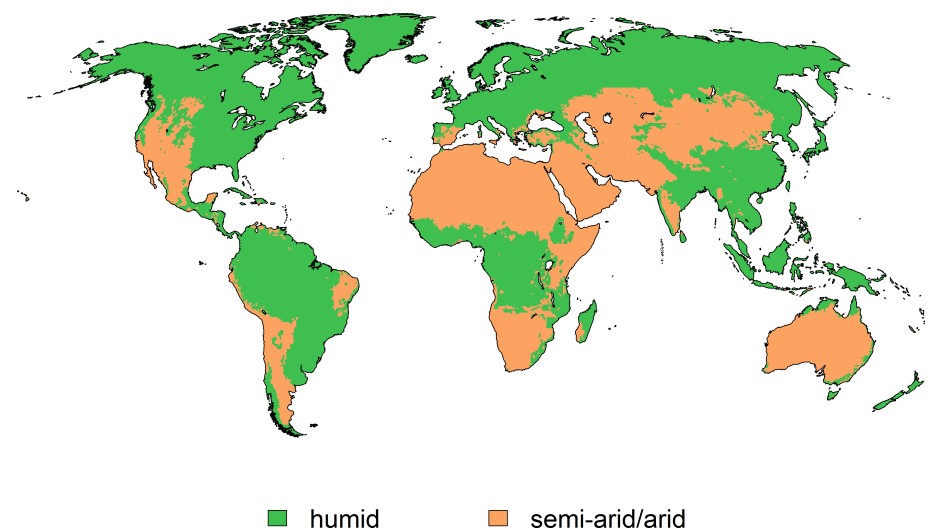

**Figure B1.** Static definition of humid and semi-arid/arid grid cells.

## Appendix C: Land cover input

WGHM is using a static land cover input map (Fig C1) which is derived from Moderate Resolution Imaging Spectroradiometer (MODIS, MODIS, Friedl et al. (2010)) data for the year 2004 (Dörr, 2015). The primary land cover attribute at the original resolution of 500 m is used as basis. In case a 500 m MODIS primary land cover is defined as "urban area", "permanent
wetland" or "water body", the secondary land cover was used instead as those land cover types are included as separate input (for lakes/wetlands the GLWD dataset, Sect. 4.6, urban areas are implemented as impervious areas, Sect. 4.4.3). Finally, the dominant IGBP land cover type (primary land cover) was selected for each $0.5° \times 0.5°$ grid cell.

## Appendix D: Additional figures

This section consists of additional figures, which might help to understand specific contents of the main text. It consists of
1135 regional modification of model parameters and further performance assessments.

## Appendix E: Integration of GLWD and GRanD data of lakes, reservoirs and wetlands (LResW) into WGHM

WGHM uses the Global Lakes and Wetland Database (GLWD) (Lehner and Döll, 2004) and a preliminary but updated version of the Global Reservoir and Dam (GRanD) database (Lehner et al., 2011) to define location, area and other attributes of

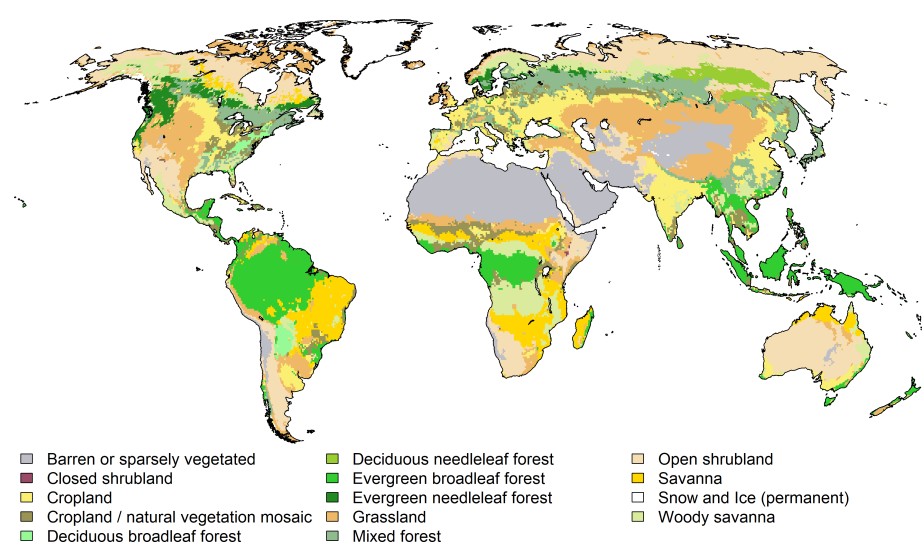

**Figure C1.** Land cover classification of WaterGAP 2.2d.

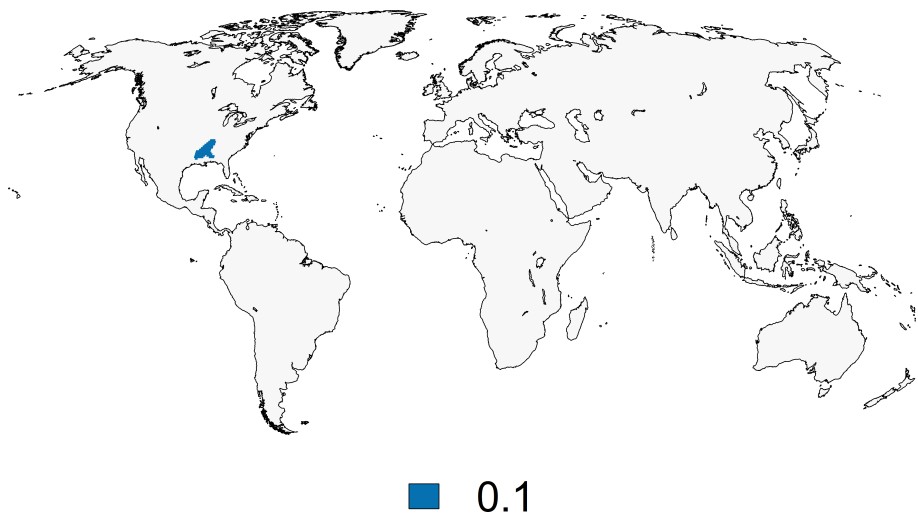

**Figure D1.** Regional correction of the groundwater factor $f_g$ to allow more realistic groundwater recharge rates.

LResW. The GLWD database consists of three data sets. GLWD-1 contains shoreline polygons of 3067 large lakes (area is >=
50 km$^2$) and 645 large reservoirs (capacity >= 0.5 km$^3$), GLWD-2 contains shoreline polygons of approximately 2,500,000 smaller lakes, reservoirs, and rivers and GLWD-3 is a 30 arc-sec raster data set with lakes, reservoirs, rivers and wetland types, including both GLWD-1 and GLWD-2 water bodies. The GRanD v1.1 database includes 6,824 reservoir polygons (Lehner et al., 2011). Information from these databases was translated to the six categories of LResW implemented in WaterGAP and




**Table C1.** Parameters of the leaf area index model from Müller Schmied et al. (2014).

| No | Land cover type | $L_{max}$ | Fraction of deciduous plants $f_d$ | L reduction factor for evergreen plants $C_e$ | Initial days to start/end with growing season (d) |
|---|---|---|---|---|---|
| 1 | Evergreen needleleaf forest | $4.02^a$ | 0 | 1 | 1 |
| 2 | Evergreen broadleaf forest | $4.78^b$ | 0 | 0.8 | 1 |
| 3 | Deciduous needleleaf forest | 4.63 | 1 | 0.8 | 10 |
| 4 | Deciduous broadleaf forest | $4.49^c$ | 1 | 0.8 | 10 |
| 5 | Mixed forest | $4.34^d$ | 0.25 | 0.8 | 10 |
| 6 | Closed shrubland | 2.08 | 0.5 | 0.8 | 10 |
| 7 | Open shrubland | 1.88 | 0.5 | 0.8 | 10 |
| 8 | Woody savanna | 2.08 | 0.5 | 0.3 | 10 |
| 9 | Savanna | 1.71 | 0.5 | 0.5 | 10 |
| 10 | Grassland | 1.71 | 0 | 0.5 | 10 |
| 11 | Cropland | 3.62 | 0 | 0.1 | 10 |
| 12 | Cropland/natural vegetation mosaic | 3.62 | 0.5 | 0.5 | 10 |
| 13 | Snow and ice | 0 | 0 | 0 | 0 |
| 14 | Bare ground | 1.31 | 0 | 1 | 10 |

[a] $L_{max}$ is assumed to be the mean value of TeENL and BoENL land cover classes of Scurlock et al. (2001), ; [b] only value for TrEBL and not TeEBL from Scurlock et al. (2001) as in WaterGAP this class is mainly in the tropics; [c] mean value from TeDBL and TrDBL from Scurlock et al. (2001); [d] mean value of all forest classes. Fraction of deciduous plants and L reduction factor for evergreen plants based on IMAGE (Alcamo et al. (1998)) initial days to start/end with growing season are estimated.

assigned to the 0.5° × 0.5° grid cells (see Table. E1). Fig. E1 shows the spatial distribution of the maximum extent of all LResW (all six categories) in terms of fractional coverage.

- Implementation of wetlands

GLWD-3 provides approximately the temporal maximum of wetland extent as wetland outlines were mainly derived from maps and are used to determine $A_{max}$. In case of various input data sets, a wetland was assumed to be present if at least one of the data sets showed one. The wetland types "coastal wetland" (covering 660,000 km$^2$) and "intermittent wetland/lake" (690,000 km$^2$) which are in GLWD-3 are not included in WGHM. Inclusion of coastal wetlands would require the simulation of ocean-land interaction, while intermittent wetlands/lakes of GLWD-3 cover very large parts of the deserts (comp. Fig. 5 in Lehner and Döll (2004)) that cannot be assumed to be covered totally by water at any time but rather represent areas where very rarely and at different points in time some parts may be flooded. Rivers shown in





**Table C2.** Attributes for IGBP land cover classes used in WaterGAP2.2d from Müller Schmied et al. (2014). Water has an albedo of 0.08, snow 0.6.

| No | Land cover type | Rooting depth[a] (m) | Albedo[a] (-) | Snow albedo (-) | Emissivity[b] (-) | Degree-day factor $D_F$[c] (mm d$^{-1}$ °C$^{-1}$) |
|---|---|---|---|---|---|---|
| 1 | Evergreen needleleaf forest | 2 | 0.11 | 0.278 | 0.9956 | 1.5 |
| 2 | Evergreen broadleaf forest | 4 | 0.07 | 0.3 | 0.9956 | 3 |
| 3 | Deciduous needleleaf forest | 2 | 0.13 | 0.406 | 0.99 | 1.5 |
| 4 | Deciduous broadleaf forest | 2 | 0.13 | 0.558 | 0.99 | 3 |
| 5 | Mixed forest | 2 | 0.12 | 0.406 | 0.9928 | 2 |
| 6 | Closed shrubland | 1 | 0.13 | 0.7 | 0.9837 | 3 |
| 7 | Open shrubland | 0.5 | 0.2 | 0.7 | 0.9541 | 4 |
| 8 | Woody savanna | 1.5 | 0.2 | 0.558 | 0.9932 | 4 |
| 9 | Savanna | 1.5 | 0.3 | 0.7 | 0.9932 | 4 |
| 10 | Grassland | 1 | 0.25 | 0.7 | 0.9932 | 5 |
| 11 | Cropland | 1 | 0.23 | 0.376 | 0.9813 | 4 |
| 12 | Cropland/natural vegetation mosaic | 1 | 0.18 | 0.3 | 0.983 | 4 |
| 13 | Snow and ice | 1 | 0.6 | 0.7 | 0.9999 | 6 |
| 14 | Bare ground | 0.1 | 0.35 | 0.7 | 0.9412 | 6 |

[a] Adapted from the IMAGE model (Alcamo et al., 1998); [b] Wilber et al. (1999); [c] Maniak (1997), WMO (2009).

GLWD-3 are considered to be (lotic) wetlands and included as wetlands in WGHM. It is assumed that only a river with adjacent wetlands (floodplain) is wide enough to appear as a polygon on the coarse-scale source maps (Lehner and Döll, 2004). For the fractional wetland type "50-100% wetland", an arbitrary value of 75% grid cell coverage with wetland is assumed, for "25-50% wetland" a value of 35% and for "wetland complex" a value of 15%. The large floodplain wetland of the lower Ganges-Brahmaputra in GLWD-3, covering almost all of Bangladesh, is not simulated as a wetland in WGHM, as during most of the time, only a small part of Bangladesh is inundated.

All wetlands subsumed in fractional classes are assumed to be local, i.e. locally-fed. In case of all other wetland types, global wetlands fed by the whole catchment were identified as follows. All wetland polygons with a direct connection to a major river (as defined by the big_river.shp file available from ESRI) are assumed to receive inflow from a large



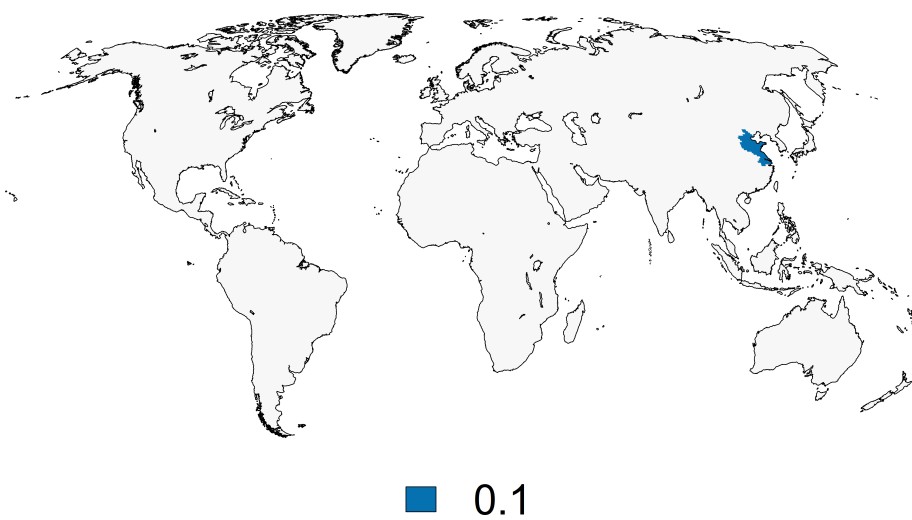

**Figure D2.** Regional correction of calibration parameter $\gamma$ to allow more realistic groundwater recharge rates.

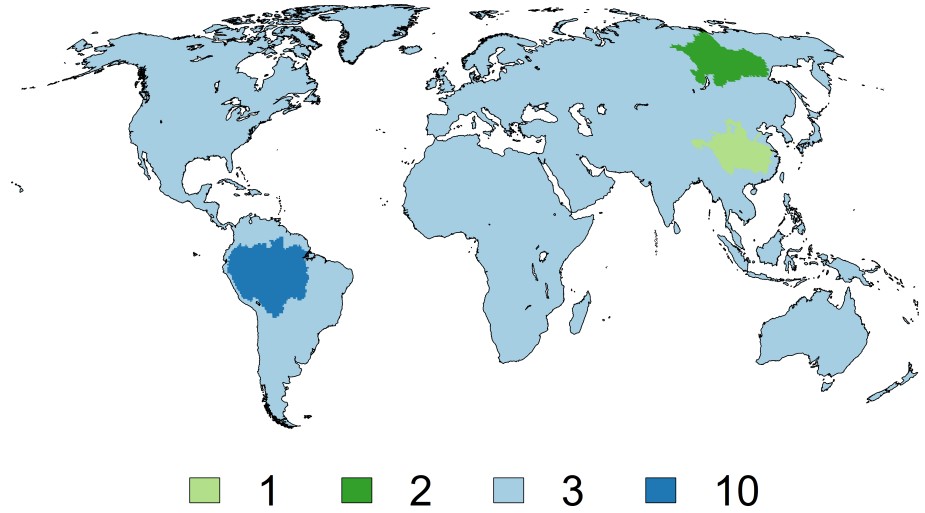

**Figure D3.** Region-specific multiplier for river roughness.

upstream area and are therefore categorized as global. However, if rivers in this file are categorized as intermittent, the adjacent wetlands are categorized as local in WGHM. All other wetlands are first buffered (to the inside, using a GIS) by a 10 km wide ring such that the outer 10 km of a wetland are considered to be local and the core wetland area inside this buffer ring is considered to be global.

- Implementation of lakes, man-made reservoirs and regulated lakes

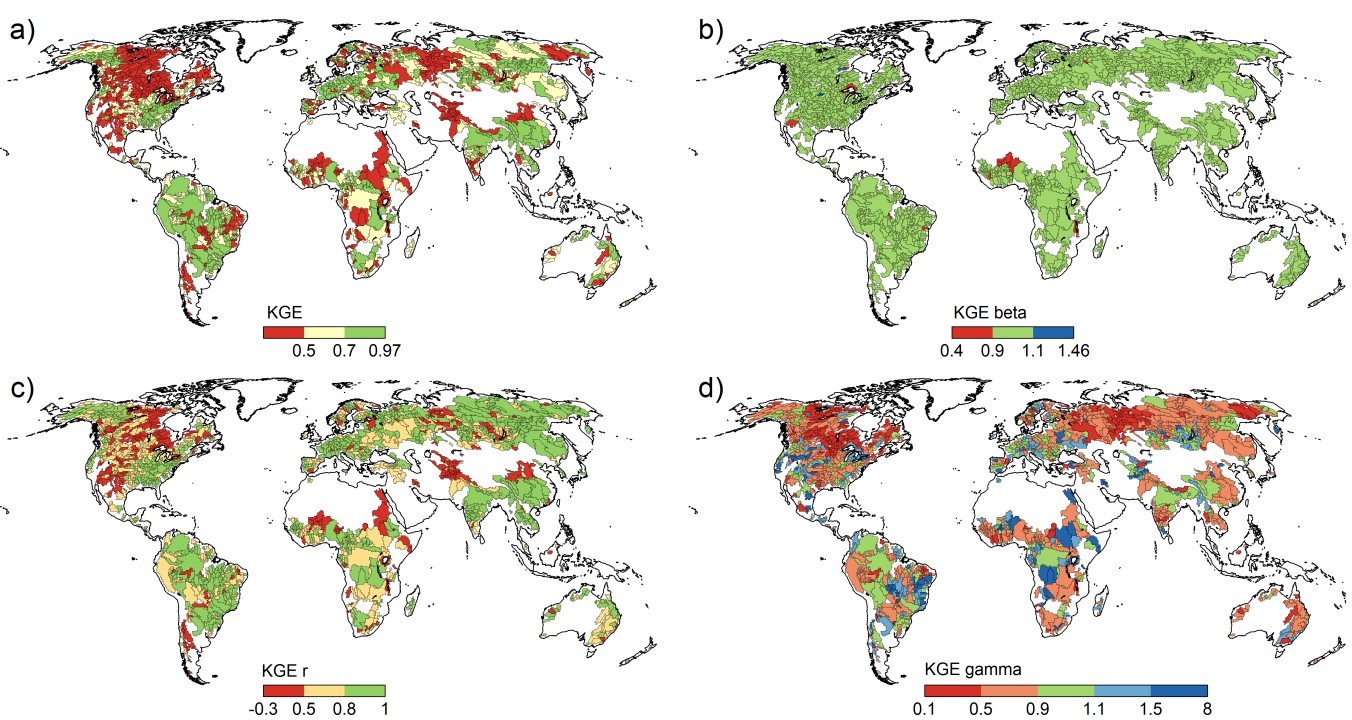

**Figure D4.** KGE and its components range at 1319 river basins for WaterGAP 2.2

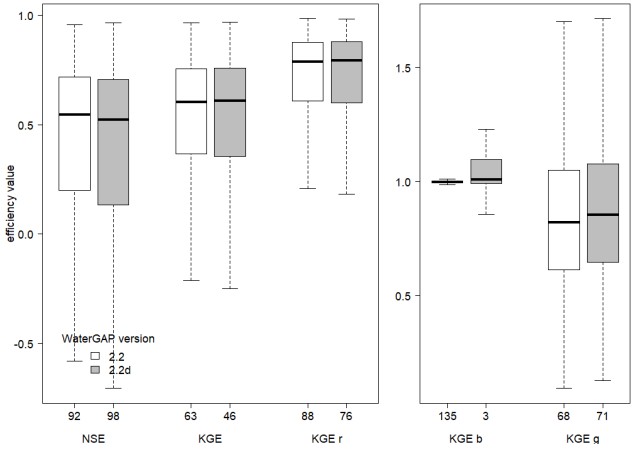

**Figure D5.** Efficiency of streamflow for the 1319 river basins in comparison of model versions WaterGAP 2.2d and WaterGAP 2.2 showing similar model performance. Outliers are excluded but number of outliers indicated at x axis.

The $0.5° \times 0.5°$ outflow cell of each global lake is determined based on the GLWD lake polygon and the DDM30 drainage direction map. If more than one global lake has the same outflow cell, the lakes are treated as one lake by adding the lake areas. The same procedure is done in case of reservoirs/regulated lakes. There are 43 grid cells with 2


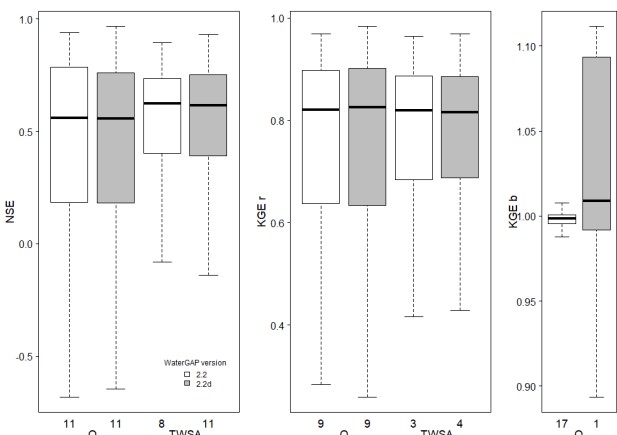

**Figure D6.** Efficiency of streamflow and TWSA for the river basins larger than 200,000 km$^2$ in comparison of model versions WaterGAP 2.2d and WaterGAP 2.2 showing similar model performance. Outliers are excluded but number of outliers indicated at x axis.

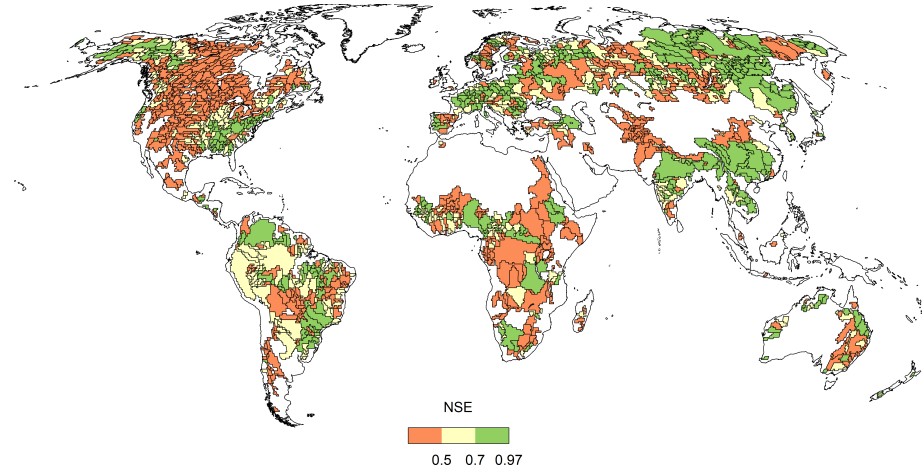

**Figure D7.** Classified $NSE$ efficiency metric represented for the 1319 river basins and WaterGAP 2.2.

reservoirs, 6 grid cells with 3 reservoirs, 2 grid cells with 1 regulated lake and 1 reservoir, 1 grid cell with 2 regulated lakes and 1 grid cell with 1 global lake and 1 regulated lake. Each cell can be the outflow cell of both a global lake and a global reservoir/regulated lake but if there is a regulated lake and a reservoir in one outflow cell, then they are aggregated. The commissioning year and main purpose of the larger reservoir/regulated lake is used. The commissioning year of 1175 the resulting 1109 reservoirs/regulated lakes that are simulated as individual reservoirs/regulated lakes was obtained mainly from the GRanD database but also other sources. In the commissioning year, the reservoir area is increases to its full extent (thus land area fraction is adjusted), the reservoir starts filling and reservoir algorithm is enabled. The





**Table E1.** LResW representation in WGHM. The total continental area represented in WaterGAP is 136.782 million $km^2$ (Antarctica is not included in WaterGAP) and 134.396 million $km^2$ without Greenland. The minimum land area (without Greenland), i.e. continental area minus maximum LResW area, is 124.449 million $km^2$

| No | Surface water body type | Data source | Area description | Maximum global area [million $km^2$] | Definition |
|---|---|---|---|---|---|
| 1 | Local wetland | GLWD-3 | % of cell area | 3.743 | Wetland types 10, 11, 12, part of wetland types 4, 5, 7, and 8 of GLWD-3 (see description in E)*. |
| 2 | Global wetland | GLWD-3 | % of cell area | 3.752 | Part of wetland types 4, 5, 7 and 8*. |
| 3 | Local lake | GLWD-1, GLWD-2 | % of cell area | 0.850 | Lakes with area < 100 $km^2$ and reservoirs where a maximum storage capacity < 0.5 $km^3$. |
| 4 | Global lake | GLWD 1 | % of cell area, total area of water body | 1.010 | Lakes with area >= 100 $km^2$ |
| 5 | Global reservoir | GRanD | % of cell area, total area of water body | 0.404 | Man-made reservoirs with a maximum storage capacity >= 0.5 $km^3$. |
| 6 | Global regulated lake | GRanD | % of cell area, total area of water body | 0.188 | Global lakes that are regulated and simulated like global reservoirs. Maximum storage capacity provided by GRanD is only the additional storage due to dam construction. |

[*] wetland categories of GLWD-3: 4-freshwater marsh, floodplain, 5- swamp forest, flooded forest, 7- pan, brackish/saline wetland, 8- bog, fen, mire, 10- 50-100% wetland (using 75% of area as local wetland), 11- 25-50% wetland (using 35% of area as local wetland), 12- wetland complex (0-25% wetland) (using 15% of area as local wetland)

storage capacity of the reservoirs which are in operation in the model initialization year is set to the maximum value (Müller Schmied, 2017).



**Figure E1.** figure single fractions and sum grid cell area covered (maximum extent) and land fraction (minimum)



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
