# Peer review of "The global water resources and use model WaterGAP v2.2d: Model description and evaluation"

_Geoscientific Model Development, 2020_

## Referee Comment (RC1) · Anonymous Referee #1 · 10 Sep 2020

General comments

The authors provide a detailed model description of the latest WaterGAP global hydrological model and specification and validation of its standard output data. The model description part covers the entire model but puts extra weights on the improvements and advances since Müller-Schmied et al. (2014) which reported the last model updates. The standard output data part concisely compiles related information for the potential users.

WaterGAP is a great model that has lead the field of global water resources research for two decades. I believe this paper provides a foundation and a benchmark to the

research community.

It is noteworthy that the model description includes the detailed procedure of hydrological parameter tuning. One of the distinct characteristics of WaterGAP is that the developers have conducted painstaking manual hydrological parameter tuning at more than one thousand basins. This feature brings a distinct performance compared to other global hydrological models (mostly untuned), but the procedure was virtually unseeable for non-developers since available descriptions were quite old (Döll et al., 2003; Hunger and Döll, 2008). The clear and detailed description in this paper will be helpful for understanding the outputs of WaterGAP, in particular, those who are interested in intercomparing models.

The disclosure of standard output should be highly appreciated. Although numerous model intercomparison projects have been conducted (e.g. WaterMIP, ISIMIP/global water), the performance of models tends to fall short of that of under the original (model-optimum) condition. The broad community will be benefited from the provided data.

The manuscript is certainly long, but well structured and written. The major contents are, as noted earlier, the description of the latest model and outputs which is essentially a summary of past six years of peer reviewed papers. Due to the nature of this manuscript, I haven't rigorously examined the methods and results one by one. Rather I have read and commented this manuscript from the viewpoint of a learner of the model and a user of the outputs. Hope the specific comments below are useful for further revision.

Specific comments

Line 11: This sentence is too long. Better to split into two or three.

Line 20 'Environmental Performance Index': This term needs a reference.

Line 86: 'Hyungjun' reads 'Kim'.

Line 137 'Cropping patterns and growing periods are generated for every year': A bit confusing. The authors wrote that growing period is fixed at 150 days. What does this part mean?

Line 138 'the respective 30-year climate averages': A bit hard to read. Which climate variables are year-specific and which are 30-year mean?

Line 265 'Increases in soil water storage in irrigated areas are not taken into account': I am wondering how evapotranspiration from irrigated area is estimated in this model. I guess abstracted water for irrigation is directly added to evapotranspiration of a grid cell, but this should be clearly elaborated.

Line 318 Equation 7: Seems LAI was used only for canopy storage calculation. Is this really the case? I wish to see the list of variables which are directly affected by the daily dynamics of LAI. This point must be important to understand/interpret the outputs of WaterGAP model.

Line 611 'Unsatisfied water use is added to NAs of the next day until the end of the calendar year': It sounds that this treatment can result in a quite unnatural hydrograph. For example, for the rivers affected by the monsoon system, the increase in wet season's discharge must be substantially delayed, because the initial increase in runoff is used for the 'repayment of water loan' accumulated in the preceding dry season. Please consider adding a note on the consequences of this assumption/treatment which would be helpful for the readers. Finally, as a hydrologist living in an Asian country, I need to write here that this assumption/treatment is quite odd. The drastic seasonal change in water availability is the heart of the water scarcity problem in our region, which seems largely (if not completely) unaccounted by this model (see discussion in Hanasaki et al. 2008, HESS).

Line 629 'areal correction factor (CFA)': Why is this term called 'areal'?

Line 647 'For global water balance assessment the mass balance is kept by adjusting

the actual evapotranspiration component': Does this mean that the actual evapotranspiration is simply calculated by P – Q?

Line 779 "NSE and logarithmic NSE": NSE is usually calculated between two time-series at a single location (e.g monthly simulated and observed discharge). How NSE in Figure 5 was calculated? Seems nation-wise NSE was calculated using five-year interval time series (i.e. the typical interval of FAO AQUASTAT is five-year), then averaged globally, but is this really the case?

Line 785 "However, NSE Values below 0 for 259 stations show the complete failure of WaterGAP2.2d to simulate streamflow dynamics in one fifth of the evaluated basins": I feel that this sentence is a bit unclear. What I understood is that monthly and annual variations were not properly reproduced for these 259 stations, although the simulated mean annual discharge agrees well with that of observation due to the calibration.

Line 775 "reasonable quality": I don't know what this phrase exactly indicates. The log-log scatter plot (Figure 5) is not very helpful for judging model performance. At least some additional notes are needed for the results of industrial sector (Figure 5e) which indicates frequent occurrence of two orders of magnitude overestimation between simulation and observation.

Line 938-945 "In case of negative NAs...": Highly technical and hard to read. It would be helpful for readers if the authors add links to the directly relevant model description parts (e.g. subsection or equation).

Line 949 Table 6: What does negative values for 'actual net abstraction from groundwater' indicate? Does it mean that groundwater recharge has been increased by humanities? I am quite confused because Table 7 indicates that the groundwater is being depleted globally. Similarly, add some extra notes for the negative values for 'change of total water storage' which look constantly increasing by time. What are the key reasons for this?

---

## Referee Comment (RC2) · Gemma Coxon (Referee) · 6 Oct 2020

**Review of "The global water resources and use model WaterGAP v2.2d: Model description and evaluation" by Müller Schmied et al.**

This paper describes the global hydrological model WaterGAPv2.2d and evaluates its outputs. Overall, the paper is well written and a comprehensive model description that will be useful to a wide array of environmental scientists. There is a clear description of the model updates from WaterGAP 2.2 and each of the model components is well described. It is great to see all the model outputs being made available and there is some interesting model evaluation. Overall my comments are relatively minor revisions, however, I have a couple of broader comments on calibration and I do think the paper is too long and that section 7 and 8 need to be significantly shortened or removed.

**General Comments**

*Summary Table.* The description of each of the model components is generally very clear and comprehensive. However, it would be useful to have a summary table of parameters, stores and fluxes for each of the models. See for example Table 1 in Brauer et al (2014) (doi:10.5194/gmd-7-2313-2014). This would be useful for a reader to refer to, provide a central place where a lot of the acronyms in the paper can be found (there are a lot of acronyms!) and be very helpful as a summary when comparing models in model intercomparisons. If it cannot fit in the main paper, this would still be useful in an appendix or supplementary info.

*Calibration.* There are a lot of correction factors applied to the outputs for each basin before a suitable result (defined here as +/-10% of the long term mean annual flow) is obtained. Figure 4 is interesting but the results are not analysed in section 4.9.3 – where do the authors think the major errors are coming from (input data, model process representation) and how does this vary spatially? Given the significant correction factors for a lot of basins, it warrants a discussion on how appropriate the model outputs are for future conditions when you assume that these correction factors remain stationary over time (when in reality they could change depending on what errors they are accounting for!) and the fact that you correct to the mean but do not consider extremes (so how appropriate the model results may be for floods/droughts). It would also be useful to make these correction factors available as part of the standard model outputs (if they are not already).

*Model Application.* Section 7 and 8 do not add to the paper and both need to be significantly shortened or removed. Currently, the paper is very long and while the material before this is very relevant, the results presented in Section 7 do not provide any new information or significant results to the reader. Section 8 could also be significantly shorter. If these sections were shortened then you could expand a little on the interesting discussion of the future model developments that you discuss in the conclusion and perhaps link this in with broader developments in the GHM community. Furthermore, it would make the paper shorter and more readable for readers.

*Code and Data Availability.* I understand the difficulties with making the code open source. Do you have a timeline of when it will be made open source and how it will be made open source (for example on a platform like GitHub?).

*Appendices and Supplementary Information.* It would be worth thinking about whether some of your appendix materials (particularly Appendix D) may be better placed in supplementary material rather than the appendix of the paper.

**Minor Comments**

*Abstract L9.* I would replace 'can be done' with 'can be achieved'

*Introduction L11.* This opening sentence is too long – it needs rewriting.

*Introduction L28.* What do you mean by 'proper simulation'?

*Introduction L43.* It would be good to have some specific references here of where this variant of the model has been used.

*Section 3.2.2 L180.* Can you provide a reference or website for the Environmental Data Explorer?

*Section 4.6 L415.* Can you add the specific version of the GRanD database you are using?

*Section 4.6 L416.* I think 'Sect. E' should be 'Appendix E'?

*Section 4.9.1 L621.* "to avoid that average water resources are misrepresented" – this isn't clear as written, can you be more specific?

*Section 4.9.1 L634.* One of the key outcomes from Coxon et al (2015) was that the discharge uncertainty varied significantly between gauging stations and over the flow range. It may be worth adding a sentence somewhere in the paper stating that you recognise that the discharge uncertainty is unlikely to be stationary in space and time but there are no further data to better constrain the uncertainties at these gauging stations so a representative value of +/-10% is used.

*Section 6.4.1.* Missing 'h' from 'withdrawals' in title

*Figure 5.* I wasn't sure whether the NSE and logNSE values presented in Figure 5 were calculated based on all the monthly country data or were a median value for each country? It is not clear how it was calculated for these variables.

*Section 6.4.2 L785.* "is rather satisfying" – I would remove this and just present the results

*Section 6.5.* Can you attribute some of these improvements in model performance to specific changes made to the model?

*Conclusions.* An additional development here could be improving the lakes/wetlands and reservoir regulation as you noted this being a limiting factor to good model performance in North America/Canada?

---

## Author Comment (AC2) · 24 Nov 2020

**Answer to comments of Gemma Coxon**

The original comments of Referee Gemma Coxon are in black color and indicated by "R:". Replies by the authors ("A") are colored in green. Actions are introduced by "Action:", changes in the manuscript are in italics.

R: This paper describes the global hydrological model WaterGAPv2.2d and evaluates its outputs. Overall, the paper is well written and a comprehensive model description that will be useful to a wide array of environmental scientists. There is a clear description of the model updates from WaterGAP 2.2 and each of the model components is well described. It is great to see all the model outputs being made available and there is some interesting model evaluation. Overall my comments are relatively minor revisions, however, I have a couple of broader comments on calibration and I do think the paper is too long and that section 7and 8 need to be significantly shortened or removed.

A: Thank you for the overall positive feedback and the comments to which we will answer below.

**General Comments**

R: *Summary Table.* The description of each of the model components is generally very clear and comprehensive. However, it would be useful to have a summary table of parameters, stores and fluxes for each of the models. See for example Table 1 in Brauer et al (2014) (doi:10.5194/gmd-7-2313-2014). This would be useful for a reader to refer to, provide a central place where a lot of the acronyms in the paper can be found (there are a lot of acronyms!) and be very helpful as a summary when comparing models in model intercomparisons. If it cannot fit in the main paper, this would still be useful in an appendix or supplementary info.

A: Thank you for the very good suggestion which we are following.

Action: We added a table with describing the symbols used in the equations and one with explaining the acronyms, both in the Supplement. While compiling the list of equations, we discovered 3 symbols which have been used twice but with different meaning and minor inconsistencies in the equations and units which we have corrected.

R: *Calibration.* There are a lot of correction factors applied to the outputs for each basin before a suitable result (defined here as+/-10% of the long term mean annual flow) is obtained. Figure 4 is interesting but the results are not analysed in section 4.9.3 –where do the authors think the major errors are coming from (input data, model process representation) and how does this vary spatially? Given the significant correction factors for a lot of basins, it warrants a discussion on how appropriate the model outputs are for future conditions when you assume that these correction factors remain stationary over time (when

in reality they could change depending on what errors they are accounting for!) and the fact that you correct to the mean but do not consider extremes (so how appropriate the model results may be for floods/droughts). It would also be useful to make these correction factors available as part of the standard model outputs (if they are not already).

A: Thank you for this suggestion. Indeed, we have not discussed the potential reasons for the usage of those calibration factors as a thorough assessment would require much more assessments (e.g. with other precipitation input data, model representations or calibration setups) which is outside of the scope of the manuscript. We have done such an assessment in Müller Schmied et al., 2014 with a sensitivity study and do not want to repeat it. We also do not want to introduce a lengthy discussion in Sect. 4 which is solely a model description. The dominance of the different error sources is spatially and temporally varying and it is difficult to make concise and general statements.

The discussion regarding the usage of historically-derived calibration parameters for future conditions is touched initially in line 624 and referred to the discussion within Krysanova et al. (2018) which we do not want to repeat in the context of this manuscript. Also, a thorough discussion of the impact of calibrating to the mean with respect to floods / droughts (esp. when considering the calibration status) would be an extra study but this suggestion is well received. The many assessments of WaterGAP in context of the Inter-Sectoral Model Intercomparison Project (ISIMIP) model evaluation shows that the model is for large spatial domains the best performing model also for low / high flows (e.g. Zaherpour et al., 2018, Veldkamp et al, 2018, Krysanova et al., 2020). However, we have not yet analyzed yet whether the well performing extreme flows are a result of the calibration or simply of the model structure. We agree that this is a very interesting floor for a separate study but we do not want to overspeculate in this manuscript.

Besides this, we agree that it is a very good idea to provide the calibration parameters to the public.

Action: 1. We have added to the repository four netcdf files with a) gamma, b) CFA, c) CFS and d) the calibration status. 2. We have extended line 626 with the reference Krysanova et al. (2020) as there an assessment of model performance and credibility of climate change impact was done.

R: *Model Application.* Section 7 and 8 do not add to the paper and both need to be significantly shortened or removed. Currently, the paper is very long and while the material before this is very relevant, the results presented in Section 7 do not provide any new information or significant results to the reader. Section 8 could also be significantly shorter. If these sections were shortened then you could expand a little on the interesting discussion of the future model developments that you discuss in the conclusion and perhaps link this in with broader developments in the GHM community. Furthermore, it would make the paper shorter and more readable for readers.

A: Thank you for the suggestion. We disagree and strongly believe that Section 7 does provide new and relevant information to readers who want to understand the WaterGAP model and what is can be applied for. We think that the GMD manuscript type "Model description paper" should include the presentation of model results beyond those that can be compared to observations or independent data (we do this in Section 6 Model Evaluation). A model description paper should also describe the model output for which no observations are available but which has been the whole purpose of model development as it provides information about the global water situation that cannot be obtained without the model. Therefore, we do not want to remove (or shorten) Section 7, which we kept as concise as possible.

We certainly agree that the paper is very long. To shorten it, we agree with the reviewer to remove Section 8 from the main text. We think it is best to move it to the supplement and refer to it in Section 7; it is worth keeping in the supplement as especially for a new model output user (or code user, see the answer to a later referee comment), it can be beneficial to know in which fields the model has been used.

Action: We kept Sect. 7 as is. We moved Sect. 8 to the supplement and renamed the heading to "*WaterGAP application fields*".

R: *Code and Data Availability.* I understand the difficulties with making the code open source. Do you have a timeline of when it will be made open source and how it will be made open source (for example on a platform like GitHub?).

A: Thank you for your understanding. WaterGAP is developed since the mid 90es and therefore a many people have been involved in model development. Since roughly one year we have collected the written consents of all model developers to grant open access code. However, the formal process of making it open accessible is still in negotiations between the Universities where the main parts of code development has been done. We cannot give an estimate when those issues are being solved but hope this is the case in the next couple of weeks or months. The code itself is already integrated in GitHub and can be made available in short time once the license issues are clarified. Nevertheless, this does not necessarily mean that the model can be executed immediately by an external user. This would need a number of input data sets, configurations, user manuals and a strategy of how the model development might be shared within a community. To be able to provide that information and to setup a community strategy, but also to rewrite the model code to a modern style, we are currently seeking for funding opportunities which is not trivial especially as the code is not yet open source.

Action: None required.

R: *Appendices and Supplementary Information.* It would be worth thinking about whether some of your appendix materials (particularly Appendix D) may be better placed in supplementary material rather than the appendix of the paper.

A: Thank you for the suggestion to reduce the paper length.

Action: We moved Appendix D with its seven figures to the supplementary material.

**Minor Comments**

R: Abstract L9. I would replace 'can be done' with 'can be achieved'

A: Thanks.

Action: replaced as suggested

R: Introduction L11. This opening sentence is too long –it needs rewriting.

A: Thanks, also pointed out by Referee #1.

Action: We modified the first sentence to "*A globalized world is characterized by large flows of virtual water among river basins (Hoff et al., 2014) and by international responsibilities for the sustainable development of the Earth System and its inhabitants. The foundation of a sustainable management of water, and more broadly the Earth system, are quantitative estimates of water flows and storages as well as of water demand by humans and freshwater biota on all continents of the Earth (Vörösmarty et al., 2015).*"

R: Introduction L28. What do you mean by 'proper simulation'?

A: Proper in the sense of "sufficient" or "reasonable" in terms of high simulation quality.

Action: We modified "proper" to "*high performing*".

R: Introduction L43. It would be good to have some specific references here of where this variant of the model has been used.

A: Thanks, we intended to distinguish the model version family 2 (operating at 0.5 deg resolution) and 3 (operating at 5 arc min) but this can be misunderstood.

Action: We revised the beginning of the section to "*Water – Global Assessment and Prognosis (WaterGAP), which has been developed since 1996, is one of the pioneers in this field. WaterGAP as described here operates with a spatial resolution of 0.5° x 0.5° and is called the model family WaterGAP 2. Key model versions are WaterGAP 2.1d (Alcamo et al. (2003), Döll et al. (2003), Kaspar (2004)), 2.1e (Schulze & Döll 2004), 2.1f (Hunger & Döll (2008), Döll & Fiedler (2008)), 2.1g (Döll et al., 2008), 2.1h (Döll et al., 2012), 2.2 (Müller Schmied et al. (2014)), 2.2a (Döll et al., 2014)), 2.2(ISIMIP2a) (Müller Schmied et al., 2016), 2.2b (Müller Schmied (2017), Döll et al. (2020)), 2.2c (description submitted to this journal) and 2.2d (this manuscript). In addition, a model family with 5'x 5' is named WaterGAP 3 (Eisner 2015). While the model family 3 has similar algorithms than the model family 2, this paper only refers to the recent model version WaterGAP 2.2d.*"

R: Section 3.2.2 L180. Can you provide a reference or website for the Environmental Data Explorer?

A: Thank you. Actually this was a reference to a website which is listed in the references but due to missing entries in the bib.tex file, the year disappeared in the text. We apologize for any inconvenience this might have raised.

Action: We modified this and any other references to websites where appropriate. The sentence reads now as follows: "*Additionally, population numbers beyond 2005 as well as information on the ratio of rural to urban population of each grid cell come from UNEP (2015).*"

R: Section 4.6 L415. Can you add the specific version of the GRanD database you are using?

A: This is sadly not possible as it was a preliminary and unpublished version of the one published in Lehner et al. (2011).

Action: none

R: Section 4.6 L416. I think 'Sect. E' should be 'Appendix E'?

A: Thank you, good observation!

Action: modified as suggested.

R: Section 4.9.1 L621. "to avoid that average water resources are misrepresented" –this isn't clear as written, can you be more specific?

A: Good point, the sentence is indeed not very specific.

Action: We modified the beginning of this section to "*The main purpose of WaterGAP is to quantify water resources and water stress for both historical time periods and scenarios of the future. Not only due to very uncertain global climate input data, uncalibrated global hydrological models may compute very biased runoff and streamflow values (e.g. Haddeland et al. 2011). To reduce the bias and simulate at least mean streamflow and thus renewable water resources with a reasonable reliability, WGHM has been calibrated to match observed long-term average annual streamflow at gauging stations on all continents (Döll et al., 2003, Kaspar, 2004). Calibration is required…*"

R: Section 4.9.1 L634. One of the key outcomes from Coxon et al (2015) was that the discharge uncertainty varied significantly between gauging stations and over the flow range. It may be worth adding a sentence somewhere in the paper stating that you recognise that the discharge uncertainty is unlikely to be stationary in space and time but there are no further data to better constrain the uncertainties at these gauging stations so a representative value of +/-10% is used.

A: Thank you for this very good advice.

Action: We added the following sentence to line 635: "*It is noteworthy that the discharge uncertainty (approximated here with +/- 10%) is unlikely to be stationary in space and time (Coxon et al, 2015) but there are no further data available to better constrain the specific uncertainty of each gauging station.*"

R: Section 6.4.1. Missing 'h' from 'withdrawals' in title

A: Thank you!

Action: typo solved

R: Figure 5. I wasn't sure whether the NSE and logNSE values presented in Figure 5 were calculated based on all the monthly country data or were a median value for each country? It is notclear how it was calculated for these variables.

A: Thank you for pointing out this potential source of misunderstanding. NSE (and log NSE) in Figure 5 was calculated using each single data point (yearly) of FAO AQUASTAT and the corresponding simulated value.

Action: We added the sentence "*The evaluation metrics (Sect. 6.3.1) are calculated using each single data point of AQUASTAT, without any temporal aggregation by country.*" at the end of Sect. 6.2.1.

R: Section 6.4.2 L785. "is rather satisfying" – I would remove this and just present the results

A: Good point.

Action: We modified the sentence to: "*The performance of WaterGAP 2.2d in terms of monthly streamflow time series at 1319 gauging station (Fig. 8) reaches a median NSE(KGE) of 0.52 (0.61).*"

R: Section 6.5. Can you attribute some of these improvements in model performance to specific changes made to the model?

A: Without an extensive experiment setup which would include simulations (and calibrations) of each modification in a stepwise manner, a scientifically based answer is not possible, and we decided not to speculate.

Action: none

R: Conclusions. An additional development here could be improving the lakes/wetlands and reservoir regulation as you noted this being a limiting factor to good model performance in North America/Canada?

A: Thank you for the suggestion. Indeed, there are many areas for future development such as the improvements mentioned. However, all of these potential improvements require funding to do research and implementation, hence we have listed only those lines that are currently under development (see also Line 1018).

Action: none

References

Alcamo, J., Döll, P., Henrichs, T., Kaspar, F., Lehner, B., Rösch, T., & Siebert, S. (2003). Development and testing of the WaterGAP 2 global model of water use and availability.

*Hydrological Sciences Journal*, *48*(3), 317–337.
https://doi.org/10.1623/hysj.48.3.317.45290

Döll, P., & Fiedler, K. (2008). Global-scale modeling of groundwater recharge. *Hydrology and Earth System Sciences*, *12*(3), 863–885. https://doi.org/10.5194/hess-12-863-2008

Döll, P., Fiedler, K., & Zhang, J. (2009). Global-scale analysis of river flow alterations due to water withdrawals and reservoirs. *Hydrology and Earth System Sciences*, *13*(12), 2413–2432. https://doi.org/10.5194/hess-13-2413-2009

Döll, P., Hoffmann-Dobrev, H., Portmann, F. T., Siebert, S., Eicker, A., Rodell, M., Strassberg, G., & Scanlon, B. R. (2012). Impact of water withdrawals from groundwater and surface water on continental water storage variations. *Journal of Geodynamics*, *59–60*, 143–156. https://doi.org/10.1016/j.jog.2011.05.001

Döll, P., Kaspar, F., & Lehner, B. (2003). A global hydrological model for deriving water availability indicators: model tuning and validation. *Journal of Hydrology*, *270*(1–2), 105–134. https://doi.org/10.1016/S0022-1694(02)00283-4

Döll, P., Müller Schmied, H., Schuh, C., Portmann, F. T., & Eicker, A. (2014). Global-scale assessment of groundwater depletion and related groundwater abstractions: Combining hydrological modeling with information from well observations and GRACE satellites. *Water Resources Research*, *50*(7), 5698–5720. https://doi.org/10.1002/2014WR015595

Eisner, S. (2015). *Comprehensive evaluation of the WaterGAP3 model across climatic, physiographic, and anthropogenic gradients* [PhD thesis]. Kassel University.

Hunger, M., & Döll, P. (2008). Value of river discharge data for global-scale hydrological modeling. *Hydrology and Earth System Sciences*, *12*(3), 841–861. https://doi.org/10.5194/hess-12-841-2008

Krysanova, V., Zaherpour, J., Didovets, I., Gosling, S. N., Gerten, D., Hanasaki, N., Müller Schmied, H., Pokhrel, Y., Satoh, Y., Tang, Q., & Wada, Y. (2020). How evaluation of global hydrological models can help to improve credibility of river discharge projections under climate change. *Climatic Change*. https://doi.org/10.1007/s10584-020-02840-0

Lehner, B., Liermann, C. R., Revenga, C., Vörösmarty, C., Fekete, B., Crouzet, P., Döll, P., Endejan, M., Frenken, K., Magome, J., Nilsson, C., Robertson, J. C., Rödel, R., Sindorf, N., & Wisser, D. (2011). High-resolution mapping of the world's reservoirs and dams for sustainable river-flow management. *Frontiers in Ecology and the Environment*, *9*(9), 494–502. https://doi.org/10.1890/100125

Müller Schmied, H. (2017). *Evaluation, modification and application of a global hydrological model* [Goethe-University Frankfurt]. http://publikationen.ub.uni-frankfurt.de/frontdoor/index/index/year/2017/docId/44073

Müller Schmied, H., Adam, L., Eisner, S., Fink, G., Flörke, M., Kim, H., Oki, T., Portmann, F. T., Reinecke, R., Riedel, C., Song, Q., Zhang, J., & Döll, P. (2016). Variations of global and continental water balance components as impacted by climate forcing uncertainty and human water use. *Hydrology and Earth System Sciences*, *20*(7), 2877–2898. https://doi.org/10.5194/hess-20-2877-2016

Müller Schmied, H., Eisner, S., Franz, D., Wattenbach, M., Portmann, F. T., Flörke, M., & Döll, P. (2014). Sensitivity of simulated global-scale freshwater fluxes and storages to input data, hydrological model structure, human water use and calibration. *Hydrology and Earth System Sciences*, *18*(9), 3511–3538. https://doi.org/10.5194/hess-18-3511-2014

Schulze, K., & Döll, P. (2004). Neue Ansätze zur Modellierung von Schneeakkumulation und -schmelze im globalen Wassermodell WaterGAP. In R. Ludwig, D. Reichert, & W. Mauser (Eds.), *Tagungsband zum 7. Workshop zur großskaligen Modellierung in der Hydrologie* (Issue November 2003, pp. 145–154). Kassel University Press. https://www.upress.uni-kassel.de/katalog/abstract_en.php?978-3-89958-072-3

UNEP. (2015). The Environmental Data Explorer, as compiled from United Nations Population Division. http:/ede.grid.unep.ch

Veldkamp, T. I. E., Zhao, F., Ward, P. J., De Moel, H., Aerts, J. C. J. H., Schmied, H. M., Portmann, F. T., Masaki, Y., Pokhrel, Y., Liu, X., Satoh, Y., Gerten, D., Gosling, S. N., Zaherpour, J., & Wada, Y. (2018). Human impact parameterizations in global hydrological models improve estimates of monthly discharges and hydrological extremes: A multi-model validation study. *Environmental Research Letters*, *13*(5), 055008. https://doi.org/10.1088/1748-9326/aab96f

Zaherpour, J., Gosling, S. N., Mount, N., Schmied, H. M., Veldkamp, T. I. E., Dankers, R., Eisner, S., Gerten, D., Gudmundsson, L., Haddeland, I., Hanasaki, N., Kim, H., Leng, G., Liu, J., Masaki, Y., Oki, T., Pokhrel, Y., Satoh, Y., Schewe, J., & Wada, Y. (2018). Worldwide evaluation of mean and extreme runoff from six global-scale hydrological models that account for human impacts. *Environmental Research Letters*, *13*(6), 065015. https://doi.org/10.1088/1748-9326/aac547

---

## Author Comment (AC3) · 24 Nov 2020

**Additional modifications of the manuscript**

While working on the valuable referee comments but also by direct feedback from readers of the discussion paper, we modified the manuscript as follows:

- L435: replaced 1386 global lakes by 1355 which is the correct value
- L635: replaced 1978 by 1980
- Table 1: corrected Pangaea file name for global wetland storage to glowetlandstor (was locwetlandstor)
- Replaced the reference Kaspar (2003) by the more appropriate published version Kaspar (2004)
- Replaced citation of ERA5 to the now published journal publication
- ISIMIP written correctly as Model (was Modelling)
- small rephrasing (e.g. Abstract) and corrections for consistency reasons
- clarified $NA_{pot,g}$ and $NA_{pot,s}$ as potential net abstractions where appropriate
- added Figure caption for Fig. D1 (was missing)

---

## Author Response (AR1)

**Answer to comments of Anonymous Referee #1**

The original comments of Referee #1 are in black color and indicated by "R:". Replies by the authors ("A") are colored in green. Actions are introduced by "Action:", changes in the manuscript are in italics.

**General comments:**

R: The authors provide a detailed model description of the latest WaterGAP global hydrological model and specification and validation of its standard output data. The model description part covers the entire model but puts extra weights on the improvements and advances since Müller-Schmied et al. (2014) which reported the last model updates. The standard output data part concisely compiles related information for the potential users.

WaterGAP is a great model that has lead the field of global water resources research for two decades. I believe this paper provides a foundation and a benchmark to the research community.

It is noteworthy that the model description includes the detailed procedure of hydrological parameter tuning. One of the distinct characteristics of WaterGAP is that the developers have conducted painstaking manual hydrological parameter tuning at more than one thousand basins. This feature brings a distinct performance compared to other global hydrological models (mostly untuned), but the procedure was virtually unseeable for non-developers since available descriptions were quite old (Döll et al., 2003; Hunger and Döll, 2008). The clear and detailed description in this paper will be helpful for understanding the outputs of WaterGAP, in particular, those who are interested in intercomparing models.

The disclosure of standard output should be highly appreciated. Although numerous model intercomparison projects have been conducted (e.g. WaterMIP, ISIMIP/globalwater), the performance of models tends to fall short of that of under the original (model-optimum) condition. The broad community will be benefited from the provided data.

The manuscript is certainly long, but well structured and written. The major contents are, as noted earlier, the description of the latest model and outputs which is essentially a summary of past six years of peer reviewed papers. Due to the nature of this manuscript, I haven't rigorously examined the methods and results one by one. Rather I have read and commented this manuscript from the viewpoint of a learner of the model and a user of the outputs. Hope the specific comments below are useful for further revision.

A: Thank you for the overall very positive comments and encouragement. Regarding the comment on the parameter tuning please allow us a clarification. The parameter tuning is not done manually but using a – specifically developed for WaterGAP - automatic calibration framework on multiple nodes of a computation cluster.

**Specific comments**

R: Line 11: This sentence is too long. Better to split into two or three.

A: We agree.

Action: We modified the introduction sentence as follows: "*A globalized world is characterized by large flows of virtual water among river basins (Hoff et al., 2014) and by international responsibilities for the sustainable development of the Earth System and its inhabitants. The foundation of a sustainable management of water, and more broadly the Earth system, are quantitative estimates of water flows and storages as well as of water demand by humans and freshwater biota on all continents of the Earth (Vörösmarty et al., 2015).*"

R: Line 20 'Environmental Performance Index': This term needs a reference.

A: Thanks. Indeed the EPI in parentheses is a citation (and referred to in the literature) but we agree that it is ambiguous as it can be understood as simple abbreviation. But while cross-checking this indicator (https://epi.yale.edu/), it came up that the EPI methodology does not consider the "water scarcity" indicator (where global models are being used) after the 2010 version of the indicator.

Action: We deleted this indicator from the list.

R: Line 86: 'Hyungjun' reads 'Kim'.

A: Thanks for the hint and sorry for this mistake.

Action: we corrected the citation.

R: Line 137 'Cropping patterns and growing periods are generated for every year': A bit confusing. The authors wrote that growing period is fixed at 150 days. What does this part mean?

A: The principle is described in Section 3.1.1. The individual growing period in a given grid cell have a **fixed length** of 150-days, whereas the **start (day of the year)** is not fixed, but depends on climate data of each grid cell: 30-year-average monthly temperature, precipitation (30-year-average monthly sums and number of rain days) and potential evapotranspiration (30-year-average monthly shortwave radiation and fixed mask of arid/humid grid cells, for Priestley-Taylor approach, see Eq. (7) in line 318).

As described in line 120, using the ranking criteria explained in Döll and Siebert (2002), "The most highly ranked 150-day period(s) is/are defined as growing season(s)"

In the current model version, for the calculation of cropping patterns and growing periods, each year the climate averages are calculated externally to WaterGAP as running means with the current year as the central year (exception: at the beginning or ending part/15 years of the time series).

This is consistent with the assumption that farmers' choice of cropping patterns and growing periods are rather based on long-term experience than on short-term weather.

Action: We have added a reference to section 3.1.1 in line 137.

R: Line 138 'the respective 30-year climate averages': A bit hard to read. Which climate variables are year-specific and which are 30-year mean?

A: As mentioned in the previous comment, for the calculation of cropping patterns and growing periods 30-year averages are used (monthly variables: temperature, precipitation, number of rain days, shortwave radiation).

Action: As mentioned above, we added a reference to section 3.1.1 in line 137.

R: Line 265 'Increases in soil water storage in irrigated areas are not taken into account': I am wondering how evapotranspiration from irrigated area is estimated in this model. I guess abstracted water for irrigation is directly added to evapotranspiration of a gridcell, but this should be clearly elaborated.

A: The model calculates soil evapotranspiration Es (Eq. 17), sublimation Esn (Eq. 14), evaporation from canopy Ec (Eq. 6), evaporation from water bodies (Eq. 22) which in sum can be seen as actual evapotranspiration. For the output described in Table 2 as actual evapotranspiration Ea we add the actual consumptive water use WCa (which is the sum of NAs and NAg (Section 3.3)) to consider the "lost" water to the atmosphere as additional part of evapotranspiration. Hence, consumed water (for all water uses) are included in the model output provided as Ea. We, however agree that this note on Table 2 might not be prominent enough.

Action: We added a sentence after the mentioned line 265. "*To consider anthropogenic consumptive water use in the output variable of actual evapotranspiration Ea (Table 2), we sum up all evapo(transpi)ration components and actual consumptive water use WCa (see note 5 in Table 2).*"

R: Line 318 Equation 7: Seems LAI was used only for canopy storage calculation. Is this really the case? I wish to see the list of variables which are directly affected by the daily dynamics of LAI. This point must be important to understand/interpret the outputs of WaterGAP model.

A: The referee is right. The LAI model is only effective for canopy storage calculation (Eq. 3-6). Hence, it is Ec and Sc directly affected. LAI development in terms of plant growth for irrigation is done specifically in the submodel of GIM (Sect. 3.1). Transpiration is not simulated separately, only jointly with the soil evaporation (Sect. 4.4).

Action: We added the following two sentences before Eq. 6 for clarification: "*It is noteworthy that in WaterGAP L only affects the calculation of the canopy water balance. L is not taken into account in computing consumptive water use of irrigated crops (Sect. 3.1) and evapotranspiration from land (Sect. 4.4).*"

R: Line 611 'Unsatisfied water use is added to NAs of the next day until the end of the calendar year': It sounds that this treatment can result in a quite unnatural hydrograph. For example, for the rivers affected by the monsoon system, the increase in wet season's discharge must be substantially delayed, because the initial increase in runoff is used for the 'repayment of water loan' accumulated in the preceding dry season. Please consider adding a

note on the consequences of this assumption/treatment which would be helpful for the readers. Finally, as a hydrologist living in an Asian country, I need to write here that this assumption/treatment is quite odd. The drastic seasonal change in water availability is the heart of the water scarcity problem in our region, which seems largely (if not completely) unaccounted by this model (see discussion in Hanasaki etal. 2008, HESS).

A: Thank you for rising this important issues. With the delayed option we are aiming at compensating that WaterGAP likely underestimates storage of water e.g. by small tanks and dams, and because of the generic reservoir operation scheme. The delayed satisfaction scheme may, however, overestimate satisfaction of surface water demand in particular in highly seasonal flow regimes. With regards to the effect of adding unsatisfied use to NAs of the next day, we have done an additional simulation with this feature disabled. We took the Nash-Sutcliffe-Efficiency as indicator for substantial deviations of the hydrograph. From the 1319 river basins assessed, there are only 20 river basins where this indicator deviates by more than +-0.1 (for 3 stations NSE improved by more than 0.1; the median NSE slightly decreases from 0.5226 in 2.2d to 0.5225 in the variant without delayed satisfaction scheme) in the two model variants, all outside of monsoon regions. For monsoon regions (as the Yangtze river, see Fig. 1 below) the effect is – even though there are large potential NAs values calculated in this basin - not visible in the hydrograph as the seasonal change in the hydrograph is much stronger than the effect of delayed satisfaction of NAs. However, in non-monsoon-regions (Fig. 2 & 3), there are certain effects visible, especial in (or better after) dry phases. Furthermore, we have assessed the per cent satisfaction of actual NAs to potential NAs with and without delayed satisfaction of NAs. With the delayed satisfaction of potential NAs as computed in GWSWUSE, 92.5% of global potential NAs during 1981-2010 is satisfied, but only 82.2% in case of the alternative option that surface water demand needs to be satisfied by available water on the same day. Fig. 4 below provides information of the spatial distribution of these differences. Overall, switching off the delayed reduces the satisfaction of NAs especially in the dry regions. We believe that especially there, local storage systems are installed which might be represented by the delayed satisfaction of NAs.

Action: We have added the three hydrographs to the supplement, added the following paragraph after line 614 and hope that with this additional text we have covered the concerns of the referee including Hanasaki et al. (2008): "*Delayed satisfaction aims at compensating that WaterGAP likely underestimate storage of water e.g. by small tanks and dams, and because of the generic reservoir operation scheme. Without delayed satisfaction, less than 50% of potential NAs could be satisfied in many semi-arid regions (Fig. S8). The delayed satisfaction scheme may overestimate satisfaction of surface water demand in particular in highly seasonal flow regimes. However, this effect is hardly visible in the hydrograph of the monsoonal Yangtze river (Fig. S9) but more visible in semi-arid regions (Fig. S10, S11). With delayed satisfaction of potential NAs, 92.5% of global potential NAs during 1981-2010 is satisfied, but only 82.2% in case of the alternative option that surface water demand needs to be satisfied by available surface water on the same day.*"

Answers to referee comments to https://doi.org/10.5194/gmd-2020-225

[Figure]

*Figure 1: Hydrograph of Yangtze river at Datong station with standard 2.2d and a variant without delayed satisfaction of water use as well as with the GRDC data included.*

[Figure]

*Figure 2 Hydrograph of Syr Darya river at Bekabad station with standard 2.2d and a variant without delayed satisfaction of water use as well as with the GRDC data included.*

[Figure]

*Figure 3 Hydrograph of Murray river at Lock 9 station with standard 2.2d and a variant without delayed satisfaction of water use as well as with the GRDC data included.*

[Figure]

*Figure 4: The spatial impact of delayed satisfaction of NAs, showing a lower satisfaction especially in dry regions compared to the standard variant. Values are expressed in percent.*

R: Line 629 'areal correction factor (CFA)': Why is this term called 'areal'?

A: In contrast of the station correction factor CFS which is effective only in the outflow cell of the calibration basin, the areal correction factor (CFA) is spatially distributed in the river basin. In line 641 we have briefly explained the calculation of CFA.

Action: We referred to the calibration status CS3 and CS4 and to Hunger & Döll 2008 in line 629.

R: Line 647 'For global water balance assessment the mass balance is kept by the actual evapotranspiration component': Does this mean that the actual evapotranspiration is simply calculated by P − Q?

A: With applying CFS we destroy the water balance in the grid cell where this factor is applied. For example, a streamflow Qsim, of let's assume 1000 m3/s, needs to be multiplied by a factor of 0.5 (Qmod 500 m3/s) to match to the observed streamflow, the water balance lacks of 500 m3/s. One may choose to add this amount to evapotranspiration. But this might lead to an overestimation of evapotranspiration if the reason for the CFS is that precipitation is overestimated by the climate forcing. If, with a CFS > 1; evapotranspiration would need to be reduced and could get therefore to negative values, the water balance stays unclosed. However, for water balance calculations as in Table 6, we have considered the CFS effect by adding (removing) the CFS-adapted streamflow to (from) the actual evapotranspiration. As the second referee pointed out that it would be of benefit to provide the CFS (and gamma and CFA) values for clarity, we follow this advice.

Action: We added gamma, CFA and CFS and calibration status to the model output at Pangaea and added the sentence after line 675: "*Additionally, the calibration factors γ, CFA,*

*CFS and the calibration status (Sect. 4.9) are provided.*" and we modified the sentence in line 647 by adding "*by the amount CFS modified streamflow*".

R: Line 779 "NSE and logarithmic NSE": NSE is usually calculated between two time-series at a single location (e.g monthly simulated and observed discharge). How NSE in Figure 5 was calculated? Seems nation-wise NSE was calculated using five-year interval time series (i.e. the typical interval of FAO AQUASTAT is five-year), then averaged globally, but is this really the case?

A: NSE (and log NSE) in Figure 5 was calculated using each single data point of FAO AQUASTAT and the corresponding simulated value. We want to show the general skill of the water use assessment. The logarithmic NSE is used to elaborate differences in small numbers.

Action: We added the sentence "*The evaluation metrics (Sect. 6.3.1) are calculated using each single data point of AQUASTAT, without any temporal aggregation by country.*" at the end of Sect. 6.2.1.

R: Line 785 "However, NSE Values below 0 for 259 stations show the complete failure of WaterGAP2.2d to simulate streamflow dynamics in one fifth of the evaluated basins": I feel that this sentence is a bit unclear. What I understood is that monthly and annual variations were not properly reproduced for these 259 stations, although the simulated mean annual discharge agrees well with that of observation due to the calibration.

A: Thank you for this issue with wording and suggestion to revise.

Action: We modified this sentence to: "*However, NSE values below 0 for 259 stations show that WaterGAP2.2d cannot reproduce monthly and annual streamflow dynamics in one fifth of the evaluated basins, although the simulated mean annual streamflow fits to the observations due to the calibration.*"

R: Line 775 "reasonable quality": I don't know what this phrase exactly indicates. The log-log scatter plot (Figure 5) is not very helpful for judging model performance. At least some additional notes are needed for the results of industrial sector (Figure 5e) which indicates frequent occurrence of two orders of magnitude overestimation between simulation and observation.

A: We refer with this phrase to the NSE and logNSE values which indicates quite good values (between 0.67 and 0.92, Fig. 5 d-f). The log-log plot was chosen to avoid distortion due to few high values. The alternative would be to show no-log axes as shown in Figure 5 below which does in our perspective not allow a better assessment. Nevertheless, we see the value of adding this Figure to the supplement. With respect to the mismatch of WaterGAP estimates and FAO AQUASTAT for some data points of the industrial sector, we provide additional notes as suggested by the referee. In general, a mismatch between WaterGAP outputs and data from FAO AQUASTAT can occur through the use of different sources

because WaterGAP does not build on AQUASTAT data rather on national statistics (Flörke et al, 2013).

[Figure]

*Figure 5: Same as Figure 5 of the manuscript but not with log-log axes.*

Action: 1) We added Fig. 5 to the supplement and referred to it in the text.

2) We added additional text after Line 781 to further explain the model performance with regard to industrial water uses. "*In terms of overestimated values, values for India and Germany dominate the differences in the time intervals 2008-2012 and 2013-2016, respectively. Water withdrawals of 56 km³ for the industry sector (including thermoelectric) was assessed by India's National Commission on Integrated Water Resources Development for 2010 (Bhat, 2014). Here AQUASTAT reports 17 km³ yr⁻¹ and WaterGAP simulates 72 km³ yr⁻¹. In case of Germany, AQUASTATs reports only the water use of manufacturing sector but omits the water abstractions of cooling water for thermal electricity production that is included in the WaterGAP results. (i). (ii) The underestimation of industrial water uses >200 km³ yr⁻¹ (Fig. S12e) is particularly biased by the reported numbers from the US statistics. While AQUASTAT data includes both freshwater and saline water abstractions from manufacturing, thermoelectric and mining, WaterGAP only accounts for the freshwater part of the manufacturing and thermoelectric abstractions.*"

R: Line 938-945 "In case of negative NAs…": Highly technical and hard to read. It would be helpful for readers if the authors add links to the directly relevant model description parts (e.g. subsection or equation).

A: Thank you for pointing out that the description should be better phrased.

Action: We modified the section starting in line 936 to: "*As noted in Sect. 4.8, the actual net abstractions can differ from its potential values. The ratio of actual to potential net surface water abstractions NAs (Fig. 15c) shows a heterogeneous pattern, with adjacent grid cells with values below 0.9 and above 1.1. This is explained by the option to satisfy water demand from a neighboring grid cell. In case of negative NAs, potential and actual values are always the same as it is assumed in the model that NAg can always be fulfilled so that return flows to surface water are not changed. There are only a few longer river stretches where actual NAs is smaller than the potential value.*

*Actual NAg is equal to potential NAg except in a few grid cells where potential NAs cannot be fulfilled and there is irrigation with surface water (Fig. 15d). In these cells, return flows to groundwater decrease and actual values of NAg increase compared to their potential values. For example, in case of a positive (negative) potential NAg, a ratio of 1.1 (0.9) means that the difference between actual and potential NAg is 10% of the absolute value of potential NAg. In most grid cells, actual NAg is equal to the potential value.*"

R: Line 949 Table 6: What does negative values for 'actual net abstraction from groundwater' indicate? Does it mean that groundwater recharge has been increased by humanities? I am quite confused because Table 7 indicates that the groundwater is being depleted globally. Similarly, add some extra notes for the negative values for 'change of total water storage' which look constantly increasing by time. What are the key reasons for this?

A: Thank you for the good question. We feel that the description in section 7.3.2 (lines 956 to 967) are not carefully expressed, hence we have rewritten this section, and added an explanation for positive and negative values of NAs and NAg.

Action: 1) In line 936, we added an explanation for positive and negative values of NAs and NAg. "*Positive values of NAs and NAg indicate that human water use results in a net subtraction of water from surface water bodies and groundwater while negative values indicate a man-made addition of water to these water storage compartments.*"

2) We have revised L 958 to L 962 by "*The negative value of actual net abstraction from groundwater in Table 6 indicates that globally aggregated, the groundwater compartment is recharged by return flows from irrigation with surface water (addition of the positive and negative values of NAg in Fig. 5b). A globally averaged anthropogenic increase in groundwater recharge is consistent with a decrease of groundwater storage that is mainly caused the net groundwater abstractions. The global groundwater storage, however, has decreased (Table 7) mainly due to groundwater depletion in those grid cells where (positive) NAg is higher than groundwater recharge (Döll et al. 2014). The anthropogenic net recharge of groundwater in the grid cells with negative NAg in Fig. 5b does not lead to a substantial increase in groundwater storage but mainly increases groundwater discharge to surface water bodies. The decreasing trend of total water storage is dominated by increasing water storage losses that were balanced in earlier periods by increased water storage in newly*

*constructed reservoirs while dam construction became less during the last three decades (Table 7, Cáceres et al., 2020)."*

A: Thank you for the very good suggestion which we are following.

Action: We added a table with describing the symbols used in the equations and one with explaining the acronyms, both in the Supplement. While compiling the list of equations, we discovered 3 symbols which have been used twice but with different meaning and minor inconsistencies in the equations and units which we have corrected.

R: *Calibration.* There are a lot of correction factors applied to the outputs for each basin before a suitable result (defined here as+/-10% of the long term mean annual flow) is obtained. Figure 4 is interesting but the results are not analysed in section 4.9.3 –where do the authors think the major errors are coming from (input data, model process representation) and how does this vary spatially? Given the significant correction factors for a lot of basins, it warrants a discussion on how appropriate the model outputs are for future conditions when you assume that these correction factors remain stationary over time (when

in reality they could change depending on what errors they are accounting for!) and the fact that you correct to the mean but do not consider extremes (so how appropriate the model results may be for floods/droughts). It would also be useful to make these correction factors available as part of the standard model outputs (if they are not already).

A: Thank you for this suggestion. Indeed, we have not discussed the potential reasons for the usage of those calibration factors as a thorough assessment would require much more assessments (e.g. with other precipitation input data, model representations or calibration setups) which is outside of the scope of the manuscript. We have done such an assessment in Müller Schmied et al., 2014 with a sensitivity study and do not want to repeat it. We also do not want to introduce a lengthy discussion in Sect. 4 which is solely a model description. The dominance of the different error sources is spatially and temporally varying and it is difficult to make concise and general statements.

The discussion regarding the usage of historically-derived calibration parameters for future conditions is touched initially in line 624 and referred to the discussion within Krysanova et al. (2018) which we do not want to repeat in the context of this manuscript. Also, a thorough discussion of the impact of calibrating to the mean with respect to floods / droughts (esp. when considering the calibration status) would be an extra study but this suggestion is well received. The many assessments of WaterGAP in context of the Inter-Sectoral Model Intercomparison Project (ISIMIP) model evaluation shows that the model is for large spatial domains the best performing model also for low / high flows (e.g. Zaherpour et al., 2018, Veldkamp et al, 2018, Krysanova et al., 2020). However, we have not yet analyzed yet whether the well performing extreme flows are a result of the calibration or simply of the model structure. We agree that this is a very interesting floor for a separate study but we do not want to overspeculate in this manuscript.

Besides this, we agree that it is a very good idea to provide the calibration parameters to the public.

Action: 1. We have added to the repository four netcdf files with a) gamma, b) CFA, c) CFS and d) the calibration status. 2. We have extended line 626 with the reference Krysanova et al. (2020) as there an assessment of model performance and credibility of climate change impact was done.

R: *Model Application.* Section 7 and 8 do not add to the paper and both need to be significantly shortened or removed. Currently, the paper is very long and while the material before this is very relevant, the results presented in Section 7 do not provide any new information or significant results to the reader. Section 8 could also be significantly shorter. If these sections were shortened then you could expand a little on the interesting discussion of the future model developments that you discuss in the conclusion and perhaps link this in with broader developments in the GHM community. Furthermore, it would make the paper shorter and more readable for readers.

A: Thank you for the suggestion. We disagree and strongly believe that Section 7 does provide new and relevant information to readers who want to understand the WaterGAP model and what is can be applied for. We think that the GMD manuscript type "Model description paper" should include the presentation of model results beyond those that can be compared to observations or independent data (we do this in Section 6 Model Evaluation). A model description paper should also describe the model output for which no observations are available but which has been the whole purpose of model development as it provides information about the global water situation that cannot be obtained without the model. Therefore, we do not want to remove (or shorten) Section 7, which we kept as concise as possible.

We certainly agree that the paper is very long. To shorten it, we agree with the reviewer to remove Section 8 from the main text. We think it is best to move it to the supplement and refer to it in Section 7; it is worth keeping in the supplement as especially for a new model output user (or code user, see the answer to a later referee comment), it can be beneficial to know in which fields the model has been used.

Action: We kept Sect. 7 as is. We moved Sect. 8 to the supplement and renamed the heading to "*WaterGAP application fields*".

R: *Code and Data Availability.* I understand the difficulties with making the code open source. Do you have a timeline of when it will be made open source and how it will be made open source (for example on a platform like GitHub?).

A: Thank you for your understanding. WaterGAP is developed since the mid 90es and therefore a many people have been involved in model development. Since roughly one year we have collected the written consents of all model developers to grant open access code. However, the formal process of making it open accessible is still in negotiations between the Universities where the main parts of code development has been done. We cannot give an estimate when those issues are being solved but hope this is the case in the next couple of weeks or months. The code itself is already integrated in GitHub and can be made available in short time once the license issues are clarified. Nevertheless, this does not necessarily mean that the model can be executed immediately by an external user. This would need a number of input data sets, configurations, user manuals and a strategy of how the model development might be shared within a community. To be able to provide that information and to setup a community strategy, but also to rewrite the model code to a modern style, we are currently seeking for funding opportunities which is not trivial especially as the code is not yet open source.

Action: None required.

R: *Appendices and Supplementary Information.* It would be worth thinking about whether some of your appendix materials (particularly Appendix D) may be better placed in supplementary material rather than the appendix of the paper.

A: Thank you for the suggestion to reduce the paper length.

Action: We moved Appendix D with its seven figures to the supplementary material.

**Minor Comments**

R: Abstract L9. I would replace 'can be done' with 'can be achieved'

A: Thanks.

Action: replaced as suggested

R: Introduction L11. This opening sentence is too long –it needs rewriting.

A: Thanks, also pointed out by Referee #1.

Action: We modified the first sentence to *"A globalized world is characterized by large flows of virtual water among river basins (Hoff et al., 2014) and by international responsibilities for the sustainable development of the Earth System and its inhabitants. The foundation of a sustainable management of water, and more broadly the Earth system, are quantitative estimates of water flows and storages as well as of water demand by humans and freshwater biota on all continents of the Earth (Vörösmarty et al., 2015)."*

R: Introduction L28. What do you mean by 'proper simulation'?

A: Proper in the sense of "sufficient" or "reasonable" in terms of high simulation quality.

Action: We modified "proper" to *"high performing"*.

R: Introduction L43. It would be good to have some specific references here of where this variant of the model has been used.

A: Thanks, we intended to distinguish the model version family 2 (operating at 0.5 deg resolution) and 3 (operating at 5 arc min) but this can be misunderstood.

Action: We revised the beginning of the section to *"Water – Global Assessment and Prognosis (WaterGAP), which has been developed since 1996, is one of the pioneers in this field. WaterGAP as described here operates with a spatial resolution of 0.5° x 0.5° and is called the model family WaterGAP 2. Key model versions are WaterGAP 2.1d (Alcamo et al. (2003), Döll et al. (2003), Kaspar (2004)), 2.1e (Schulze & Döll 2004), 2.1f (Hunger & Döll (2008), Döll & Fiedler (2008)), 2.1g (Döll et al., 2008), 2.1h (Döll et al., 2012), 2.2 (Müller Schmied et al. (2014)), 2.2a (Döll et al., 2014)), 2.2(ISIMIP2a) (Müller Schmied et al., 2016), 2.2b (Müller Schmied (2017), Döll et al. (2020)), 2.2c (description submitted to this journal) and 2.2d (this manuscript). In addition, a model family with 5'x 5' is named WaterGAP 3 (Eisner 2015). While the model family 3 has similar algorithms than the model family 2, this paper only refers to the recent model version WaterGAP 2.2d."*

R: Section 3.2.2 L180. Can you provide a reference or website for the Environmental Data Explorer?

A: Thank you. Actually this was a reference to a website which is listed in the references but due to missing entries in the bib.tex file, the year disappeared in the text. We apologize for any inconvenience this might have raised.

Action: We modified this and any other references to websites where appropriate. The sentence reads now as follows: "*Additionally, population numbers beyond 2005 as well as information on the ratio of rural to urban population of each grid cell come from UNEP (2015).*"

R: Section 4.6 L415. Can you add the specific version of the GRanD database you are using?

A: This is sadly not possible as it was a preliminary and unpublished version of the one published in Lehner et al. (2011).

Action: none

R: Section 4.6 L416. I think 'Sect. E' should be 'Appendix E'?

A: Thank you, good observation!

Action: modified as suggested.

R: Section 4.9.1 L621. "to avoid that average water resources are misrepresented" –this isn't clear as written, can you be more specific?

A: Good point, the sentence is indeed not very specific.

Action: We modified the beginning of this section to "*The main purpose of WaterGAP is to quantify water resources and water stress for both historical time periods and scenarios of the future. Not only due to very uncertain global climate input data, uncalibrated global hydrological models may compute very biased runoff and streamflow values (e.g. Haddeland et al. 2011). To reduce the bias and simulate at least mean streamflow and thus renewable water resources with a reasonable reliability, WGHM has been calibrated to match observed long-term average annual streamflow at gauging stations on all continents (Döll et al., 2003, Kaspar, 2004). Calibration is required…*"

R: Section 4.9.1 L634. One of the key outcomes from Coxon et al (2015) was that the discharge uncertainty varied significantly between gauging stations and over the flow range. It may be worth adding a sentence somewhere in the paper stating that you recognise that the discharge uncertainty is unlikely to be stationary in space and time but there are no further data to better constrain the uncertainties at these gauging stations so a representative value of +/-10% is used.

A: Thank you for this very good advice.

Action: We added the following sentence to line 635: "*It is noteworthy that the discharge uncertainty (approximated here with +/- 10%) is unlikely to be stationary in space and time (Coxon et al, 2015) but there are no further data available to better constrain the specific uncertainty of each gauging station.*"

R: Section 6.4.1. Missing 'h' from 'withdrawals' in title

A: Thank you!

Action: typo solved

R: Figure 5. I wasn't sure whether the NSE and logNSE values presented in Figure 5 were calculated based on all the monthly country data or were a median value for each country? It is notclear how it was calculated for these variables.

A: Thank you for pointing out this potential source of misunderstanding. NSE (and log NSE) in Figure 5 was calculated using each single data point (yearly) of FAO AQUASTAT and the corresponding simulated value.

Action: We added the sentence "*The evaluation metrics (Sect. 6.3.1) are calculated using each single data point of AQUASTAT, without any temporal aggregation by country.*" at the end of Sect. 6.2.1.

R: Section 6.4.2 L785. "is rather satisfying" – I would remove this and just present the results

A: Good point.

Action: We modified the sentence to: "*The performance of WaterGAP 2.2d in terms of monthly streamflow time series at 1319 gauging station (Fig. 8) reaches a median NSE(KGE) of 0.52 (0.61).*"

R: Section 6.5. Can you attribute some of these improvements in model performance to specific changes made to the model?

A: Without an extensive experiment setup which would include simulations (and calibrations) of each modification in a stepwise manner, a scientifically based answer is not possible, and we decided not to speculate.

Action: none

R: Conclusions. An additional development here could be improving the lakes/wetlands and reservoir regulation as you noted this being a limiting factor to good model performance in North America/Canada?

A: Thank you for the suggestion. Indeed, there are many areas for future development such as the improvements mentioned. However, all of these potential improvements require funding to do research and implementation, hence we have listed only those lines that are currently under development (see also Line 1018).

Action: none

- Replaced citation of ERA5 to the now published journal publication
- Acknowledgements: added „The publication of this article was funded by the Open Access Fund of the Leibniz Association"

[revised manuscript text omitted]

**Abbreviations**

| | |
|---|---|
| AAI | area actually irrigated |
| AEI | area equipped for irrigation |
| CRU | Climatic Research Unit |
| CFA | areal correction factor |
| CFS | station correction factor |
| CS | calibration status |
| CSR | Center of Space Research |
| CU | consumptive water use |
| FAO | Food and Agriculture Organization of the United Nations |
| GHM | global hydrological model |
| GIA | glacial isostatic adjustment |
| GIM | Global Irrigation Model |
| GLWD | Global Lakes and Wetlands Database |
| GMIA | Global Map of Irrigation Area |
| GPS | Global Positioning System |
| GRACE | Gravity Recovery And Climate Experiment |
| GRanD | Global Reservoir and Dam database |
| GRDC | Global Runoff Data Centre |
| GSFC | Goddard Space Flight Center |
| GVA | gross value added |
| GWSWUSE | Groundwater-Surface Water Use |
| HID | Historical Irrigation Data set |
| ICU | irrigation consumptive water use |
| ISIMIP | Inter-Sectoral Impact Model Intercomparison Project |
| JPL | Jet Propulsion Laboratory |
| lg | global lakes |
| ll | local lakes |
| LResW | lakes, man-made reservoirs and wetlands |

| netCDF | network Common Data Form |
| res | global man-made reservoirs |
| TWSA | total water storage anomalies |
| WaterGAP | Water - Global Assessment and Prognosis |
| WGHM | WaterGAP Global Hydrology Model |
| wg | global wetlands |
| wl | local wetlands |
| WU | withdrawal water use |

**Symbols used**

Table S1: Symbols used for WaterGAP variables and parameters in the main paper. Note that there are many other model variables and parameters (e.g., downward shortwave and downward longwave radiation is also a model input).

| Symbol | description | unit | equations |
|--------|-------------|------|-----------|
| **Model input: spatially distributed input variables** | | | |
| $P$ | precipitation | $\mathrm{mm\,d^{-1}}$ | 2, 3, 22, 24 |
| $T$ | daily air temperature | $^\circ\mathrm{C}$ | 8, 10, 12, 13 |
| **Model input: spatially distributed input data (temporally constant)** | | | |
| $A_{cont}$ | continental area | $\mathrm{m^2}$ | 35 |
| $A_{max}$ | maximum extent of the water body | $\mathrm{m^2}$ | 23, 26 |
| $D_{r,bf}$ | river depth at bankfull conditions | m | 33 |
| $f_{d,lc}$ | fraction of deciduous plants | – | 5 |
| $S_{res,max}$ | storage capacity of reservoirs/regulated lakes | $\mathrm{m^3}$ | 29 |
| **Spatially distributed model parameters derived from spatially distributed input data (some derived using model parameters)** | | | |
| $f_g$ | groundwater recharge factor | – | 19 |
| $R_{g_{max}}$ | soil-texture specific maximum groundwater recharge | $\mathrm{mm\,d^{-1}}$ | 19 |
| $s$ | river bed slope | $\mathrm{m\,m^{-1}}$ | 32 |
| $S_{c,max}$ | maximum canopy storage | mm | 3, 4, 6 |
| $S_{s,max}$ | maximum soil water content | mm | 17, 18 |
| $S_{l,max}$ | maximum storage of the lake | $\mathrm{m^3}$ | 24 |
| $S_{r,max}$ | maximum volume of the river | $\mathrm{m^3}$ | 33 |
| $S_{res,w,max}$ | maximum storage of the reservoir/regulated lake and wetland | $\mathrm{m^3}$ | 25 |
| $l$ | river length | m | 31, 33, 34 |
| $L_{max}$ | maximum value of $L$ | – | 5 |
| $L_{min}$ | minimum value of $L$, | – | 5 |
| $W_{r,bf}$ | river top width at bankfull conditions | m | 33 |
| **Model output: storages** | | | |
| $S_c$ | canopy storage | mm | 2, 3, 6 |
| $S_g$ | groundwater storage | $\mathrm{m^3}$ | 20, 21 |
| $S_l$ | volume of water stored in the lake | $\mathrm{m^3}$ | 24 |

| $S_{l,res,w}$ | volume of water stored in the water body | m³ | 22 |
|---|---|---|---|
| $S_{ll,wl}$ | local lake or local wetland storage | m³ | 27 |
| $S_{lg,wg}$ | global lake or global wetland storage | m³ | 28 |
| $S_r$ | volume of water stored in the river | m³ | 30, 31, 34 |
| $S_{res}$ | reservoir/regulated lake storage | m³ | 29 |
| $S_{res,w}$ | volume of water stored in reservoir/regulated lake or wetland | m³ | 25 |
| $S_s$ | soil water storage | mm | 15, 17, 18 |
| $S_{sn}$ | snow storage | mm | 11, 13, 14 |

Model output: flows

| $E_c$ | evaporation from the canopy | $\mathrm{mm\,d^{-1}}$ | 2, 6, 14, 17 |
|---|---|---|---|
| $E_s$ | actual evapotranspiration from the soil | $\mathrm{mm\,d^{-1}}$ | 15, 17 |
| $E_{sn}$ | sublimation | $\mathrm{mm\,d^{-1}}$ | 11, 14 |
| $ICU$ | irrigation consumptive water use (crop specific) | $\mathrm{mm\,d^{-1}}$ | 1 |
| $M$ | snowmelt | $\mathrm{mm\,d^{-1}}$ | 11, 13, 16 |
| $NA_g$ | net abstraction from groundwater | $\mathrm{m^3\,d^{-1}}$ | 20 |
| $P_{sn}$ | the part of $P_t$ that falls as snow | $\mathrm{mm\,d^{-1}}$ | 11, 12, 16 |
| $Q_g$ | groundwater discharge | $\mathrm{m^3\,d^{-1}}$ | 20 , 21 |
| $Q_{r,out}$ | streamflow or river discharge | $\mathrm{m^3\,d^{-1}}$ | 30, 31, 35 |
| $R$ | net radiation | $\mathrm{mm\,d^{-1}}$ | 7 |
| $R_g$ | diffuse groundwater recharge | $\mathrm{mm\,d^{-1}}$ | 19, 20 |
| $R_{g_{l,res,w}}$ | point groundwater recharge from surface water bodies | $\mathrm{m^3\,d^{-1}}$ | 20, 22, 26 |
| $R_l$ | runoff from land | $\mathrm{mm\,d^{-1}}$ | 15, 18, 19 |
| $R_{nc}$ | net cell runoff | $\mathrm{mm\,d^{-1}}$ | 35 |

Model parameters

| $\alpha$ | Priestley-Taylor parameter | — | 7 |
|---|---|---|---|
| $a$ | outflow exponent for local lakes and local wetlands | — | 27 |
| $c_{e,lc}$ | reduction factor for evergreen plants per land cover type | — | 5 |
| $D_F$ | land-cover specific degree-day factor | $\mathrm{mm\,d^{-1}\,{}^\circ C}$ | 13 |
| $E_{pot,max}$ | maximum potential evapotranspiration | $\mathrm{mm\,d^{-1}}$ | 17 |
| $\gamma$ | runoff coefficient | — | 18 |
| $g$ | psychrometric constant | $\mathrm{k\,Pa\,{}^\circ C^{-1}}$ | 7, 9 |
| $k$ | surface water outflow coefficient | $\mathrm{d^{-1}}$ | 27, 28 |
| $k_g$ | globally constant groundwater discharge coefficient | $\mathrm{d^{-1}}$ | 21 |

| | | | |
|---|---|---|---|
| $K_{gw_{l,res,w}}$ | groundwater recharge constant below LResW | $\mathrm{m\,d^{-1}}$ | 26 |
| $k_{rele}$ | reservoir release factor | $-$ | 29 |
| $l_h$ | latent heat | $\mathrm{MJkg^{-1}}$ | 9, 10 |
| $m_c$ | canopy storage parameter | mm | 4 |
| $p$ | reduction exponent | $-$ | 24, 25 |
| $p_a$ | atmospheric pressure of the standard atmosphere | kPa | 9 |
| $r$ | reduction factor for surface water bodies | $-$ | 23, 24, 25, 26 |
| $s_a$ | slope of the saturation vapour pressure-temperature relationship | $\mathrm{kPa\,^{\circ}C^{-1}}$ | 7, 8 |
| $T_f$ | snow freeze temperature | $^{\circ}\mathrm{C}$ | 12 |
| $T_m$ | snow melt temperature | $^{\circ}\mathrm{C}$ | 12 |

**Internal variables**

| | | | |
|---|---|---|---|
| $A$ | global (or local) water body surface area | $\mathrm{m^2}$ | 22, 23 |
| $D_r$ | river water depth | m | 34 |
| $E_{pot}$ | potential evapotranspiration | $\mathrm{mm\,d^{-1}}$ | 6, 7, 14, 17, 22 |
| $E_{pot_c}$ | crop-specific optimal evapotranspiration | $\mathrm{mm\,d^{-1}}$ | 1 |
| $L$ | one-side leaf area index | $-$ | 5 |
| $n$ | river bed roughness | $-$ | 32 |
| $NA_{l,res}$ | net abstraction from the lakes and reservoirs | $\mathrm{m^3\,d^{-1}}$ | 22 |
| $NA_{s,r}$ | net abstraction of surface water from the river | $\mathrm{m^3\,d^{-1}}$ | 30 |
| $P_{eff}$ | effective precipitation | $\mathrm{mm\,d^{-1}}$ | 15, 16, 18 |
| $P_{irri,eff}$ | effective precipitation for irrigation | $\mathrm{mm\,d^{-1}}$ | 1 |
| $P_t$ | throughfall (fraction of $P$ that reaches the soil) | $\mathrm{mm\,d^{-1}}$ | 2, 3, 12, 16 |
| $Q_{in}$ | inflow into water body from upstream | $\mathrm{m^3\,d^{-1}}$ | 22 |
| $Q_{out}$ | outflow from the water body to other surface water bodies including river storage | $\mathrm{m^3\,d^{-1}}$ | 22, 27, 28 |
| $Q_{r,in}$ | inflow into the river compartment | $\mathrm{m^3\,d^{-1}}$ | 30, 35 |
| $R_h$ | hydraulic radius of the river channel | m | 32 |
| $v$ | river flow velocity | $\mathrm{m\,d^{-1}}$ | 31, 32 |
| $W_{r,bottom}$ | river bottom width | m | 33, 34 |

**WaterGAP application fields**

[revised manuscript text omitted]